# CausalAffect: Causal Discovery for Facial Affective Understanding

## Abstract

Understanding human affect from facial behavior requires not only accurate recognition but also structured reasoning over the latent dependencies that drive muscle activations and their expressive outcomes. Although Action Units (AUs) have long served as the foundation of affective computing, existing approaches rarely address how to infer psychologically plausible causal relations between AUs and expressions directly from data. We propose **CausalAffect**, the first framework for causal graph discovery in facial affect analysis. CausalAffect models AU→AU and AU→Expression dependencies through a two-level polarity and direction aware causal hierarchy that integrates population-level regularities with sample-adaptive structures. A feature-level counterfactual intervention mechanism further enforces true causal effects while suppressing spurious correlations. Crucially, our approach requires neither jointly annotated datasets nor handcrafted causal priors, yet it recovers causal structures consistent with established psychological theories while revealing novel inhibitory and previously uncharacterized dependencies. Extensive experiments across six benchmarks demonstrate that CausalAffect advances the state of the art in both AU detection and expression recognition, establishing a principled connection between causal discovery and interpretable facial behavior. All trained models and source code will be released upon acceptance.

## 1 Introduction

Understanding human emotions from facial behavior is a core task in affective computing, with broad implications for human–computer interaction, assistive technologies, and robotics (Picard, 2000). A key bridge from facial behavior to emotion understanding is the Facial Action Coding System (FACS) (Ekman & Friesen, 1978), a widely adopted framework that decomposes facial expressions into a set of Action Units (AUs), which are elemental muscle movements associated with facial activity. Capturing co-activation structures, both among AUs (i.e., AU→AU interactions) and between AUs and expressions (i.e., AU→Expression linkages), has been shown to significantly improve *Action Unit Detection (AUD)* and *Facial Expression Recognition (FER)*. Existing approaches for modeling such dependencies fall into three paradigms: **(i) Cognitive prior-based methods**, which incorporate expert-defined co-activation patterns from psychological studies (Du et al., 2014); **(ii) Data-driven learning methods**, which leverage relational inductive biases (e.g., graph neural networks, GNNs) to learn dependencies from data (Song et al., 2021a; Luo et al., 2022; Liu et al., 2020); and **(iii) Statistical co-occurrence approaches**, which infer relational patterns from co-activation statistics in jointly annotated datasets (Song et al., 2015; Kollias & Zafeiriou, 2021; Kollias et al., 2024).

Although grounded in different principles, all above paradigms suffer from four key limitations: **(i) Lack of psychological plausibility:** As evidenced in Fig. 4 (GNN), data-driven learning approaches often induce entangled patterns that, while superficially interpretable, diverge from human-aligned causal structures. Similarly, statistical co-occurrence approaches only reflects dataset-specific frequencies rather than genuine causal mechanisms and are susceptible to demographic biases (e.g., culture, age, gender); **(ii) Dependence on joint annotations:** Most methods rely on datasets with joint annotations, which are scarce and rarely overlapping, leaving most single-task datasets underutilized; **(iii) Limited to rigid global relations:** Although cognitive studies emphasize psychological plausibility, they restricted to fixed population-level relations, overlooking context-adaptive patterns. For example, FACS (Ekman & Friesen, 1978) links AU6 (cheek raiser) and AU12 (lip-corner puller) to happiness, yet both *Duchenne* and *Social* smiles involve these AUs but convey distinct meanings

(see analysis in Section 5.3); **(iv) Neglect of directionality and inhibitory effects:** All these studies often treat AU relations as symmetric, especially in AU→AU interactions, whereas in practice muscle activations are directional (e.g., AU9 (nose wrinkler) mechanically triggering AU10 (upper lip raiser), but not vice versa). They also rarely account for inhibitory relations, where certain AUs weaken or mask emotional expressions (e.g., AU6 reducing Sadness or AU12 diminishing Disgust), thereby limiting the modeling of the full spectrum of affective dynamics.

To address these limitations, we introduce **CausalAffect**, a unified graph framework for discovering *directed (asymmetric), polarity-aware (excitatory and inhibitory), and both population-level and sample-adaptive* causal relationships between AUs and expressions. Operating in a weakly supervised setting, CausalAffect requires neither co-annotated labels nor handcrafted priors. A key innovation is the **feature-level counterfactual intervention**, which enables causal discovery by perturbing disentangled AU representations. Unlike prior counterfactual methods that manipulate high-level attributes (e.g., age, gender) in image space using generative models (Cheong et al., 2022; Liu et al., 2024; Ramesh et al., 2024)—approaches that are computationally expensive, semantically coarse, and ill-suited to structured causal reasoning—CausalAffect intervenes directly in the AU latent space, yielding semantically faithful, computationally efficient, and interpretable causal modeling. In summary, our contributions are threefold:

- **Causal Discovery.** To the best of our knowledge, this is the first data-driven framework that learns asymmetric, polarity-aware causal dependencies in both AU→AU and AU→Expression relations, yielding structures that are causally grounded and psychologically plausible.
- **CausalAffect Framework.** We introduce CausalAffect, a unified causal discovery framework that jointly captures population-level and sample-adaptive causal structure through global graph induction, sample-adaptive graph refinement, and feature-level counterfactual intervention.
- **Feature-level Counterfactual Intervention.** We propose a bottleneck-based intervention mechanism that perturbs latent AU features directly, eliminating the need for image-level synthesis and its associated complexity. This design enables efficient and scalable counterfactual reasoning, while injecting causal supervision to promote structurally meaningful dependency learning.
- **State-of-the-Art Performance.** Extensive experiments on six benchmarks, spanning both AU detection and expression recognition in image and video settings, demonstrate that CausalAffect achieves state-of-the-art results under diverse training configurations.

## 2 RELATED WORK

Existing work on facial affect analysis explores both structured relation modeling and causal or counterfactual reasoning, but each direction has notable limitations. Prior methods model AU–AU and AU–expression dependencies using expert-curated FACS rules (Ekman & Friesen, 1978; Du et al., 2014) or data-driven graph and attention architectures (Song et al., 2021b; Wang et al., 2023b; Liu et al., 2020), yet these approaches often produce dense, undirected affinity graphs that lack directionality, polarity, and human-aligned interpretability (Kakkad et al., 2023; Wang et al., 2023a). Co-occurrence–based relations (Eleyan & Demirel, 2009; Kollias & Zafeiriou, 2021) suffer from dataset bias (Chen & Joo, 2021; Dominguez-Catena et al., 2024) and require joint AU–expression labels. Meanwhile, causal and counterfactual models in computer vision (Cheong et al., 2022; Chen et al., 2022; Li et al., 2024c; Pan & Bareinboim, 2024) predominantly intervene in pixel-level or coarse semantic spaces, often relying on generative models (Melistas et al., 2024; Ramesh et al., 2024) and assuming fixed global causal structures (Gao et al., 2021). None of these methods address the need for directed, polarity-aware, and sample-adaptive causal reasoning. Our work differs by performing causal discovery, enabling weakly supervised multi-dataset learning, and introducing efficient feature-level counterfactual interventions in a disentangled AU latent space.

## 3 PROBLEM FORMULATION

The objective is to uncover *causally grounded* and *psychologically plausible* structures that govern dependencies among AUs and between AUs and expressions, without relying on co-occurring label annotations. We assume access to two type of disjoint heterogenious datasets: AU-labeled dataset $\mathcal{D}_{\text{AU}} = \{(I_k^{\text{AU}}, y_k^{\text{AU}})\}_{k=1}^{K_{\text{AU}}}$ with multi-label annotations $y_k^{\text{AU}} \in \{0, 1\}^{N_{\text{AU}}}$, and expression-labeled

dataset $\mathcal{D}_{\text{Expr}} = \{(I_k^{\text{Expr}}, y_k^{\text{Expr}})\}_{k=1}^{K_{\text{Expr}}}$ with categorical labels $y_k^{\text{Expr}} \in \{1, \ldots, N_{\text{Expr}}\}$. Each image $I$ is encoded into a set of disentangled AU-level embeddings $\mathbf{F}_{\text{AU}} = \{\mathbf{f}_{\text{AU}}^{(i)}\}_{i=1}^{N_{\text{AU}}}$, where $\mathbf{f}_{\text{AU}}^{(i)} \in \mathbb{R}^d$ denotes the representation of the $i$-th AU. We formalize causal discovery as learning two types of directed graphs: **AU→AU** graphs, where target nodes are causally aggregated AU embeddings capturing inter-AU dependencies, and **AU→Expression** graphs, where target nodes are expression representations causally aggregated from AU features. For each type, we maintain two levels of graphs: a **global causal graph** $\mathcal{G}_g$ that models population-level dependencies consistent with cognitive and psychological literature (ensuring psychological plausibility), and a **sample-adaptive causal graph** $\mathcal{G}_s(I)$ that refines the global relations to capture instance-specific causal patterns. Depending on the dataset, the framework can be instantiated for AU-only graph discovery (AU detection), AU→Expression graph discovery (expression recognition), or jointly in a multi-task setting.

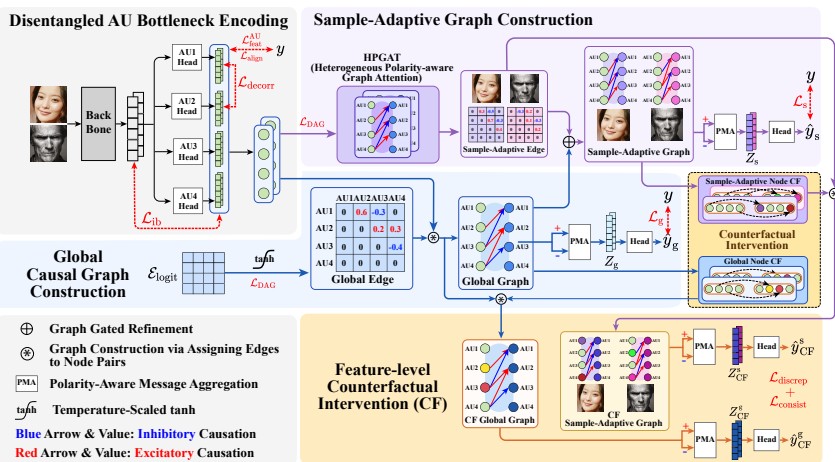

Figure 1: Overview of **CausalAffect**, consisting of four key components: (1) **Disentangled AU Bottleneck Encoding**, which extracts semantically independent AU features; (2) **Global Causal Graph Construction**, which learns a polarity-aware acyclic graph capturing population-level dependencies; (3) **Sample-Adaptive Graph Refinement**, which adjusts the global graph for each instance via residual adaptation; and (4) **Feature-Level Counterfactual Intervention**, which enforces causal inductive bias through latent perturbations and factual–counterfactual consistency. The figure illustrates the AU→AU branch; the AU→Expression branch is analogous and omitted for brevity.

## 4 CAUSALAFFECT FRAMEWORK

In this section, we present the **CausalAffect** framework, the overall structure is illustrated in Fig. 1.

### 4.1 DISENTANGLED AU BOTTLENECK ENCODING

The first stage of CausalAffect focuses on learning semantically meaningful and explicitly disentangled AU features, which form the foundation for reliable causal discovery. Disentanglement is essential because each AU representation must isolate the information specific to that AU, without identity cues, shared nuisance factors, or spurious co-variations. By ensuring that AU features encode only their intended semantic content, this stage removes confounders, suppresses correlated noise, and enables both the construction of meaningful AU–AU and AU–Expression causal structures and the execution of effective feature-level counterfactual interventions.

To achieve this, we first obtain global visual features from the input image $I$ and then project them into AU-specific representations. Our framework is backbone-agnostic, we adopt ConvNeXt followed by a Squeeze-and-Excitation (SE) block (Hu et al., 2018) to produce a compact embedding $z_{\text{img}} \in \mathbb{R}^D$. On top of the embedding, we introduce $N_{\text{AU}}$ lightweight projection heads $\{\phi_i\}_{i=1}^{N_{\text{AU}}}$, where each $\phi_i : \mathbb{R}^D \to \mathbb{R}^d$ maps $z_{\text{img}}$ into an AU-specific feature $\mathbf{f}_{\text{AU}}^{(i)}$. This yields a feature matrix $\mathbf{F}_{\text{AU}} = [\phi_1(z_{\text{img}}), \ldots, \phi_{N_{\text{AU}}}(z_{\text{img}})]^\top \in \mathbb{R}^{N_{\text{AU}} \times d}$. To ensure that each AU representation is discriminative. Each $\mathbf{f}_{\text{AU}}^{(i)}$ is further fed into a binary classifier to predict the activation $\hat{y}_{\text{AU}}^{(i)}$, supervised by the binary cross-entropy (BCE) loss: $\mathcal{L}_{\text{feat}}^{\text{AU}} = \frac{1}{N_{\text{AU}}} \sum_{i=1}^{N_{\text{AU}}} \text{BCE}(\hat{y}_{\text{AU}}^{(i)}, y_{\text{AU}}^{(i)})$.

**Information Bottleneck via HSIC-Based Disentanglement.** To ensure that each AU-specific representation preserves only task-relevant information while discarding confounding factors (e.g., age, gender, identity) embedded in $z_{\text{img}}$, we adopt the Deep Information Bottleneck principle (Tishby et al., 2000; Tishby & Zaslavsky, 2015) for disentanglement. This principle imposes two core constraints: **(i) *Minimize* $\mathcal{L}_{ib}$:** the statistical dependence between AU representation $\mathbf{f}_{\text{AU}}^{(i)}$ and $z_{\text{img}}$ to suppress task-irrelevant signals; **(ii) *Maximize* $\mathcal{L}_{align}$:** the dependence between $\mathbf{f}_{\text{AU}}^{(i)}$ and its corresponding label to retain discriminative semantics. Beyond these, we introduce an additional objective: **(iii) *Minimize* $\mathcal{L}_{decorr}$:** the statistical dependence between different AU representations $\mathbf{f}_{\text{AU}}^{(i)}$ and $\mathbf{f}_{\text{AU}}^{(j)}$ $(i \neq j)$ to reduce representational redundancy. This constraint promoting semantically disentangled and structurally independent features, which is essential for reliable causal attribution. Since direct mutual information estimation is computationally unstable and difficult to optimize, we instead adopt the HSIC (Gretton et al., 2005) as a reliable, non-parametric surrogate for all dependency objectives. HSIC enables efficient and theoretically grounded disentanglement through kernel-based statistical measurement. The three objectives defined as follows (see **Appendix E** for HSIC computation):

$$\mathcal{L}_{\text{ib}} = \frac{1}{N_{\text{AU}}} \sum_i \mathcal{H}(\mathbf{f}_{\text{AU}}^{(i)}, z_{\text{img}}), \quad \mathcal{L}_{\text{align}} = -\frac{1}{N_{\text{AU}}} \sum_i \mathcal{H}(\mathbf{f}_{\text{AU}}^{(i)}, \tilde{y}_{\text{AU}}^{(i)}), \quad \mathcal{L}_{\text{decorr}} = \frac{1}{N_{\text{AU}}^2} \sum_{i \neq j} \mathcal{H}(\mathbf{f}_{\text{AU}}^{(i)}, \mathbf{f}_{\text{AU}}^{(j)}). \quad (1)$$

where $\mathcal{H}(\cdot, \cdot)$ denotes the HSIC dependence measure, $N_{\text{AU}}$ denotes the number of AUs; and $\tilde{y}_{\text{AU}}^{(i)}$ represents the (pseudo-)label for AU $i$, which uses ground-truth labels when available and model predictions otherwise. Higher HSIC values indicate stronger dependence.

**Total AU Bottleneck Loss.** The final objective for learning disentangled AU bottleneck features is:

$$\mathcal{L}_{\text{AU}} = \mathcal{L}_{\text{feat}}^{\text{AU}} + \lambda_{\text{ib}} \cdot \mathcal{L}_{\text{ib}} + \lambda_{\text{decorr}} \cdot \mathcal{L}_{\text{decorr}} + \lambda_{\text{align}} \cdot \mathcal{L}_{\text{align}}, \quad (2)$$

where $\lambda_{\text{ib}}, \lambda_{\text{decorr}}, \lambda_{\text{align}}$ control the strength of each regularization term.

## 4.2 GLOBAL CAUSAL GRAPH CONSTRUCTION (GC GRAPH)

The second stage of CausalAffect aims to capture stable, population-level causal dependencies for both *AU→AU* and *AU→Expression* interactions. This is achieved by learning a shared adjacency matrix whose edges are directed (asymmetric) and polarity-aware, two properties that are essential for reflecting the true causal nature of facial behavior. AU→Expression relations are inherently **unidirectional and DAG-structured**, as expressions arise *as downstream outcomes* of underlying AU activations. In contrast, AU→AU relations often exhibit **positive reciprocal causation**, where one muscle unit reinforces or stabilizes the activation of another; therefore, we incorporate a soft-DAG constraint as a *regularization mechanism* that encourages the learned AU→AU structure to remain **sparse, directional, and dominated by strong causal influences**. Polarity-aware edges are equally crucial, since facial musculature expresses both **excitatory** (co-activating) and **inhibitory** (suppressing or dampening) effects.

Formally, we define the global graph as $\mathcal{G}_{\text{g}} = (\mathcal{N}_{\text{g, source}}, \mathcal{N}_{\text{g, target}}, \mathcal{E}_{\text{g}})$, where the source nodes $\mathcal{N}_{\text{g, source}}$ are the disentangled AU representations $\mathbf{F}_{\text{AU}}$ shared across both graph types. The target nodes $\mathcal{N}_{\text{g, target}}$ correspond to representations causally aggregated from $\mathcal{N}_{\text{g, source}}$ (expressions in AU→Expression, AUs in AU→AU). The edge weights $\mathcal{E}_{\text{g}} \in (-1, 1)$ encode both the polarity and magnitude of causal influence, thereby specifying the directed topology from source to target. To parameterize these edges, we introduce a learnable matrix $\mathcal{E}_{\text{logit}} \in \mathbb{R}^{N_{\text{target}} \times N_{\text{source}}}$, where each entry $\mathcal{E}_{\text{logit}}^{(j,i)}$ denotes the raw, unbounded causal strength from source node $i$ to target node $j$, with $N_{\text{g, source}}$ and $N_{\text{g, target}}$ is the number of source and target nodes. To obtain bounded edge matrix, we apply a temperature-scaled hyperbolic tangent transformation:

$$\mathcal{E}_{\text{g}}^{(j,i)} = \tanh\left(\tau \cdot \mathcal{E}_{\text{logit}}^{(j,i)}\right), \quad \tau = \exp(T_{\text{g}}), \quad (3)$$

where $T_{\text{g}}$ is a learnable scalar temperature. This maps the logits into the interval $(-1, 1)$, producing a signed edge matrix $\mathcal{E}_{\text{g}} \in \mathbb{R}^{N_{\text{target}} \times N_{\text{source}}}$. The sign of each entry indicates whether the influence is excitatory ($> 0$) or inhibitory ($< 0$), while the magnitude encodes causal strength.

**Polarity-Aware Message Aggregation.** To disentangle polarity effects, we decompose the signed edge into excitatory and inhibitory components:

$$\mathcal{E}_{\text{g}}^+ = \mathbb{I}[\mathcal{E}_{\text{g}} > 0] \odot \mathcal{E}_{\text{g}}, \quad \mathcal{E}_{\text{g}}^- = \mathbb{I}[\mathcal{E}_{\text{g}} < 0] \odot (-\mathcal{E}_{\text{g}}), \quad (4)$$

where $\mathbb{I}[\cdot]$ is the element-wise indicator function and $\odot$ denotes the Hadamard product. To aggregate source AU nodes into target nodes (AU or Expression) through graph, we first project the disentangled AU features into two value spaces, i.e., $V_g^+ = \mathbf{F}_{\text{AU}} W_g^+$ and $V_g^- = \mathbf{F}_{\text{AU}} W_g^-$, where $W_g^+, W_g^- \in \mathbb{R}^{d \times d_{\text{g}}}$ are the positive and negative projection matrices, $d_{\text{g}}$ denote the dimensionality of the projected representation. The aggregated representation is computed as $Z_{\text{g}} = \mathcal{E}_{\text{g}}^+ V_g^+ + \mathcal{E}_{\text{g}}^- V_g^- \in \mathbb{R}^{N_{\text{target}} \times d_{\text{g}}}$.

**AU→AU Homogeneous Soft DAG Constraint.** From a psychological and muscular perspective, facial AUs are driven by underlying muscle activations that follow directional and structured pathways rather than arbitrary feedback loops. Unlike the AU→Expression graph, which is heterogeneous and inherently acyclic due to its one-way causal semantics, the AU→AU graph is homogeneous and permits self-contained interactions among AUs, making it especially prone to cycles during learning. To mitigate this, we introduce a differentiable DAG constraint to regularize the AU→AU graph. Instead of enforcing strict acyclicity, we implement it as a **soft** regularization, guiding the model toward interpretable, directional dependencies while preserving flexibility to capture reciprocal or context-dependent AU interactions. Specifically, for the learned AU→AU edge matrix $\mathcal{E}_g^{\text{AU}} \in \mathbb{R}^{N_{\text{AU}} \times N_{\text{AU}}}$, we enforce the constraint using the trace (denoted $\text{tr}$) of its matrix exponential:

$$\mathcal{L}_{\text{DAG}} = \text{tr}\left(\exp\left(\mathcal{E}_{\text{g}}^{\text{AU}} \odot \mathcal{E}_{\text{g}}^{\text{AU}}\right)\right) - N_{\text{AU}}, \tag{5}$$

where $\odot$ is the Hadamard product. This DAG constraint not only to the global AU→AU graph, but also to the sample-adaptive AU→AU graphs introduced in the next section.

**Global Graph Loss.** For AU→AU graph, the prediction is $\hat{y}_{\text{AU}} = \Phi_{\text{AU}}(Z_{\text{g}}^{\text{AU}})$; for AU→Expression graph, the prediction is $\hat{y}_{\text{Expr}} = \Phi_{\text{Expr}}(Z_{\text{g}}^{\text{Expr}})$, where $Z_{\text{g}}^{\text{AU}} \in \mathbb{R}^{N_{\text{AU}} \times d_{\text{g}}}$ and $Z_{\text{g}}^{\text{Expr}} \in \mathbb{R}^{N_{\text{Expr}} \times d_{\text{g}}}$ are the causally aggregated features, and $\Phi_{\text{AU}}, \Phi_{\text{Expr}}$ denote task-specific classifier heads. The graph learning is supervised using the corresponding ground-truth labels $y^{\text{AU}}$ and $y^{\text{Expr}}$ via the $\mathcal{L}_{\text{g}}$ loss defined as:

$$\mathcal{L}_{\text{g}} = \begin{cases} \mathcal{L}_{\text{g}}^{\text{AU}} = \text{BCE}(\hat{y}_{\text{g}}^{\text{AU}}, y^{\text{AU}}) + \lambda_{\text{DAG}}^{\text{g}} \cdot \mathcal{L}_{\text{DAG}}^{\text{g}}, & \text{AU→AU Graph}, \\ \mathcal{L}_{\text{g}}^{\text{Expr}} = \text{CE}(\hat{y}_{\text{g}}^{\text{Expr}}, y^{\text{Expr}}), & \text{AU→Expression Graph}, \end{cases} \tag{6}$$

where $\text{BCE}(\cdot)/\text{CE}(\cdot)$ denote binary/categorical cross-entropy, $\lambda_{\text{DAG}}$ controls acyclicity constraint.

### 4.3 Sample-Adaptive Causal Graph Construction (SAC Graph)

While the global causal graph captures population-level patterns, it may overlook context-specific dependencies (e.g., Eastern vs. Western display norms, or cases where *Social* and *Duchenne* smiles share AU patterns but stem from different causal pathways). Cross-cultural studies by Jack (Jack et al., 2012; 2014) show that facial expressions combine universal AU configurations with culture- and individual-specific variations. For instance, the core AU12+AU6 smile pattern is shared globally, yet Western smiles often add AUs such as AU7, AU25, and AU26, whereas East Asian smiles modulate intensity mainly through the mouth region. A single fixed graph cannot represent such variability, motivating a model that integrates both global and sample-adaptive causal structures.

To address this, we introduce sample-adaptive causal graph $\mathcal{G}_{\text{s}}^{(k)} = (\mathcal{N}_{\text{s,source}}^{(k)}, \mathcal{N}_{\text{s,target}}^{(k)}, \mathcal{E}_{\text{s}}^{(k)})$, dynamically constructed for each input $I^{(k)}$. This graph refines the global structure via an adaptive edge residual, allowing the model to capture context-dependent variations. To instantiate $\mathcal{G}_{\text{s}}^{(k)}$, we design a multi-layer attention module termed *Heterogeneous Polarity-aware Graph Attention (HPGAT)*. Standard attention-based graph models such as GAT and GATv2 (Brody et al., 2021) cannot model the inherently **heterogeneous** and **polarity-dependent** causal mechanisms in AU→AU and AU→Expression reasoning, since they assume homogeneous node types and restrict attention weights to be non-negative, preventing the representation of inhibitory effects. HPGAT introduces three key advances: **(i) Heterogeneous node modeling**, enabling cross-type AU→Expr aggregation; **(ii) Polarity-aware edge learning**, capturing both excitatory and inhibitory causal effects within $(-1, 1)$; **(iii) Gated causal residual integration**, adaptively refining the global causal structure at the instance level. Together, these capabilities allow HPGAT to learn **sample-adaptive, polarity-aware causal graphs** that cannot be expressed by conventional homogeneous attention mechanisms.

For each sample $k$, HPGAT infers cross-node attentional affinities between $\mathcal{N}_{\text{s, target}}^{(k)}$ and $\mathcal{N}_{\text{s, source}}^{(k)}$. At the final layer, the learned edge attention is incorporated as a residual refinement to the global edge matrix $\mathcal{E}_{\text{g}}$ in Eq. 3, yielding the final sample-specific graph $\mathcal{E}_{\text{s}}^{(k)}$. Concretely, for both the AU→AU

and AU→Expression graphs, source AU nodes $\mathcal{N}_{\text{s, source}}^{(k)}$ serve as attention **Keys** and **Values** and are fixed across layers, i.e., $K_s^{(k,l)} = V_s^{(k,l)} = \mathcal{N}_{\text{s, source}}^{(k)} = \mathbf{F}_{\text{AU}}^{(k)}$. The **Queries** depend on the target node type $\mathcal{N}_{\text{s, target}}^{(k)}$. At the first layer ($l = 1$), for the AU→AU graph we set $Q^{(1)} = \mathbf{F}_{\text{AU}}^{(k)}$, whereas for the AU→Expression graph we initialize queries with learnable expression prototypes $\mathbf{P}_{\text{Expr}} \in \mathbb{R}^{N_{\text{Expr}} \times d}$. For subsequent layers ($l > 1$), queries are updated recursively as $Q^{(k,l)} = \mathcal{N}_{\text{s, target}}^{(l-1)}$. At layer $l$, the HP-GAT computes the signed polarity-aware edge matrix $\mathcal{E}_s^{(k,l)} \in \mathbb{R}^{N_{\text{target}} \times N_{\text{source}}}$ as:

$$\mathcal{E}_s^{(k,l)} = \tanh\left(\mathbf{a}^\top \sigma\left(W_Q^{(k,l)} Q^{(k,l)} + W_K^{(k,l)} K_s^{(k,l)}\right)\right), \tag{7}$$

where $\sigma(\cdot)$ denotes the LeakyReLU activation, $W_Q^{(k,l)}, W_K^{(k,l)} \in \mathbb{R}^{d \times d_h}$ are learnable projection matrices, $\mathbf{a} \in \mathbb{R}^{d_h}$ is a learnable attention vector, and $d_h$ denotes the hidden dimension of the projected feature space. The $\tanh(\cdot)$ bounds edge weights to $(-1, 1)$.

**Polarity-Aware Message Aggregation.** Analogous to Eq. 4, decompose causal polarity components:

$$\mathcal{E}_s^{+(k,l)} = \mathbb{I}\left[\mathcal{E}_s^{(k,l)} > 0\right] \odot \mathcal{E}_s^{(k,l)}, \quad \mathcal{E}_s^{-(k,l)} = \mathbb{I}\left[\mathcal{E}_s^{(k,l)} < 0\right] \odot \left(-\mathcal{E}_s^{(k,l)}\right). \tag{8}$$

The updated target node embedding $\mathcal{N}_{\text{s,target}}^{(k,l)} = Z_s^{(k,l)}$ are obtained by aggregating the polarity-aware edges over the value projections: $Z_s^{(k,l)} = \mathcal{E}_s^{+(k,l)} V_s^{(k,l)} W_V^{+(k,l)} + \mathcal{E}_s^{-(k,l)} V_s^{(k,l)} W_V^{-(k,l)}$, where $W_V^{+(k,l)}, W_V^{-(k,l)} \in \mathbb{R}^{d \times d_h}$ are learnable projections for excitatory and inhibitory messages.

**Sample-Adaptive Graph Gated Refinement.** At the final layer $L$, we refine the global edge structure $\mathcal{E}_{\text{g}}$ with the sample-specific residual edge $\mathcal{E}^{(k,L)}$ through a learnable gating mechanism. In particular, a non-linear projection layer $\mathcal{G}(\cdot)$ maps $\mathcal{E}_s^{(k,L)}$ into a scalar gating value, and fused edge computed as:

$$\mathcal{E}_{\text{s}}^{(k)} = \sigma\left(\mathcal{G}\left(\mathcal{E}_s^{(k,L)}\right)\right) \odot \mathcal{E}_{\text{g}} + \left(1 - \sigma\left(\mathcal{G}\left(\mathcal{E}_s^{(k,L)}\right)\right)\right) \odot \mathcal{E}_s^{(k,L)}, \tag{9}$$

where $\sigma(\cdot)$ denotes the sigmoid function. Following Eq. 8, the edge matrix is further decomposed into excitatory and inhibitory components, $\mathcal{E}_s^{+(k,L)}$ and $\mathcal{E}_s^{-(k,L)}$, and the final sample-adaptive target node are updated as $\mathcal{N}_{\text{s, target}}^{(k)} = Z_s^{(k,L)} = \mathcal{E}_s^{+(k,L)} V_s^{(k,L)} W_V^{+(k,L)} + \mathcal{E}_s^{-(k,L)} V_s^{(k,L)} W_V^{-(k,L)}$.

**Sample-Adaptive Graph Loss.** For the AU→AU graph, the prediction is $\hat{y}_s^{\text{AU}} = \Phi_{\text{AU}}(Z_s^{\text{AU}})$; for the AU→Expression branch, the prediction is $\hat{y}_s^{\text{Expr}} = \Phi_{\text{Expr}}(Z_s^{\text{Expr}})$, $\Phi_{\text{AU}}, \Phi_{\text{Expr}}$ denote task-specific classifier heads. The learning is supervised using the labels $y^{\text{AU}}$ and $y^{\text{Expr}}$ via the $\mathcal{L}_{\text{s}}$ loss defined as:

$$\mathcal{L}_{\text{s}} = \begin{cases} \mathcal{L}_{\text{s}}^{\text{AU}} = \text{BCE}(\hat{y}_s^{\text{AU}}, y^{\text{AU}}) + \lambda_{\text{DAG}}^{\text{s}} \cdot \mathcal{L}_{\text{DAG}}^{\text{s}}, & \text{AU→AU Graph}, \\ \mathcal{L}_{\text{s}}^{\text{Expr}} = \text{CE}(\hat{y}_s^{\text{Expr}}, y^{\text{Expr}}), & \text{AU→Expression Graph}, \end{cases} \tag{10}$$

where $\lambda_{\text{DAG}}^{\text{s}}$ controls the acyclicity regularization applied only to the AU→AU graph.

## 4.4 FEATURE-LEVEL COUNTERFACTUAL INTERVENTION

While the global and sample-adaptive graphs capture structural dependencies, they may still encode spurious correlations. To enforce relationships that are truly *causal* rather than merely *correlational*, we introduce a **Feature-level Counterfactual Intervention (CF)** module. The core principle is that if a source node genuinely exerts a causal influence on a target, then intervening on the source should induce a meaningful change in the target's representation and prediction, whereas intervening on non-causal nodes should produce little to no effect.

We realize this via a thresholded soft mask with edge-aware saliency, which perturbs source node embeddings conditioned on the graph. By contrasting model behaviors under factual and counterfactual conditions, the model receives consistency and discrepancy signals that guide graph refinement, reinforcing informative edges and suppressing spurious ones. Specifically, given a graph edge matrix $\mathcal{E} \in (-1, 1)^{N_{\text{target}} \times N_{\text{source}}}$ (from either global or sample-adaptive graphs), we compute a soft importance mask $\mathbf{M}_{\text{soft}} \in (0, 1)^{N_{\text{target}} \times N_{\text{source}}}$ as

$$\mathbf{M}_{\text{soft}} = \sigma\left(\gamma \cdot (|\mathcal{E}| - \boldsymbol{\theta})\right), \tag{11}$$

where $\boldsymbol{\theta} \in [0,1]^{N_{\text{target}}}$ is a learnable threshold vector, $\gamma$ is a sharpness factor, and $\sigma(\cdot)$ is the element-wise sigmoid. Larger $\gamma$ yields sharper distinctions between relevant and irrelevant source nodes. For each target node $j$, we extract a row vector $\mathbf{m}^{(j)} \in [0,1]^{N_{\text{source}}}$, and define two complementary intervention masks. The **consistency mask** is given by $\mathbf{m}_{\text{consist}}^{(j)} = 1 - \mathbf{m}^{(j)}$, which masks non-causal source nodes, while the **discrepancy mask** is defined as $\mathbf{m}_{\text{discrep}}^{(j)} = \mathbf{m}^{(j)}$, which masks causal nodes. Counterfactual interventions are implemented by injecting Gaussian perturbations:

$$\mathbf{F}_{\text{AU-CF}}^{(j,\star)} = \mathbf{F}_{\text{AU}} + \boldsymbol{\epsilon}^{(j)} \odot \mathbf{m}_{\star}^{(j)}, \quad \boldsymbol{\epsilon}^{(j)} \sim \mathcal{N}(0, \tau^2), \quad \star \in \{\text{consist}, \text{discrep}\}. \tag{12}$$

executing graph aggregation yields counterfactual target node representations $Z_{\text{CF}}^{(j,\star)}$ and predictions $\hat{y}_{\text{CF}}^{(j,\star)}$, $\tau$ controls the perturbation. We define two complementary losses to guide causal behavior:

**Consistency loss** $\mathcal{L}_{\text{consist}}$: Guarantees output invariance under non-causal source perturbations:

$$\mathcal{L}_{\text{consist}} = \frac{1}{N_{\text{target}}} \sum_{j=1}^{N_{\text{target}}} \left[ \delta_{\text{feat}} \cdot \underbrace{(1 - \cos(Z^{(j)}, Z_{\text{CF}}^{(j,\text{consist})}))}_{\text{feature consistency}} + \delta_{\text{logit}} \cdot \underbrace{\mathcal{D}_{\text{consist}}^{\text{logit}}(\hat{y}^{(j)}, \hat{y}_{\text{CF}}^{(j,\text{consist})})}_{\text{logit consistency}} \right], \tag{13}$$

**Discrepancy loss** $\mathcal{L}_{\text{discrep}}$: Ensures interventions on causal source nodes induce meaningful changes:

$$\mathcal{L}_{\text{discrep}} = \frac{1}{N_{\text{target}}} \sum_{j=1}^{N_{\text{target}}} \left[ \eta_{\text{feat}} \cdot \underbrace{\left(1 + \cos(Z^{(j)}, Z_{\text{CF}}^{(j,\text{discrep})})\right)}_{\text{feature discrepancy}} + \eta_{\text{logit}} \cdot \underbrace{\left(1 - \mathcal{D}_{\text{discrep}}^{\text{logit}}(\hat{y}^{(j)}, \hat{y}_{\text{CF}}^{(j,\text{discrep})})\right)}_{\text{logit discrepancy}} \right], \tag{14}$$

where $\delta_{\text{feat}}, \eta_{\text{feat}}$ and $\delta_{\text{logit}}, \eta_{\text{logit}}$ are the feature- and logit-level weights, $\cos(\cdot, \cdot)$ denotes cosine similarity, and $\mathcal{D}_{\text{consist}}^{\text{logit}}(\cdot, \cdot)$ is a logit-level discrepancy (MSE for AU, KL divergence for expression).

The counterfactual intervention loss, weighted by $\lambda_{\text{consist}}$ and $\lambda_{\text{discrep}}$, is defined as:

$$\mathcal{L}_{\text{cf}} = \lambda_{\text{consist}} \cdot \mathcal{L}_{\text{consist}} + \lambda_{\text{discrep}} \cdot \mathcal{L}_{\text{discrep}}. \tag{15}$$

### 4.5 UNIFIED TRAINING OBJECTIVE

CausalAffect is trained end-to-end under a weakly supervised setting, where AU and expression labels are disjoint across samples. To accommodate this, we apply prediction-related losses ($\mathcal{L}_{\text{feat}}^{\text{AU}}, \mathcal{L}_{\text{AU}}^{\text{g}}, \mathcal{L}_{\text{AU}}^{\text{s}}, \mathcal{L}_{\text{Expr}}^{\text{g}}, \mathcal{L}_{\text{Expr}}^{\text{s}}$) only to samples that contain the respective ground-truth annotations. In contrast, the representation and regularization objectives ($\mathcal{L}_{\text{ib}}, \mathcal{L}_{\text{decorr}}, \mathcal{L}_{\text{align}}, \mathcal{L}_{\text{cf}}$) are applied to all samples, regardless of label availability. This design enables **weakly supervised joint learning** across disjoint datasets. We define the total objective for each branch as follows:

$$\mathcal{L}_{\text{total}}^{\text{AU}} = \mathcal{L}_{\text{AU}} + \mathcal{L}_{\text{AU}}^{\text{g}} + \mathcal{L}_{\text{AU}}^{\text{s}} + \mathcal{L}_{\text{cf}}, \qquad \mathcal{L}_{\text{total}}^{\text{Expr}} = \mathcal{L}_{\text{Expr}}^{\text{g}} + \mathcal{L}_{\text{Expr}}^{\text{s}} + \mathcal{L}_{\text{cf}}. \tag{16}$$

The final training loss combines both branches: $\mathcal{L} = \mathcal{L}_{\text{total}}^{\text{AU}} + \lambda_{\text{Expr}} \cdot \mathcal{L}_{\text{total}}^{\text{Expr}}$, where $\lambda_{\text{Expr}}$ controls the relative weighting of the expression modeling task.

## 5 EXPERIMENTS

To demonstrate that CausalAffect generalizes from **static images** to **dynamic videos**, we evaluate on six datasets. For AU detection, we use three video datasets (BP4D, DISFA, GFT) and one large-scale in-the-wild image dataset (EmotioNet). For expression recognition, since video analysis primarily relies on identifying emotions from key frames, we instead adopt two large-scale in-the-wild image datasets (RAF-DB, AffectNet), providing greater diversity in environment, identity, and illumination.

Additional details are provided in the supplementary material, including **Appendix A:** Detailed Ablation Analysis, **Appendix B:** Sensitivity to AU Composition, **Appendix C:** Detailed Global Causal Relation Analysis, **Appendix D:** Extensive Sample-Adaptive Case Studies, **Appendix E:** HSIC computation, **Appendix F:** Implementation Details, Sensitivity Analysis of Regularization Parameters, and **Appendix G:** Training and Inference Efficiency.

Table 1: Comparison of CausalAffect with SOTA methods. Results are reported in F1-score (%); For RAF-DB average accuracy (%) is used. **SG** denotes single-dataset training; **+BP4D/DISFA/GFT/EmotioNet/RAF-DB** indicates joint training with the corresponding dataset; **+All** denotes training with all datasets. Statistical significance is evaluated using paired t-tests over five matched runs, CausalAffect achieves a significant improvement with $p = 0.03$ ($p < 0.05$ are considered statistically significant).

| Methods | BP4D (AU, Video DB) | | | | | | | | | | | | | DISFA (AU, Video DB) | | | | | | | | |
|---|---|---|---|---|---|---|---|---|---|---|---|---|---|---|---|---|---|---|---|---|---|---|
| | 1 | 2 | 4 | 6 | 7 | 10 | 12 | 14 | 15 | 17 | 23 | 24 | Avg. | 1 | 2 | 4 | 6 | 9 | 12 | 25 | 26 | Avg. |
| JÂA-Net (Shao et al., 2021a) | 53.8 | 47.8 | 58.2 | 78.5 | 75.8 | 82.7 | 88.2 | 63.7 | 43.3 | 61.8 | 45.6 | 49.9 | 62.4 | 62.4 | 60.7 | 67.1 | 41.1 | 45.1 | 73.5 | 90.9 | 67.4 | 63.5 |
| PIAP (Tang et al., 2021) | 54.2 | 47.1 | 54.0 | 79.0 | 78.2 | 86.3 | 89.5 | 66.1 | 49.7 | 63.2 | 49.3 | 52.0 | 64.1 | 50.2 | 51.8 | 71.9 | 50.6 | 54.5 | 79.7 | 94.1 | 57.2 | 63.8 |
| FAUT (Jacob & Stenger, 2021) | 51.7 | 49.3 | 61.0 | 77.8 | 79.5 | 82.9 | 86.3 | 67.6 | 51.9 | 63.0 | 43.7 | 56.3 | 64.2 | 46.1 | 48.6 | 72.8 | 56.7 | 50.0 | 72.1 | 90.8 | 55.4 | 61.5 |
| EmoLA (Li et al., 2024a) | 57.4 | 52.4 | 61.0 | 78.1 | 77.8 | 81.9 | 89.5 | 60.5 | 49.3 | 64.9 | 46.0 | 52.4 | 64.2 | 50.5 | 56.9 | 83.5 | 55.2 | 43.1 | 80.1 | 91.6 | 60.0 | 65.1 |
| FG-Net (Yin et al., 2024) | - | - | - | - | - | - | - | - | - | - | - | - | 64.3 | - | - | - | - | - | - | - | - | 65.4 |
| KSRL (Chang & Wang, 2022) | 53.3 | 47.4 | 56.2 | 79.4 | 80.7 | 85.1 | 89.0 | 67.4 | 55.9 | 61.9 | 48.5 | 49.0 | 64.5 | 60.4 | 59.2 | 67.5 | 52.7 | 51.5 | 76.1 | 91.3 | 57.7 | 64.5 |
| ReCoT (Li et al., 2023) | 51.5 | 47.8 | 58.9 | 79.2 | 80.2 | 84.9 | 88.4 | 61.5 | 53.3 | 64.6 | 51.8 | 55.4 | 64.8 | 51.3 | 36.2 | 66.8 | 50.1 | 52.4 | 78.8 | 95.3 | 69.7 | 62.6 |
| MEGraph (Luo et al., 2022) | 52.7 | 44.3 | 60.9 | 79.9 | 80.1 | 85.3 | 89.2 | 69.4 | 55.4 | 64.4 | 49.8 | 51.1 | 65.5 | 52.5 | 45.7 | 76.1 | 51.8 | 46.5 | 76.1 | 92.9 | 57.6 | 62.4 |
| CLEF (Zhang et al., 2023c) | 55.8 | 46.8 | 63.3 | 79.5 | 77.6 | 83.6 | 87.8 | 67.3 | 55.2 | 63.5 | 53.0 | 57.8 | 65.9 | 64.3 | 61.8 | 68.4 | 49.0 | 52.2 | 72.9 | 89.9 | 57.0 | 64.8 |
| MCM (Zhang et al., 2024) | 54.4 | 48.5 | 60.6 | 79.1 | 77.0 | 84.0 | 89.1 | 61.7 | 59.3 | 64.7 | 53.0 | 60.5 | 66.0 | 49.6 | 44.1 | 67.2 | 65.5 | 49.0 | 81.5 | 85.9 | 71.8 | 64.3 |
| MDHRM (Wang et al., 2024) | 54.6 | 49.7 | 61.0 | 79.9 | 79.4 | 85.4 | 88.5 | 67.8 | 56.8 | 63.2 | 50.9 | 55.4 | 66.1 | 65.4 | 60.2 | 75.2 | 50.2 | 52.4 | 74.3 | 93.7 | 58.2 | 66.2 |
| AUFormer (Yuan et al., 2024) | - | - | - | - | - | - | - | - | - | - | - | - | 66.2 | - | - | - | - | - | - | - | - | 66.4 |
| CausalAffect (SG) | 63.4 | 47.1 | 56.8 | 82.1 | 78.8 | 85.8 | 88.8 | 71.4 | 57.5 | 67.9 | 46.2 | 56.4 | 66.9±0.2 | 54.8 | 61.7 | 75.4 | 50.8 | 62.6 | 67.5 | 89.5 | 73.3 | 67.0±0.1 |
| CausalAffect (+GFT) | 63.7 | 41.4 | 60.7 | 80.0 | 76.8 | 85.7 | 88.5 | 71.5 | 55.9 | 67.0 | 51.4 | 57.1 | 66.6±0.1 | 56.7 | 66.1 | 77.0 | 68.3 | 63.7 | 72.2 | 91.9 | 68.6 | 70.6±0.3 |
| CausalAffect (+EmotioNet) | 67.1 | 43.6 | 66.0 | 80.1 | 79.1 | 84.8 | 88.9 | 71.1 | 55.6 | 66.6 | 47.5 | 58.8 | 67.4±0.3 | 68.1 | 63.2 | 77.6 | 64.1 | 74.0 | 69.3 | 83.7 | 68.7 | 71.1±0.2 |
| CausalAffect (+RAF-DB) | 63.2 | 45.7 | 61.4 | 81.2 | 79.1 | 85.1 | 88.5 | 71.0 | 55.3 | 66.5 | 51.3 | 56.5 | 67.1±0.1 | 65.6 | 65.2 | 73.4 | 56.5 | 61.6 | 70.5 | 89.2 | 74.4 | 69.5±0.2 |
| CausalAffect (+AffectNet) | 64.4 | 44.6 | 61.2 | 81.3 | 80.2 | 85.5 | 88.8 | 71.5 | 52.6 | 67.4 | 45.5 | 56.3 | 66.6±0.2 | 59.0 | 65.7 | 72.3 | 62.4 | 58.3 | 69.5 | 83.7 | 74.8 | 68.2±0.2 |
| CausalAffect (+All) | 65.3 | 42.1 | 58.0 | 81.0 | 78.8 | 85.0 | 87.9 | 71.0 | 56.3 | 66.0 | 48.7 | 60.3 | 66.7±0.2 | 72.5 | 63.7 | 80.0 | 51.9 | 67.1 | 75.0 | 92.6 | 69.0 | 71.5±0.2 |

| EmotioNet (AU, Image DB) | F1 |
|---|---|
| Res50 | 44.0 |
| ME-GraphAU (Luo et al., 2022) | 64.9 |
| JAA-Net (Shao et al., 2018) | 51.8 |
| AUNets (Romero et al., 2022) | 64.6 |
| FBNet (Kollias et al., 2019) | 54.0 |
| CTC (Zhou et al., 2023) | 64.4 |
| FUXI (Zhang et al., 2023b) | 65.4 |
| SITU (Liu et al., 2023) | 64.2 |
| CausalAffect (SG) | 66.4±0.1 |
| CausalAffect (+DISFA) | 66.0±0.3 |
| CausalAffect (+BP4D) | 65.4±0.1 |
| CausalAffect (+All) | 65.0±0.1 |

| GFT (AU, Video DB) | F1 |
|---|---|
| EAC-Net (Li et al., 2018) | 46.1 |
| TCAE (Li et al., 2019) | 44.2 |
| CDAU (Ertugrul et al., 2020) | 45.3 |
| ARL (Shao et al., 2019) | 50.1 |
| MoCo (He et al., 2020) | 52.4 |
| TR (Lu et al., 2020) | 54.7 |
| JÂA-Net (Shao et al., 2021a) | 53.7 |
| EmoCo (Sun et al., 2021) | 58.6 |
| EmoLA (Li et al., 2024b) | 62.1 |
| CausalAffect (SG) | 61.1±0.2 |
| CausalAffect (+DISFA) | 62.5±0.3 |
| CausalAffect (+BP4D) | 60.4±0.4 |
| CausalAffect (+All) | 62.4±0.3 |

| AffectNet / RAF-DB (Expr, Image DB) | AffN. | RAF. |
|---|---|---|
| VTFF (Ma et al., 2021) | 61.9 | 81.2 |
| MViT (Li et al., 2021) | 64.6 | 80.4 |
| FBNet (Kollias et al., 2021) | 65.0 | 78.0 |
| DACL (Farzaneh & Qi, 2021) | 65.2 | 80.4 |
| ARM (Shi et al., 2021) | 65.2 | 82.8 |
| Ad-Corre (Fard & Mahoor, 2022) | 63.4 | 79.0 |
| DDAMFN++ (Zhang et al., 2023a) | 67.4 | 84.6 |
| FRA (Gao & Patras, 2024) | 66.1 | 83.4 |
| CAGE (Wagner et al., 2024) | 66.4 | 83.2 |
| ExpLLM (Zhang et al., 2025) | 65.9 | - |
| LOFI (Lan et al., 2025) | 65.7 | 84.7 |
| CausalAffect (+DISFA) | 65.6±0.2 | 83.7±0.1 |
| CausalAffect (+BP4D) | 67.7±0.1 | 85.3±0.2 |
| CausalAffect (+All) | 66.5±0.1 | 84.9±0.2 |

## 5.1 COMPARISON OF CAUSALAFFECT WITH STATE-OF-THE-ART (SOTA)

**For AU detection**, CausalAffect surpasses all prior SOTA methods under the single-dataset setting (SG). Incorporating additional AU datasets (e.g., BP4D+EmotioNet, DISFA+GFT) further improves performance by expanding AU categories and enabling transferable causal structures, while auxiliary supervision from expression datasets (e.g., BP4D+RAF-DB, DISFA+AffectNet) brings additional gains. **For expression recognition**, CausalAffect outperforms SOTA methods on AffectNet and RAF-DB without expression-specific encoders—predictions are causally derived from AU features. Richer AU supervision (e.g., BP4D 12 AUs vs. DISFA 8 AUs) provides stronger causal priors, enhancing both accuracy and interpretability (see **Appendix B** for sensitivity to AU composition).

## 5.2 GLOBAL CAUSAL RELATION ANALYSIS

**CausalAffect vs. Existing Relations.** As shown in Figure 4, prior AU→Expression relations (FACS, cognitive, statistical, or GNN-based) rely only on hard-coded excitatory links or yield dense entangled patterns. In contrast, CausalAffect learns **polarity-aware, sparse graphs** that capture both excitatory and inhibitory pathways. For AU→AU, existing methods reduce to symmetric co-occurrence within single datasets, missing the **asymmetric interactions** observed in psychology (Rinn, 1984),whereas CausalAffect integrates non-overlapping datasets to construct asymmetric dependencies in a psychologically consistent manner (see **Appendix C** for detailed analysis).

**Psychological Plausibility.** To assess human alignment, we compare CausalAffect's discovered relations with well-established findings (Table 2). The learned graphs exhibit strong consistency with canonical psychological FACS (Ekman & Friesen, 1976), cognitive studies (Du et al., 2014), and empirical mappings (Lucey et al., 2010; Clark et al., 2020). Beyond recovering these canonical pathways, CausalAffect reveals **novel and literature-supported relations**, such as AU7→Sadness (Miller et al., 2022) and AU10→Fear (Blasberg et al., 2023), as well as **inhibitory effects** (e.g., AU6⊣Sadness, AU26⊣Sadness) highlighting the role of suppression in reliable emotion inference. These findings demonstrate that CausalAffect not only expands structural expressiveness beyond prior methods but also yields psychologically grounded causal relations (More evidence in **Appendix C.1**).

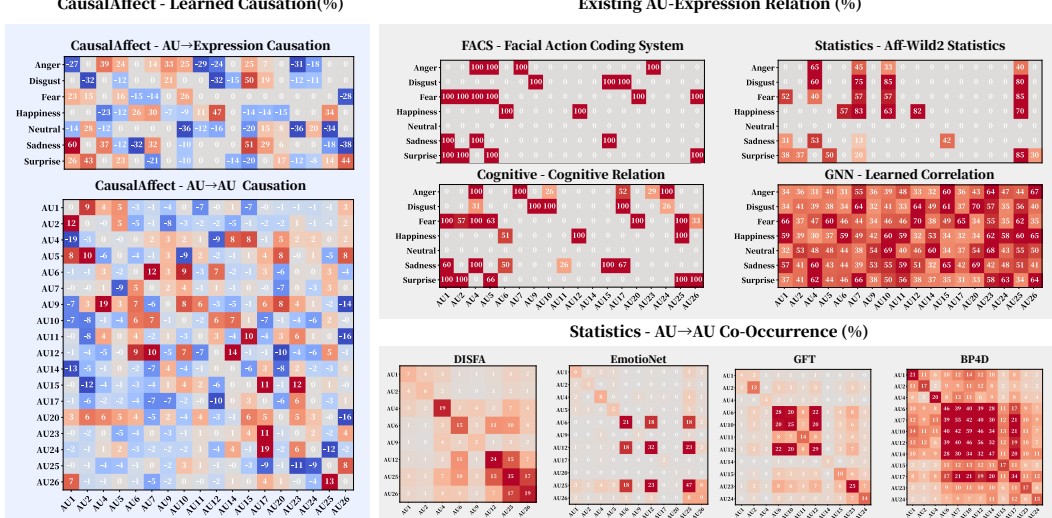

Figure 2: Comparison of CausalAffect-learned causal relations (trained on All-DB) with existing approaches. **For AU→Expression relations**, we compare against FACS (Ekman & Friesen, 1978), cognitive priors (Du et al., 2014), statistical co-occurrence from Aff-Wild2 (Kollias et al., 2024), and GNN-learned correlations (Kipf, 2016). **For AU→AU relations**, we compare co-occurrence statistics derived from four commonly used datasets.

Table 2: Comparative AU→Expression relations from prior literature and our global causal graph. **Bold** entries denote AUs reported in the literature and also identified as positive by CausalAffect.

| | (Ekman & Friesen, 1976) | (Lucey et al., 2010) | (Karthick & Jasmine, 2013) | (Du et al., 2014) | (Clark et al., 2020) | Ours (Global Causal Graph) |
|---|---|---|---|---|---|---|
| Sadness | **1, 4, 15** | **1, 4, 11, 15, 17** | **1, 4, 15, 17** | **1, 4**, 6, 11, **15, 17** | **1, 4, 15, 17** | **1, 4**, 7, **15, 17** |
| Surprise | **1, 2, 5, 26** | **1, 2, 5, 25**, 27 | **1, 2, 5, 26**, 27 | **1, 2, 5, 25, 26** | **1, 2, 5, 26**, 27 | **1, 2, 5**, 20, **25, 26** |
| Happy | **6, 12** | **6, 12**, 25 | **6, 12**, 25 | **6, 12**, 25 | **6, 12** | **6**, 7, 11, **12**, 25 |
| Disgust | **9, 15, 17** | **9**, 10, **15, 17** | **9, 17** | 4, **9**, 10, **17**, 24 | **9**, 10, **17** | **9, 15, 17** |
| Fear | **1, 2**, 4, **5**, 20, 26 | **1, 2**, 4, **5** | **1**, 4, **5**, 7 | **1, 2**, 4, **5**, 20, 25, 26 | **1, 2**, 4, **5**, 20, 25 | **1, 2, 5**, 10 |
| Anger | **4, 5, 7**, 23 | **4, 5**, 9, 10, **15, 17**, 23, 24 | **4, 5, 7**, 23, 24 | **4**, 7, **10, 17**, 23, 24 | **4, 5, 7, 10, 17**, 23, 25 | **4, 5, 7**, 9, **10, 15, 17** |

## 5.3 CASE STUDY: SAPMLE-ADAPTIVE CAUSAL RELATION ANALYSIS

**For AU→Expression**, the sample-adaptive graph (Figure 3, upper blue) demonstrates the ability to remain consistent with the global structure while *capturing instance-specific adaptations*. In *Sample 2*, the SAC-Graph highlights AU6, AU12, and AU25 as dominant contributors, closely aligning with the Global Graph. Meanwhile, it shows strong ***adaptability*** to contextual cues: in *Sample 1*, AU17 (chin raiser) is prioritized for its visual salience, whereas the Global Graph emphasizes the inactive AU4 (brow lowerer); in *Sample 3*, AU4 and AU25 are elevated for their visible activation, while globally emphasized but inactive AUs such as AU10 and AU2 are downweighted.

**For AU→AU** (Figure 3, lower yellow), *Sample 1* remains consistent with the Global Graph, where AU26 is driven by AU25 and AU5 by AU1/2/26. Beyond these canonical links, CausalAffect uncovers that many **non-root AUs** emerge not through direct excitation but through the absence of their suppressors, highlighting the role of **inhibitory dependencies**. For example, in *Sample 1* AU4 (brow lowerer) is largely inferred via suppression by AU1 (inner brow raiser), consistent with their antagonistic muscular relation in psychology (Karmann et al., 2015; Cattaneo et al., 2007). Moreover, **psychological plausibility** is evident when comparing samples with similar active AUs but divergent structures (Ekman et al., 1990; Surakka & Hietanen, 1998): *Sample 2 (Social Smile)* is dominated by inhibitory links, indicative of strained or socially modulated expressions, whereas *Sample 3 (Duchenne Smile)* exhibits a coherent feedforward graph where AUs mutually reinforce one another, consistent with spontaneous, genuine smiles. These results demonstrate CausalAffect's ability to model context-dependent dynamics, aligning with theories of expressive modulation. (More analysis in **Appendix D**)

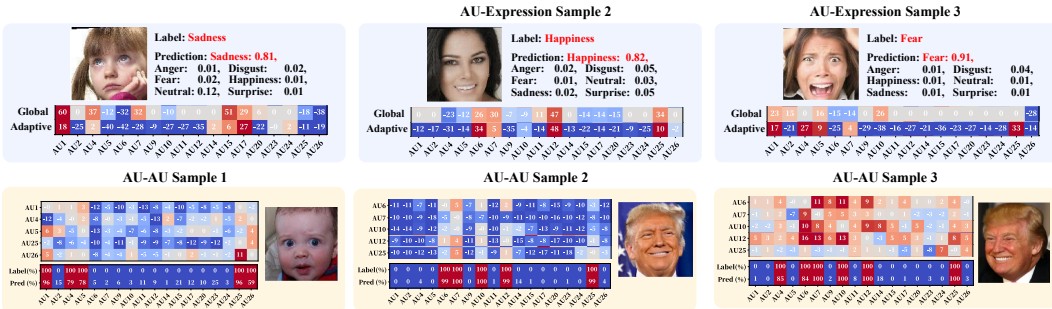

Figure 3: Sample-adaptive causal relations inferred for individual cases. Examples are shown for AU→Expression (top, blue) and AU→AU (bottom, yellow), illustrating how CausalAffect adapts to context-specific structures. For clarity, only subgraphs involving the predicted expression or active AUs are visualized.

## 5.4 ABLATION STUDY

To better understand the contributions of individual modules and their interactions, we conduct an extensive ablation study on CausalAffect (+All) using 15 model variants (Rows 1–15 in Table 4). The ablation study highlights the complementary roles of different modules. The GC captures stable population-level dependencies, while the SAC Graph introduces context-adaptive flexibility. Disentanglement provides psychologically meaningful features. Counterfactual further prunes spurious associations but only becomes effective when disentangled representations are available. Finally, the DAG constraint ensures structural clarity and interpretability. (More analysis in **Appendix A**)

| Idx | Model Variant | w/o | AffectNet | RAF-DB | DISFA | BP4D | GFT | EmotioNet |
|-----|---------------|-----|-----------|--------|-------|------|-----|-----------|
| 1 | Backbone | GC + SAC + CF | 58.9 | 70.0 | 53.0 | 57.2 | 57.9 | 47.5 |
| 2 | Backbone + Dis | GC + CF + CF | 57.1 | 69.3 | 54.2 | 55.5 | 57.8 | 59.2 |
| 3 | Backbone + GC | Dis + CF + SAC | 62.3 | 80.2 | 62.4 | 61.0 | 60.9 | 61.7 |
| 4 | Backbone + GC + Dis | CF + SAC | 61.9 | 78.2 | 61.1 | 59.8 | 60.5 | 61.4 |
| 5 | Backbone + GC + CF | Dis + SAC | 62.5 | 79.8 | 61.5 | 62.1 | 61.7 | 62.3 |
| 6 | Backbone + GC + Dis + CF | SAC | 64.4 | 83.3 | 65.8 | 66.6 | 61.0 | 63.6 |
| 7 | Backbone + SAC | Dis + CF + GC | 62.0 | 78.1 | 60.5 | 61.1 | 58.9 | 60.7 |
| 8 | Backbone + SAC + Dis | CF + GC | 60.7 | 77.9 | 60.1 | 59.5 | 57.9 | 59.2 |
| 9 | Backbone + SAC + CF | Dis + GC | 61.3 | 77.5 | 60.6 | 60.7 | 57.5 | 60.2 |
| 10 | Backbone + SAC + Dis + CF | GC | 62.7 | 78.5 | 61.7 | 62.4 | 59.1 | 61.2 |
| 11 | Backbone + GC + SAC | Dis + CF | 63.1 | 82.9 | 64.1 | 61.5 | 60.4 | 62.1 |
| 12 | Backbone + GC + SAC + Dis | CF | 62.6 | 81.4 | 66.4 | 61.3 | 60.1 | 62.0 |
| 13 | Backbone + GC + SAC + CF | Dis | 64.3 | 83.4 | 62.9 | 62.8 | **62.6** | 63.4 |
| 14 | CausalAffect (w/o DAG) | w/o DAG | 65.5 | 84.6 | 71.3 | 64.4 | 62.1 | 64.7 |
| 15 | **CausalAffect (GC + SAC + Dis + CF)** | / | **66.5** | **84.9** | **71.5** | **66.7** | 62.4 | **65.0** |

Table 3: Ablation Study on CausalAffect (+All Setting), exploring the effect of Global Causal Graph (GC), Sample-Adaptive Causal Graph (SAC), Counterfactual Intervention (CF), and AU Disentanglement (Dis).

## CONCLUSION

We presented CausalAffect, a weakly supervised framework for discovering *polarity-aware*, *directed*, and *sample-adaptive* causal relations among facial AUs and between AUs and expressions. By jointly constructing a population-level global graph and a sample-adaptive graph, and by enforcing node-level counterfactual interventions, our method recovers psychologically plausible structures that align with established cognitive studies while also uncovering inhibitory pathways and novel relations. Across six benchmarks, CausalAffect advances the state of the art in both AU detection and expression recognition, yielding human-aligned and interpretable causal graphs.

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

## A    ABLATION STUDY

To better understand the contributions of individual modules and their interactions, we conduct an extensive ablation study on CausalAffect (+All) using 15 model variants (Rows 1–15 in Table 4). The goal of this experiment is to disentangle the roles of Global Causal Graph (GC), Sample-Adaptive Causal Graph (SAC), Counterfactual (CF), AU Disentanglement (Dis), and the Directed Acyclic Graph (DAG) constraint. Performance is reported across six benchmarks (AffectNet, RAF-DB, DISFA, BP4D, GFT, EmotioNet).

| Idx | Model Variant | w/o | AffectNet | RAF-DB | DISFA | BP4D | GFT | EmotioNet |
|---|---|---|---|---|---|---|---|---|
| 1 | Backbone | GC + SAC + CF | 58.9 | 70.0 | 53.0 | 57.2 | 57.9 | 47.5 |
| 2 | Backbone + Dis | GC + SAC + CF | 57.1 | 69.3 | 54.2 | 55.5 | 57.8 | 59.2 |
| 3 | Backbone + GC | Dis + CF + SAC | 62.3 | 80.2 | 62.4 | 61.0 | 60.9 | 61.7 |
| 4 | Backbone + GC + Dis | CF + SAC | 61.9 | 78.2 | 61.1 | 59.8 | 60.5 | 61.4 |
| 5 | Backbone + GC + CF | Dis + SAC | 62.5 | 79.8 | 61.5 | 62.1 | 61.7 | 62.3 |
| 6 | Backbone + GC + Dis + CF | SAC | 64.4 | 83.3 | 65.8 | 66.6 | 61.0 | 63.6 |
| 7 | Backbone + SAC | Dis + CF + GC | 62.0 | 78.1 | 60.5 | 61.1 | 58.9 | 60.7 |
| 8 | Backbone + SAC + Dis | CF + GC | 60.7 | 77.9 | 60.1 | 59.5 | 57.9 | 59.2 |
| 9 | Backbone + SAC + CF | Dis + GC | 61.3 | 77.5 | 60.6 | 60.7 | 57.5 | 60.2 |
| 10 | Backbone + SAC + Dis + CF | GC | 62.7 | 78.5 | 61.7 | 62.4 | 59.1 | 61.2 |
| 11 | Backbone + GC + SAC | Dis + CF | 63.1 | 82.9 | 64.1 | 61.5 | 60.4 | 62.1 |
| 12 | Backbone + GC + SAC + Dis | CF | 62.6 | 81.4 | 66.4 | 61.3 | 60.1 | 62.0 |
| 13 | Backbone + GC + SAC + CF | Dis | 64.3 | 83.4 | 62.9 | 62.8 | 62.6 | 63.4 |
| 14 | CausalAffect (w/o DAG) | w/o DAG | 65.5 | 84.6 | 71.3 | 64.4 | 62.1 | 64.7 |
| 15 | **CausalAffect (GC + SAC + Dis + CF)** | / | **66.5** | **84.9** | **71.5** | **66.7** | **62.4** | **65.0** |

Table 4: Ablation Study on CausalAffect (+All Setting), exploring the effect of Global Causal Graph (GC), Sample-Adaptive Causal Graph (SAC), Counterfactual Intervention (CF), and AU Disentanglement (Dis). Best results are highlighted.

**Global vs. Sample-Adaptive Graphs:** When trained individually, the Global Graph consistently outperforms the Sample-Adaptive Graph (see Rows 3–4 vs. Rows 7–8). This outcome is expected, as GC captures stable population-level causal structures that are inherently more robust across datasets. Nevertheless, the Sample-Adaptive Graph plays an essential complementary role: when combined with GC (Rows 11-13), the performance improves further, confirming that personalized inference adds value by tailoring causal reasoning to individual instances.

**Effect of AU Disentanglement** Removing AU Disentanglement leads to noticeable performance drops (e.g., Row 1 vs. Row 11). Without disentanglement, the model may rely on spurious correlations such as demographic or identity-specific biases, which can be exploited as shortcuts. When AU Dis is present, it enhances both interpretability and robustness, guiding the model toward psychologically

meaningful AU activations. Importantly, AU Dis interacts synergistically with CF: in Rows 6, 10, and 13, CF becomes more effective when paired with disentangled features, suppressing irrelevant cues and amplifying informative dependencies.

**Role of Counterfactual Intervention** Counterfactual interventions show benefits only when supported by disentangled AU features. In the absence of AU Dis (Rows 5, 9 and 12 vs. Rows 3, 7, and 11), CF may even introduce misleading signals, as it lacks structural guidance to filter spurious correlations. However, when AU Dis is enabled, CF provides strong gains by forcing the model to contrast factual and counterfactual settings. This allows the system to emphasize influential features while suppressing misleading ones. A notable example is observed in the AU6–Disgust pathway: without CF, spurious correlations dominate, but with CF, the model prunes this dependency, yielding more interpretable causal structures.

**Human-Aligned Causal Structure** Rows 6, 10 and 13 illustrate that both CF and AU Dis are essential for human-aligned causal reasoning. Removing either component leads to reduced performance and degraded interpretability. Only when both are present can CausalAffect capture reliable, semantically grounded AU→Expression relations. This validates the psychological plausibility of our learned causal graphs, bridging low-level facial activations and high-level emotion inference.

**Effect of DAG Constraint** Finally, Row 14 highlights the necessity of the DAG constraint. Without it, the learned graph may contain redundant or semantically implausible loops due to unconstrained topology. Incorporating DAG (Row 15) introduces a soft acyclicity bias that enforces sparse, directional, and interpretable causal pathways. As a result, irrelevant connections are pruned, semantic clarity is enhanced, and overall performance reaches its peak across all benchmarks.

The ablation study highlights the complementary roles of different modules. The Global Graph captures stable population-level dependencies, while the Sample-Adaptive Graph introduces instance-level flexibility. AU Disentanglement prevents shortcut exploitation and provides psychologically meaningful features. Counterfactual regularization further prunes spurious associations but only becomes effective when disentangled representations are available. Finally, the DAG constraint ensures structural clarity and interpretability, yielding the strongest overall performance. Together, these components allow CausalAffect to achieve not only higher accuracy but also more human-aligned and semantically coherent causal graphs.

# B  SENSITIVITY TO AU COMPOSITION

To investigate how the size and composition of AU supervision affect the learned causal structure, we conducted a systematic analysis across different AU subsets from BP4D. Results are summarized in Table 5, covering both AU detection (EmotioNet, GFT) and expression recognition (RAF-DB, AffectNet).

| # | Setting / Method | EmotioNet (AU) | GFT (AU) | RAF-DB (Expr) | AffectNet (Expr) |
|---|---|---|---|---|---|
| 1 | CausalAffect (SG baseline) | 66.4 | 61.1 | – | – |
| 2 | CausalAffect (+BP4D, 6 AUs) | 65.6 | 60.3 | 80.2 | 63.7 |
| 3 | CausalAffect (+BP4D, 8 AUs, Row 2 + 2 Most Frequent AUs) | 65.8 | 61.3 | 83.5 | 65.1 |
| 4 | CausalAffect (+BP4D, 8 AUs, Row 2 + 2 Least Frequent AUs) | 64.3 | 58.6 | 82.0 | 64.2 |
| 5 | CausalAffect (+BP4D, 8 AUs, 8 Most Frequent AUs) | **66.8** | **62.7** | 84.2 | 66.3 |
| 5 | CausalAffect (+BP4D, 8 AUs, 8 Least Frequent AUs) | 63.1 | 58.3 | 81.7 | 64.5 |
| 6 | CausalAffect (+BP4D, 12 AUs) | 65.4 | 60.4 | **85.3** | **67.7** |

Table 5: Effect of AU Set Size on AU Detection and Expression Recognition. Performance is reported on EmotioNet/GFT for AU detection and RAF-DB/AffectNet for expression recognition. Best results for each task are highlighted.

**From the AU detection**, the results highlight that **frequently occurring AUs** play a dominant role in shaping robust causal dependencies. Configurations relying on the most frequent AUs (Row 5) achieve the highest AU detection performance on both EmotioNet (66.8%) and GFT (62.7%), surpassing both the single-dataset baseline and larger AU sets that include low-frequency units. In contrast, incorporating rare AUs (Rows 4 and 5, least frequent) significantly degrades performance, since their sparse activations fail to provide stable co-occurrence cues and instead inject noise into causal inference, leading to fragmented and unstable structures. Interestingly, the 12-AU configuration

(Row 6) does not outperform the best 8-AU frequent setting on AU detection, further confirming that **more AUs do not necessarily yield better causal modeling when frequency imbalance is severe**.

**For expression recognition**, a different trend emerges. Larger AU sets consistently improve performance, with the 12-AU configuration achieving the highest accuracy (85.3% on RAF-DB and 67.7% on AffectNet). This indicates that expression recognition benefits from a **richer and more compositional AU basis**, as the model learns to combine fine-grained AUs into higher-level prototypes for emotion categories. Even though low-frequency AUs hinder AU detection, they still provide complementary information that enriches expression-level inference. The gap between frequent-only subsets and the full 12-AU set demonstrates that expression recognition is more tolerant to sparsity and leverages the additional granularity to form more accurate and psychologically valid AU→Expression mappings.

Overall, these results reveal a clear sensitivity of CausalAffect to AU set size and composition: **(i)** AU detection is optimized when relying on a compact set of frequent AUs, which ensures dense and stable causal relations. **(ii)** Expression recognition, however, requires broader AU coverage, where even low-frequency units contribute to refining causal prototypes of emotions. This divergence underscores the importance of tailoring AU supervision to the specific downstream task—favoring *frequency and stability* for AU detection, while emphasizing *richness and compositionality* for expression recognition.

## C  GLOBAL CAUSAL RELATION ANALYSIS

CausalAffect constructs global causal graphs over both AU→Expression and AU→AU spaces, revealing human-aligned interpretable, directional, and semantically grounded dependencies. It captures not only canonical facial expression cues and co-activation patterns but also inhibitory and statistically inaccessible relations—offering structural priors that go beyond statistics approach.

### C.1  AU-EXPRESSION GLOBAL CAUSAL RELATION

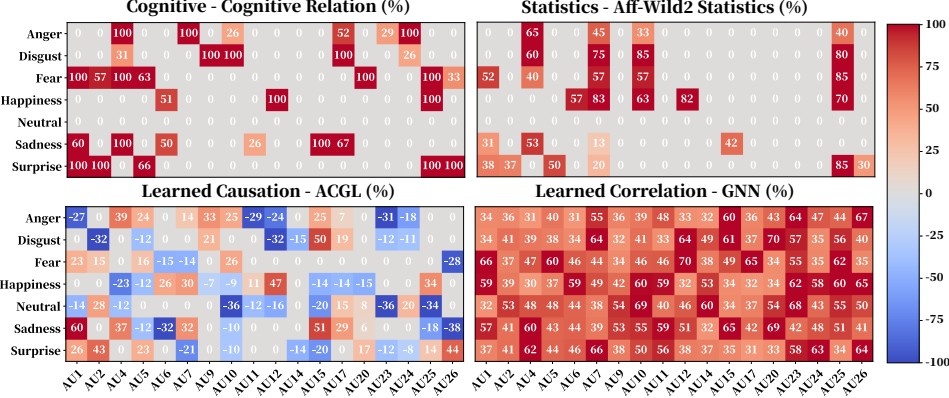

Figure 4: Comparison of AU-Expr Relations across cognitive priors, statistical co-occurrence, GNN-learned correlation(Learned on All-DB), and CausalAffect-learned causation (Learned on All-DB) (%).

**Alignment with human priors and expression semantics.** The AU→Expression causal graph (Figure 4) learned by **CausalAffect** reveals both **causally grounded** and **psychologically plausible** structures. The discovered relations align closely with established findings in facial behavior research(Ekman & Friesen, 1978), as well as with prior **statistical co-occurrence relations**(Kollias et al., 2024) and **cognitive models**(Du et al., 2014). Notably, CausalAffect recovers several canonical expression markers, including AU12 → Happiness, AU1,AU4,AU15 → Sadness, and AU2,AU26 → Surprise, all of which are supported by both *cognitive neuroscience evidence* and empirical patterns observed in datasets such as *Aff-Wild2*. These relations are consistent with well-understood affective mechanisms: AU12 (lip corner puller) is the primary indicator of enjoyment-related expressions such as happiness or amusement; AU1, AU4, and AU15 (inner brow raiser, brow

lowerer, and lip corner depressor) are prototypical components of Sadness, reflecting upper-face tension, concern, and downward mouth pull associated with grief or emotional pain; AU2 and AU26 (outer brow raiser and jaw drop) are hallmark components of Surprise, reflecting widened eyes and involuntary jaw relaxation respectively—together forming a classic upper- and lower-face response to sudden or unexpected stimuli. Importantly, these dependencies are learned without any AU–expression co-annotation. This highlights CausalAffect's ability to infer semantically aligned and interpretable structures in a fully data-driven manner—effectively bridging low-level facial actions and high-level affective understanding.

**Modeling inhibitory causal relations.** Beyond capturing canonical positive dependencies, CausalAffect also discovers *inhibitory causal relations*—directed negative influences that are largely absent from existing statistical or cognitive structures. These relations do not merely indicate suppression or co-inhibition, but rather reflect a form of **inhibitory precondition**: the absence of a particular AU becomes a *causal prerequisite* for a given expression to be inferred. For example, AU6 ⊣ Sadness and AU26 ⊣ Sadness indicate that the non-activation of smile-related or surprise-related AUs is a necessary condition for confidently inferring Sadness. AU6 (cheek raiser) is a hallmark of joy and amusement, while AU26 (jaw drop) is prominently associated with Surprise. Their presence would contradict the subdued, upper-face tension and downward lip dynamics that define Sadness, including AUs such as AU1, AU4, and AU15. These inhibitory causal links reflect the principle of inhibitory precondition: Sadness becomes a plausible interpretation not only due to the presence of its prototypical AUs, but also because **affectively incompatible** actions like AU6 and AU26 are absent. Such negative causal links highlight the model's ability to reason not only about what *must be present*, but also about what *must be absent* for an emotion to be plausible. This aligns with psychological theories of emotional exclusivity and supports more precise disambiguation in overlapping facial configurations. Overall, these inhibitory relations allow CausalAffect to move beyond symmetric co-activation and toward truly directional understanding of facial expressions.

**Causal graph as a diagnostic prior for label auditing.** Beyond modeling facial behavior, the learned causal graph also demonstrates strong **diagnostic utility**. While **Neutral** is conventionally assumed to co-occur with the absence of active AUs in psychology, CausalAffect uncovers consistent positive causal links from AU24 (lip pressor), AU2 (outer brow raiser), and AU17 (chin raiser) to Neutral. These findings **challenge conventional assumptions**, suggesting that many Neutral-labeled samples actually exhibit subtle but structured AU activations. Given the known difficulty in annotating low-intensity or ambiguous AUs, such patterns likely reflect **systematic label noise** rather than genuine neutrality. In this context, the causal graph functions as a **structural prior** that can support **label auditing**, **confidence calibration**, and improved **annotations robustness**. By identifying unexpected or semantically inconsistent activations within annotated Neutral instances, CausalAffect provides a principled mechanism for evaluating annotation quality and guiding data refinement.

## C.2 AU-AU GLOBAL CAUSAL RELATION

To our best knowledge, this work presents *the first* data-driven framework to learn human-aligned AU→AU causal dependencies from weakly labeled data. Unlike AU→Expression mappings, which have been studied in psychology and affective computing, there exists no established ground truth or cognitive theory that defines directed causal relations between AUs themselves. As a result, we evaluate the plausibility of our learned causal graph by comparing it with AU co-occurrence statistics derived from four widely-used facial expression datasets: GFT, DISFA, EmotioNet, and BP4D.

While co-occurrence statistics provide a simple way to analyze AU correlations, they suffer from several inherent limitations: **(i) Incomplete relational coverage:** Co-occurrence statistics cannot establish pairwise relations across all 18 AUs due to the lack of overlapping AU annotations among the datasets. In contrast, our CausalAffect framework learns a unified causal graph that covers the complete set of 18 AUs (see left of Figure. 5), without relying on dataset-specific label availability. **(ii) Symmetry assumption:** Co-occurrence measures are inherently symmetric by definition, i.e., $P(AU_i|AU_j) = P(AU_j|AU_i)$, which fails to capture the directional nature of inter-AU influences. In contrast, our learned causal graph reveals asymmetric causal dependencies that reflect realistic directional interactions between AU pairs. **(iii) Lack of inhibitory relations:** Co-occurrence methods can only indicate excitatory association patterns. They are unable to model inhibitory or suppressive relationships. In contrast, CausalAffect captures both excitatory and inhibitory causal effects, enabling finer-grained interpretation of inter-AU dynamics.

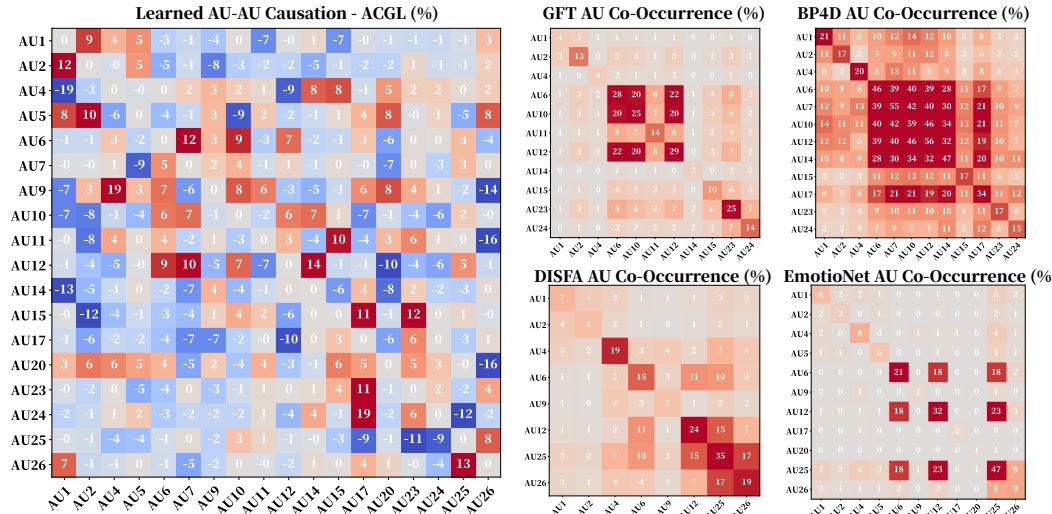

Figure 5: Comparison between the AU-AU causal relations learned by CausalAffect (trained with the +ALL setting) and AU co-occurrence statistics from four datasets (GFT, DISFA, EmotioNet, BP4D).

Compared to the AU→Expression causal links (Figure 4), which exhibit strong influence patterns, the AU→AU relations tend to have lower overall magnitudes, with the maximum absolute value reaching approximately 19. This observation is consistent with the underlying nature of facial behavior: AU-AU interactions primarily capture low-level muscular coordination or antagonism (e.g., AU6 and AU7 co-activating around the orbital region), rather than reflecting high-level semantic or emotional constructs. As a result, these causal effects are more distributed and less dominant than the structured compositional dependencies observed in AU→Expression modeling.

**Alignment with human priors and co-occurrence trends.** Despite being trained without explicit supervision from domain knowledges or handcraft prior, CausalAffect successfully recovers human-aligned AU-AU causal patterns that are consistent with known trends in facial actions. For instance, CausalAffect learns strong positive causal relations such as AU7 → AU12 and AU6 → AU12, which correspond to canonical activation pathways in genuine (Duchenne) smiles. These relations are also prominent in dataset-level statistics—e.g., GFT AU6 → AU12 co-occurrence at +22%, while BP4D shows even stronger associations for AU6 → AU12 and AU7 → AU12. Similarly, the model learns AU25 → AU26, capturing the natural progression from lip parting to jaw drop, which aligns with co-occurrence strengths observed in DISFA. Another example is AU10 → AU12, often seen in expressions of contempt or disgust, which is mirrored by co-occurrence patterns in GFT and BP4D.

**Capturing inhibitory causal relations.** One of the most distinctive advantages of CausalAffect over prior approaches is its ability to model *inhibitory causal relations* between AUs—i.e., directed negative influences that reflect mutual exclusivity or muscular suppression. This capacity is largely missing from existing dependency modeling methods, which typically rely on symmetric or co-occurrence-based statistics and thus fail to capture negative interactions. Such inhibitory relations are especially important in AU detection, where many AUs are known to be semantically incompatible. For example, CausalAffect learns a strong negative influence AU4 ⊣ AU1 , reflecting the **physiological antagonism** between brow lowering (corrugator supercilii) and inner brow raising (frontalis). Similarly, AU26 ⊣ AU20 captures the incompatibility between horizontal lip stretching and vertical jaw dropping. The relation AU26 ⊣ AU11 also illustrates this, as AU11 contributes to nasolabial deepening during expressions of effort or sneering, which typically opposes the open-jaw posture characterized by AU26. In addition, we observe semantically suppressive relationships in the upper and lower face. For instance, AU1 ⊣ AU14 reflects the tension between dimple-induced controlled smiles and inner brow raising, which signal conflicting emotional states such as restrained positivity versus concern. The relation AU2 ⊣ AU15 highlights the opposition between lip corner depression (sadness) and outer brow raising (surprise), while AU20 ⊣ AU12 encodes the mismatch between smiling and horizontal lip tension typically associated with fear or anxiety.

**Uncovering semantically important but statistically inaccessible relations.** Beyond aligning with known co-occurrence trends, CausalAffect also discovers several high-impact causal relations that

are statistically inaccessible in existing datasets due to non-overlapping AU annotations. For example, the learned relation AU15 → AU20 captures the dynamic transition from lip corner depression (sadness) to horizontal lip stretch (tension or discomfort), which is rarely annotated together in existing datasets but plays an important role in modeling affective states. In addition, CausalAffect captures several causal dependencies that are **missing from co-occurrence statistics** entirely, yet are highly meaningful for facial interpretation. For instance, AU4 → AU9 indicates a strong link between brow lowering (anger/focus) and nose wrinkling (disgust), frequently observed in complex expressions such as contempt or intense concentration. The relation AU15 → AU11 reflects a plausible lower-face interaction where lip corner depression activates muscular pathways contributing to nasolabial fold deepening. Similarly, AU17 → AU24 links chin raising with lip pressing—both associated with suppressive, high-tension affective states such as fear or frustration. These examples highlight CausalAffect's ability to infer biologically and semantically grounded causal interactions beyond what is available in co-occurrence statistics, offering richer structural priors that are critical for robust and generalizable AU-based facial analysis.

**Effect of Directed Acyclic Graph (DAG) Constraint:** To guide the model toward learning interpretable structures, we incorporate a Directed Acyclic Graph (DAG) constraint during training. This constraint serves to prioritize *directed, asymmetric, and semantically meaningful* causal relationships over symmetric statistical associations. In particular, it helps suppress noisy bidirectional correlations and encourages the model to resolve causal directionality. Interestingly, we observe that the learned causal graph still contains localized cycles—for example, AU1 → AU2 with a weight of +12 and AU2 → AU1 with +9. While this appears to violate the DAG constraint, it reflects an important characteristic of facial dynamics. In psychological and behavioral literature(Ekman & Friesen, 1978), many AU pairs are known to exhibit *symbiotic or reciprocal* relationships. AUs such as AU1 and AU2 frequently co-activate in expressions like surprise or concern. Therefore, our implementation adopts the DAG constraint as a *soft regularization* rather than a hard constraint. This design allows the model to retain the flexibility needed to capture biologically plausible reciprocity in AU behavior, while still being biased toward uncovering dominant and interpretable directional dependencies.

# D  CASE STUDY: SAMPLE-ADAPTIVE CAUSAL RELATION ANALYSIS

In this section, we present examples of both AU→Expression and AU→AU sample-adaptive causal graphs to demonstrate how **CausalAffect** dynamically constructs instance-specific causal structures. These case studies reveal how the model adapts its reasoning to each individual input, capturing both prototypical and idiosyncratic facial dynamics beyond what is reflected in global statistical trends.

## D.1  AU-EXPRESSION SAMPLE-ADAPTIVE CAUSAL RELATION

**Sample-adaptive graph aligns with global priors while preserving expression-specific consistency.** Through systematic case-by-case analysis across six primary emotions and neutral states, We observe that **Sample-adaptive graph** consistently recovers AU→Expression structures that align well with global affective patterns. For example, in **Sample 2 (Happiness)**, Sample-adaptive graph identifies AU12 and AU6 as dominant contributors—closely matching the global graph. Similarly, **Sample 4 (Surprise)** features AU25, AU26, and AU5, which are also prominent in the global graph: AU25, AU26, and AU5. In both cases, Sample-adaptive graph additionally suppresses conflicting AUs such as AU4 (in Sample 2) or AU23 (in Sample 4), maintaining semantic consistency with the global graph.

While Sample-adaptive graph accurately replicates global causal trends, it also demonstrates strong adaptability in tailoring inference to the sample-specific AU configuration. For example, in **Sample 3 (Fear)**, sample-adaptive graph prioritizes AU25, AU4, and AU1, which are visually salient in the image, but globally less emphasized compared to AU10 and AU5—both of which are inactive and thus downweighted. A similar pattern appears in **Sample 6 (Anger)**, where Sample-adaptive graph focuses nearly all attribution on AU4, whereas the global graph distributes importance across AU10, AU9, and AU15—none of which are visibly active in the instance. In **Sample 5 (Disgust)**, the sample-adaptive causal graph highlights AU10, AU7, and AU4 as the primary causal drivers—reflecting the visible upper-face tension and mid-face wrinkling characteristic of disgust. Meanwhile, AU6 and AU12 are strongly suppressed. This contrasts with the global AU→Disgust graph, which emphasizes AU15 as the dominant lower-face contributor, along with inhibitory weights on AU12 and AU2. The

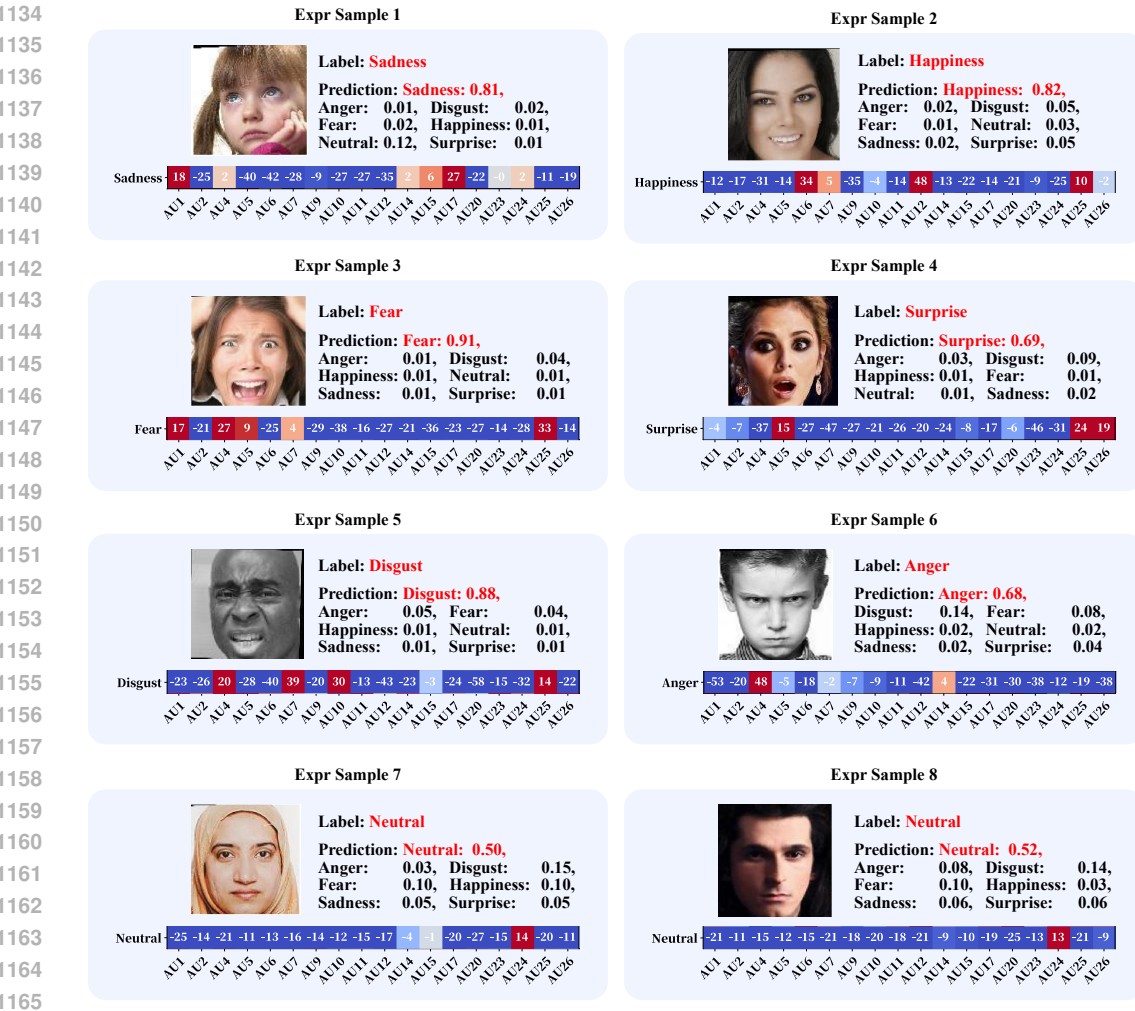

Figure 6: Sample-adaptive AU→Expression causal relations dynamically inferred for individual samples. We random sample one image per basic expression (and two for Neutral) to illustrate how CausalAffect captures instance-specific causal structures. For clarity, only the subgraph corresponding to the predicted expression is visualized.

discrepancy illustrates how CausalAffect dynamically adjusts its causal reasoning based on observed facial features, prioritizing context-relevant AUs.

**Instance-level causal graphs enable principled diagnosis of label noise.** A key strength of **CausalAffect** lies in its ability to detect annotation inconsistencies through fine-grained, instance-specific causal reasoning. This capability becomes particularly evident in **Samples 7 and 8 (Neutral)**, both of which display visually static, expressionless faces with minimal muscular activity. Despite the Neutral label, CausalAffect assigns weak positive causal weights to AU24 (lip pressor: $+14$, $+13$), while suppressing all other AUs—including AU2 (outer brow raiser, $-35$) and AU17 (chin raiser, $-29$). This discrepancy highlights a critical semantic inconsistency: *by definition*, Neutral should not be causally supported by any strong AU activation. Even the attribution of AU24 as a weak positive contributor appears questionable, particularly given the lack of visible lip pressing in the corresponding images. As a low-saliency AU prone to misinterpretation—often confounded with pre-speech tension or relaxed mouth posture—AU24 is highly susceptible to annotation noise. The fact that CausalAffect assigns marginal support to AU24, while confidently suppressing all other AUs, suggests that such annotations may reflect **systematic over-labeling** rather than true muscular expression. In this sense, the sample-adaptive causal graph provides a valuable structural prior for **label auditing, quality assessment, and data refinement**, enabling the model not only to learn from data but to question its reliability.

## D.2 AU-AU Sample-Adaptive Causal Relation

Unlike dataset-level co-occurrence, CausalAffect learns fine-grained, sample-specific AU→AU causal graphs. For instance, in **Sample 1**, AU1 is causally inferred from AU2, AU5, AU26, and AU25, all of which are visibly present in the image. This structure partially aligns with the global AU→AU graph (e.g., AU5 → AU1, AU26 → AU1), yet reflects a clearer subject-specific adjustment. Similarly, AU25 → AU26 recovers canonical part-to-drop progression, demonstrating consistency with global priors.

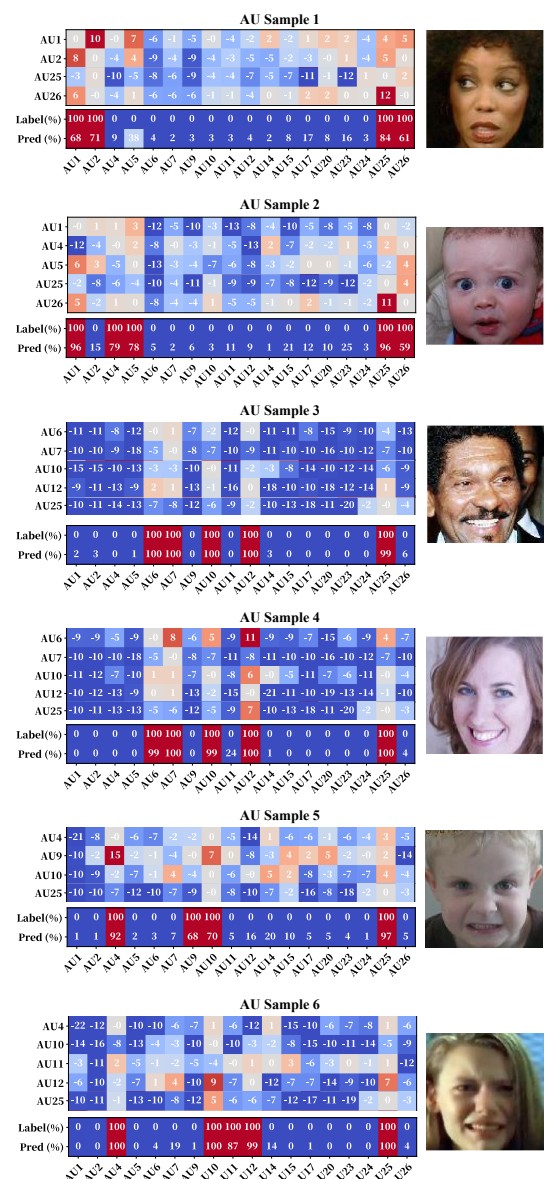

However, in this same example, AU25 is not inferred through direct excitatory causes but via exclusion-based reasoning: it receives strong *negative* causal input from AU4, AU17, and AU15, suggesting that the absence of these mutually exclusive AUs makes AU25 the most plausible outcome. A similar dynamic is observed in **Sample 5**, where AU25 is again inferred not from direct support but from the strong inhibition of AU20—a horizontal lip stretcher biomechanically incompatible with vertical lip separation. These examples illustrate CausalAffect's ability to infer causal dependencies not only through direct excitation but also through *causal exclusion*, wherein the deactivation of specific AUs increases the inferred necessity of others. Additionally, Sample 5 shows that AU9 (nose wrinkler) is causally activated by AU4 and AU10, yet AU4 is itself inhibited by AU10, suggesting muscular antagonism in the upper face. This internal tension reflects CausalAffect's capacity to capture not just co-activation, but also expressive conflict within fine-grained causal paths.

**Causal inference of non-root AU activations via inhibitory dependencies.** A central strength of CausalAffect is its ability to model **non-root AUs**—those that do not originate from direct causal initiators but instead emerge as residual consequences of **the absence of suppressive or competing AUs**. This is clearly demonstrated in **Sample 3**, where all predicted AUs (e.g., AU6, AU7, AU12, AU25) are inferred primarily via negative causal inputs. For example, AU6 is suppressed by AU20, AU5, AU1 and AU2, while AU12 is inferred through the absence of AU14 and AU20. These patterns suggest that the facial configuration reflects controlled or ambiguous expression states (e.g., social smiling or emotional masking), where expressive AUs are not initiated directly but emerge under mutual inhibition constraints.

This form of inference is primarily enabled by **soft DAG constraint**, which promotes sparse and directional structures. By discouraging indiscriminate bidirectional correlations, the DAG bias guides the model to identify minimal and interpretable causal pathways—including those where an AU's activation is inferred not through direct excitation, but via the causal absence of its antagonistic counterparts. When dominant upstream

Figure 7: Sample-adaptive AU→AU causal relations dynamically inferred for individual samples. We randomly select 6 images from the EmotioNet to illustrate how CausalAffect captures instance-specific inter-AU causal structures. For clarity, only the subgraphs involving the predicted activated AUs are visualized.

AUs are absent—as in **Sample 2**, where AU2 is inactive—the model adaptively reweighs negative contributors to AU1, incorporating AU6, AU11, AU9, and AU15 to compensate for the lack of canonical support. Similarly, in **Sample 6**—a sadness-pain expression—AU12 is inferred not from global graph drivers like AU6, but through modest support from AU10, AU25, and AU7, suggesting indirect lower-face coordination. In contrast, AU4—despite receiving strong inhibition from AU1, AU15, and others—remains active, reflecting emotionally driven override of suppressive muscular input. Together, these observations demonstrate that CausalAffect goes beyond surface-level co-activation and captures latent causal architecture shaped by antagonism, compensation, and expressive tension—key factors in modeling natural, psychologically grounded facial dynamics.

**Psychological interpretation of AU-AU causal differences across expressions.** A comparison of samples with similar active AUs but different causal structures further illustrates CausalAffect's nuanced understanding of expressive dynamics. **Samples 3 and 4** both contain AU6, AU7, AU10, AU12, and AU25, yet their AU→AU graphs diverge sharply. In **Sample 3**, these AUs are inferred primarily via inhibitory input—e.g., AU7 is negatively influenced by AU5, AU20, and AU24, and AU12 is suppressed by AU14, AU11, and AU20. This structure suggests that the system interprets these AUs as **emergent, not volitional—consistent with strained, socially modulated, or ambiguous affect (Surakka & Hietanen, 1998).** In contrast, **Sample 4** exhibits a coherent and positively coordinated causal graph: AU6 is supported by AU12, AU7, AU10, and AU25, and AU25 is driven by AU12. This feedforward configuration reflects a spontaneous and emotionally consistent smile, where lower-face AUs reinforce each other. These differences align with psychological theories of expressive modulation—such as **Duchenne vs. non-Duchenne smiles, emotional masking, or communicative gestures**—highlighting that the same AUs can play different causal roles depending on the expressive context.

## E  HSIC COMPUTATION.

We use the empirical HSIC estimator (Gretton et al., 2005) to compute the dependence between any pair of random variables $X$ and $Y$ based on their batch-wise representations. Given $n$ samples $\{(x_k, y_k)\}_{k=1}^n$, we first compute the kernel matrices $K$ and $L$ using RBF (Gaussian) kernels:

$$K_{ij} = \exp\left(-\frac{\|x_i - x_j\|^2}{2\sigma_x^2}\right), \quad L_{ij} = \exp\left(-\frac{\|y_i - y_j\|^2}{2\sigma_y^2}\right), \tag{17}$$

where $\sigma_x$ and $\sigma_y$ are bandwidth parameters (either fixed or estimated via median heuristic). The empirical HSIC is then computed as:

$$\mathcal{H}(X, Y) = \frac{1}{(n-1)^2}\text{tr}(KHLH), \tag{18}$$

where $H = I - \frac{1}{n}\mathbf{1}\mathbf{1}^\top$ is the centering matrix that removes the mean from kernel features. This formulation enables unbiased estimation of the squared Hilbert-Schmidt norm of the cross-covariance operator between $X$ and $Y$ in their respective RKHSs. In our implementation, we apply the above formulation to all three losses $\mathcal{L}_{\text{ib}}$, $\mathcal{L}_{\text{align}}$, and $\mathcal{L}_{\text{decorr}}$ using mini-batch representations of AU heads, the global image embedding $z_{\text{img}}$, and corresponding (pseudo-)labels. This HSIC-based framework offers a unified and theoretically grounded approach to disentanglement, without relying on adversarial learning or variational estimation.

## F  EXPERIMENTS

### F.1  DATASETS

We evaluate CausalAffect on six widely-used facial analysis datasets: BP4D, DISFA, EmotioNet, GFT, RAF-DB, and AffectNet. Among them, BP4D, DISFA and GFT provide frame-level annotations of facial AUs, while EmotioNet, RAF-DB and AffectNet offer image-level labels.

**AU datasets.** We follow (Shao et al., 2021b; Kollias et al., 2024) for consistent AU dataset splitting and evaluation. *BP4D*(Zhang et al., 2014) contains 41 subjects with spontaneous expressions annotated over 12 AUs. *DISFA* (Mavadati et al., 2013) includes posed and spontaneous videos with 8 selected AUs. *EmotioNet* (Fabian Benitez-Quiroz et al., 2016) consists of over 45K in-the-wild facial images, where we follow the official split and use the 11 most frequent AUs for training and

evaluation. *GFT* (Girard et al., 2017) is a proprietary dataset containing over 130K high-quality face images annotated with 10 AUs. For all AU datasets, we binarize labels using intensity thresholds if available, and compute AU-wise and average F1 scores across the selected AU subsets in each dataset.

**Expression datasets.** *RAF-DB* (Li et al., 2017) contains around 15K images with crowd-sourced annotations of 7 basic expressions. *AffectNet* (Mollahosseini et al., 2017) provides over 250K manually labeled facial images, from which we use the standard 7-class subset (Neutral + 6 basic emotions) for evaluation. We adopt the official train/validation splits provided by both datasets.

## F.2 IMPLEMENTATION DETAILS

The backbone is ConvNeXt-Base pretrained on WebFace(Yi et al., 2014). Input images are resized to $112 \times 112$ and globally pooled to obtain $z_{\text{img}} \in \mathbb{R}^{1024}$. AU-specific features $\mathbf{f}_{\text{AU}}^{(i)} \in \mathbb{R}^{64}$ are extracted. HSIC-based disentanglement is applied using RBF kernels with median bandwidth, optimizing $\mathcal{L}_{\text{ib}}$, $\mathcal{L}_{\text{align}}$, and $\mathcal{L}_{\text{decorr}}$ with weights 1.0, 1.0, and 0.2 respectively. We initialize global edge logits in $[-0.01, 0.01]$, use signed 8-head attention, and apply a soft DAG loss with $\lambda_{\text{DAG}} = 0.5$. Sample-adaptive graphs are computed via a 2-layer attention. AU→Expr graphs use learned expression prototypes(dim 64) as attention queries. Counterfactual intervention uses $\epsilon \sim \mathcal{N}(0, 0.2^2)$ with gate sharpness $\gamma = 10$ and loss weights $\lambda_{\text{consist}} = \lambda_{\text{discrep}} = 0.5$, $\delta_{\text{feat}} = \delta_{\text{logit}} = \eta_{\text{feat}} = \eta_{\text{logit}} = 1.0$. We train for 80 epochs using AdamW (lr=1e−4, wd=1e−5), batch size 360

## F.3 SENSITIVITY ANALYSIS OF REGULARIZATION PARAMETERS

In addition to evaluating the overall effectiveness of CausalAffect, we further investigate the sensitivity of its key regularization parameters, focusing on the mask sharpness parameter $\gamma$ and the feature/logit-level causal regularization weights. These parameters are not extensively tuned for each dataset but instead are guided by theoretical considerations to ensure robustness and interpretability across heterogeneous settings.

**Fixed Robustness Across Settings** A key design choice of CausalAffect is to maintain a fixed set of hyperparameters across all datasets and experiments, including multi-dataset training and ablation studies. This consistent configuration ensures robustness and transferability, demonstrating that the model architecture and loss formulation are inherently stable. As reported in main paper Table 1 and Ablation Study Table 4, CausalAffect achieves strong performance without dataset-specific tuning, highlighting its reproducibility and practical usability in real-world scenarios.

| CausalAffect (RAF-DB + BP4D) | RAF-DB | BP4D |
|---|---|---|
| $\gamma = 20$ | 84.1 | 66.7 |
| $\gamma = 10$ | **85.0** | **67.1** |
| $\gamma = 5$ | 84.3 | 66.2 |
| $\delta_{\text{feat}} = \delta_{\text{logit}} = \eta_{\text{feat}} = \eta_{\text{logit}} = 1$ | **85.0** | **67.1** |
| $\eta_{\text{feat}} = \delta_{\text{feat}} = 0.5, \eta_{\text{logit}} = \delta_{\text{logit}} = 1$ | 83.2 | 66.3 |
| $\eta_{\text{feat}} = \delta_{\text{feat}} = 1, \eta_{\text{logit}} = \delta_{\text{logit}} = 0.5$ | 84.5 | 66.6 |
| $\eta_{\text{feat}} = \delta_{\text{feat}} = \eta_{\text{logit}} = \delta_{\text{logit}} = 0.5$ | 83.0 | 66.1 |

Table 6: Sensitivity analysis of mask sharpness ($\gamma$) and causal regularization weights in CausalAffect on RAF-DB and BP4D datasets. Best results are highlighted.

**Effect of Mask Sharpness** $\gamma$ The parameter $\gamma$ controls the steepness of the sigmoid gating mask for counterfactual intervention, thereby determining the sharpness of feature selection. Setting $\gamma = 10$ achieves the best balance, producing a binary-like mask that reliably activates strong edges above the 0.5 threshold while suppressing weak connections. Increasing $\gamma$ to 20 overly sharpens the mask, reducing generalization and hindering causal learning, while decreasing $\gamma$ to 5 produces overly soft masks that blur causal boundaries. As shown in Table 6, $\gamma = 10$ consistently yields the strongest results on both RAF-DB and BP4D datasets.

**Effect of Feature and Logit Regularization** We also examine the relative importance of feature-level ($\delta_{\text{feat}}, \eta_{\text{feat}}$) and logit-level ($\delta_{\text{logit}}, \eta_{\text{logit}}$) causal regularization weights. Equal weighting (all set to 1.0) achieves the best trade-off, balancing causal separability at the feature level and stability at the prediction level. Reducing feature-level weights while keeping logit-level weights high leads to the largest performance degradation, indicating that feature-level constraints are more critical for reliable causal inference. Conversely, reducing logit-level weights produces a smaller drop in performance,

but still highlights their relevance. Setting all weights to 0.5 yields the lowest results, confirming that strong and balanced regularization is essential.

Overall, these results highlight that CausalAffect is not overly sensitive to fine-grained hyperparameter tuning. Instead, it benefits from theoretically motivated regularization choices that ensure stability, interpretability, and transferability across datasets. The optimal configuration corresponds to $\gamma = 10$ and balanced regularization weights at 1.0, which together maximize AU and expression recognition performance while maintaining psychologically interpretable causal structures.

## G  TRAINING AND INFERENCE EFFICIENCY

We report representative training and inference experiments to assess the computational efficiency of CausalAffect under different scales of training data. All experiments were conducted on a single NVIDIA A100 GPU with an Intel(R) Xeon(R) Gold 6142 CPU (2.60 GHz), running Rocky Linux. We adopt ConvNeXt-Base as the backbone with an input resolution of $112 \times 112$.

| Experiment | +All | AffectNet+BP4D | DISFA+RAF-DB | EmotioNet |
|---|---|---|---|---|
| Total Training Images | ∼800K | ∼257K | ∼100K | ∼24K |
| Batch Size | 360 | 120 | 120 | 60 |
| Training Epochs | 80 | 43 | 21 | 39 |
| Time per Epoch | ∼260s | ∼108s | ∼98s | ∼43s |
| Total Training Time | ∼5.8h | ∼1.29h | ∼0.57h | ∼0.46h |
| Peak GPU Memory Usage | 28–30 GB | 18–20 GB | 18–19 GB | 10–12 GB |
| Inference Speed | ∼200 FPS | ∼250 FPS | ∼250 FPS | ∼270 FPS |

Table 7: Training and inference efficiency of CausalAffect across different experimental configurations.

**Large-Scale Configuration:** The largest-scale training setting corresponds to **CausalAffect (+All)**, which involves approximately 800K images from six datasets. A batch size of 360 (60 images per dataset) was used. Training for 80 epochs required around 260 seconds per epoch, resulting in a total training time of ∼5.8 hours. The peak GPU memory usage was between 28–30 GB, while the inference speed reached ∼200 FPS. This demonstrates that CausalAffect remains computationally tractable even at large scale, making it feasible for deployment on high-performance GPUs.

**Medium- and Small-Scale Configurations:** For smaller-scale experiments, training time and memory requirements were substantially reduced: **AffectNet+BP4D** (∼257K images): batch size 120, training ∼1.29 hours, memory usage 18–20 GB, inference speed ∼250 FPS. **DISFA+RAF-DB** (∼100K images): batch size 120, training ∼0.57 hours, memory usage 18–19 GB, inference speed ∼250 FPS. **EmotioNet** (∼24K images): batch size 60, training ∼0.46 hours, memory usage 10–12 GB, inference speed ∼270 FPS.

These results highlight that CausalAffect scales gracefully with dataset size: larger datasets increase training time and GPU memory usage but inference remains efficient, consistently above 200 FPS across all settings.

## H  RELATED WORK

### H.1  MODELING STRUCTURED RELATIONS BETWEEN AUs AND EXPRESSIONS

Learning the structured relationships among Action Units (AUs) and between AUs and expressions is crucial for improving facial affect recognition. Psychological and cognitive science studies consistently show that facial expressions emerge through coordinated AU activations (Ekman & Friesen, 1978; Ekman, 2002; Jack et al., 2014), and AU–AU as well as AU–expression dependencies provide strong priors for enhancing both AU detection and expression recognition (Du et al., 2014; Tian et al., 2001; Jack et al., 2012; Luo et al., 2022). A model that accurately captures such dependencies is better positioned to reason about facial behavior, reduce ambiguity among visually

similar patterns (Dhall et al., 2015; Kolahdouzi et al., 2021), and provide explanations aligned with human expert interpretation.

**Expert-Curated Relations:** The Facial Action Coding System (FACS) (Ekman & Friesen, 1978) provides the foundation for describing these relations, and many early approaches incorporate cognitive or psychological priors. Prior-based methods rely on manually defined AU compositions or expert-curated AU groups (Ekman & Friesen, 1978; Du et al., 2014), ensuring a degree of interpretability by enforcing human-understandable structures. However, such approaches assume fixed population-level mappings and cannot adapt to **individual differences** (e.g., idiosyncratic AU combinations or varying AU intensities) as demonstrated in psychological studies of AU variability (Ellsworth & Scherer, 2003). Furthermore, hand-crafted rules cannot represent **inhibitory** effects or **asymmetric** AU relations frequently observed in natural facial behavior (Sackeim & Gur, 1980; Jack et al., 2012).

**Data-Driven Modeling Relations:** To address the limitations of fixed psychological priors, recent work employs graph neural networks and attention-based architectures to *learn* AU–AU and AU–expression relations directly from data (Song et al., 2021b; Wang et al., 2023b; Liu et al., 2020; Tian et al., 2001; Yang et al., 2019). These approaches introduce powerful relational inductive biases and often yield substantial gains in recognition performance. However, because they optimize primarily for discriminative accuracy, the resulting relation structures usually take the form of undirected affinity graphs rather than psychologically grounded causal mechanisms. As a consequence, the learned graphs tend to be **dense**, highly **entangled**, and **difficult to interpret** (Kakkad et al., 2023; Wang et al., 2023a), lacking the directionality, asymmetry, and polarity-aware dependencies that human experts routinely rely on when decoding facial behavior.

**Data-Driven Co-occurrence Relations:** Another class of works derives AU or expression dependencies from statistical co-occurrence in jointly annotated datasets (Eleyan & Demirel, 2009; Kollias & Zafeiriou, 2021; Kollias et al., 2021). While co-occurrence patterns provide useful weak structural cues, they are known to encode dataset-specific biases, demographic imbalances, and annotation noise (Chen & Joo, 2021; Dominguez-Catena et al., 2024). Moreover, because these methods require AU and expression labels to appear on the same images, they cannot take advantage of the many AU-only or expression-only datasets widely used in practice (Yang et al., 2019; Kollias et al., 2021). Finally, frequency-based relations reflect **shared occurrence rates** rather than underlying **excitatory or inhibitory** mechanisms, making them misaligned with human-understandable causal reasoning.

Overall, these existing approaches struggle to provide **human-aligned interpretability**: they cannot explain *which* AU activations lead to a particular expression prediction, *how* AUs influence one another, or *why* a model arrives at certain decisions. This motivates the need for a causal modeling framework that behaves more like a trained expert— one that learns directed, asymmetric, and polarity-aware relations, generalizes across heterogeneous datasets, and offers coherent, human-aligned explanations for model predictions.

### H.2 CAUSAL AND COUNTERFACTUAL MODELING IN FACIAL BEHAVIOR

Causal reasoning and counterfactual interventions have recently been explored in computer vision, often in the context of generative modeling for fairness, bias mitigation, or semantic editing (Cheong et al., 2022; Chen et al., 2022; Li et al., 2024c; Pan & Bareinboim, 2024). In facial affect analysis, several works leverage counterfactuals to debias or interpret expression classifiers, for example by generating counterfactual faces to study fairness in facial expression recognition (Cheong et al., 2022; Chen et al., 2022), or by introducing counterfactual attention mechanisms and discriminative counterfactual sequences for macro- and micro-expression recognition (Li et al., 2024c). Beyond affect analysis, a broader line of research studies counterfactual image editing and counterfactual image generation using deep generative models, where interventions are performed in pixel space or in a disentangled latent space of semantic attributes (Pan & Bareinboim, 2024; Melistas et al., 2024; Ramesh et al., 2024). Related work on causal representation learning for faces incorporates causal effects of pose or expression into generative pipelines, but still focuses on controllable synthesis rather than structured AU–expression reasoning (Gao et al., 2021).

These causal and counterfactual approaches demonstrate the benefits of intervention-based reasoning for bias mitigation, robustness, and explanation, but they do not address the fine-grained FACS structure that is central to AU-based analysis. They typically intervene at the level of **pixels** or **coarse**

**semantic attributes** (e.g. gender, race), operate with large generative models. Moreover, most of these methods assume a **fixed, global causal structure** tied to a single dataset or attribute set, and do not provide sample-adaptive refinement of the relational structure.

**Our framework differs from prior work in three key aspects**: (1) it performs *causal discovery* of asymmetric, polarity-aware AU–AU and AU–Expression relations; (2) it supports *weakly supervised multi-dataset learning* without joint annotation; (3) it introduces efficient, feature-level *counterfactual interventions* in the disentangled AU latent space.

## I EXPERT VALIDATION STUDY FOR PSYCHOLOGICALLY PLAUSIBLE CAUSAL RELATIONS

To assess whether the discovered AU–AU and AU–Expression causal relations reflect *psychologically plausible mechanisms* rather than dataset-specific statistical patterns, we conducted an additional expert validation study. This section provides the full details on participant recruitment, evaluation materials, experimental procedure, scoring metrics, and inter-expert reliability analysis.

### I.1 PARTICIPANTS

We invited **five domain experts** to independently evaluate the learned causal relations, including three specialists in affective cognition and emotion science, one researcher in cognitive science, and one psychologist with expertise in facial behavior and affect. All experts had **5–15 years of research experience** in their respective fields. Each expert was asked to judge whether the presented AU→AU and AU→Expression relations, which included a mixture of literature-supported edges and randomly generated control relations, were **psychologically plausible** by providing a `True/False` decision together with a confidence rating on a 1–5 scale.

### I.2 EVALUATION MATERIALS

We constructed **three categories of evaluation pairs**, each designed to test a different aspect of the psychological plausibility of the learned causal relations.

**(A) Literature-Supported "True" Relations (39 AU→Expression, excitatory only).** This category contains 39 AU→Expression relations drawn from established FACS documentation and affective-science literature. All relations in this set are excitatory, meaning the AU is known to *increase* the likelihood or intensity of a specific expression. Typical examples include "AU12 (Lip Corner Puller) increases Happiness," "AU4 (Brow Lowerer) increases Anger," and "AU1+AU2 (Inner and Outer Brow Raise) increase Surprise." These relations serve as ground-truth positive anchors, enabling us to verify that experts consistently recognize well-established psychological dependencies.

**(B) CausalAffect-Discovered Relations Requiring Expert Validation (55 AU→AU / 45 AU→Expression, excitatory and inhibitory).** This category contains 100 relations extracted from our learned causal graphs, covering both AU→AU and AU→Expression directions and including both excitatory and inhibitory effects. A subset of these relations overlaps with the literature-supported set (A), demonstrating that the model can recover known ground-truth patterns; the remaining edges constitute genuinely *novel* causal hypotheses, particularly the subtle inhibitory effects highlighted by the reviewer. Examples include "AU17 (Chin Raiser) increases AU24 (Lip Pressor)," "AU26 (Jaw Drop) inhibits AU20 (Lip Stretcher)," "AU15 (Lip Corner Depressor) increases Disgust," and "AU6 (Cheek Raiser) inhibits Sadness." All overlapping relations between sets A and B were removed before expert evaluation. This category tests whether the model's recovered and novel relations correspond to psychologically meaningful causal structure rather than dataset-specific artifacts.

**(C) Random Relations as Negative Controls (68 AU→AU / 28 AU→Expression, excitatory and inhibitory).** This category consists of 96 randomly sampled relations across AU→AU and AU→Expression pairs that appear in neither set A nor set B. These relations include both excitatory and inhibitory directions but lack any known support from FACS or affective psychology. Example items include "AU4 (Brow Lowerer) increases AU12 (Lip Corner Puller)," "AU20 (Lip Stretcher) inhibits Fear," and "AU9 (Nose Wrinkler) increases Happiness." These pairs serve as negative controls,

confirming that experts reliably reject implausible causal statements and providing a baseline against which the model-discovered relations in set B can be assessed.

### I.3 EVALUATION PROCEDURE

All evaluation pairs from sets A, B, and C were merged and deduplicated, then randomly shuffled for each expert independently. Experts were blind to both the originating set (A, B, or C) and whether a relation was literature-supported, recovered by the model, or entirely novel. Each expert judged every relation as **True** (psychologically plausible) or **False** (implausible), and additionally provided a **1–5 confidence score**. This procedure yielded **five independent evaluations per relation**, enabling us to assess both overall plausibility and inter-expert reliability.

### I.4 METRICS AND AGGREGATION

We report several quantitative metrics to evaluate both the psychological plausibility of the relations and the reliability of expert judgments. Throughout this section, let $N$ denote the number of relations being evaluated, and $K$ the number of experts (here $K = 5$). For each relation $i \in \{1, \ldots, N\}$ and expert $e \in \{1, \ldots, K\}$, the binary variable $y_{i,e} \in \{0, 1\}$ represents the expert's plausibility judgment, where 1 indicates that the relation was considered psychologically plausible ("True") and 0 indicates "False". The corresponding confidence rating is denoted by $c_{i,e} \in \{1, 2, 3, 4, 5\}$. The indicator function $\mathbf{1}(\cdot)$ returns 1 if its input condition is satisfied and 0 otherwise.

**Expert True Rate.** The first metric is the overall proportion of "True" decisions across all relations and experts. It quantifies how frequently experts endorse the relations in a given set as psychologically plausible. Formally, this is computed as

$$\text{TrueRate} = \frac{1}{NK} \sum_{i=1}^{N} \sum_{e=1}^{K} \mathbf{1}(y_{i,e} = 1).$$

**Majority-True Rate.** A relation is considered "majority plausible" if at least three of the five experts judged it as "True". Let $\sum_{e=1}^{K} \mathbf{1}(y_{i,e} = 1)$ denote the number of "True" votes for relation $i$. The majority-true rate is defined as

$$\text{MajTrue} = \frac{1}{N} \sum_{i=1}^{N} \mathbf{1}\left( \sum_{e=1}^{K} \mathbf{1}(y_{i,e} = 1) \geq 3 \right).$$

**High-Confidence True Rate.** To assess expert agreement under high certainty, we compute the proportion of "True" votes restricted to evaluations with confidence ratings $c_{i,e} \geq 4$. This metric is given by

$$\text{HighConfTrue} = \frac{\sum_{i=1}^{N} \sum_{e=1}^{K} \mathbf{1}(y_{i,e} = 1 \,\wedge\, c_{i,e} \geq 4)}{\sum_{i=1}^{N} \sum_{e=1}^{K} \mathbf{1}(c_{i,e} \geq 4)}.$$

**Expert Reliability.** To quantify the consistency and reliability of expert judgments, we treat the literature-supported relations in set A as ground-truth positives, and the randomly generated relations in set C as ground-truth negatives. For this analysis, let $TP$ denote the number of true-positive judgments (correctly identifying literature-supported relations as plausible), $TN$ the number of true-negatives, $FP$ the false-positives, and $FN$ the false-negatives. Using these quantities, we compute the standard classification metrics:

$$\text{Acc} = \frac{TP + TN}{TP + TN + FP + FN}, \qquad \text{Prec} = \frac{TP}{TP + FP},$$

$$\text{Rec} = \frac{TP}{TP + FN}, \qquad \text{F1} = 2\,\frac{\text{Prec} \cdot \text{Rec}}{\text{Prec} + \text{Rec}}.$$

Inter-expert agreement is further assessed using Fleiss' $\kappa$:

$$\kappa = \frac{\bar{P} - P_e}{1 - P_e},$$

where $\bar{P}$ is the observed agreement among experts and $P_e$ is the expected chance-level agreement.

**Statistical Comparison Between Model Relations and Random Controls.** To determine whether model-discovered relations (set B) are consistently judged more plausible than random controls (set C), we conduct a two-sided test of difference in proportions. Let $p_B$ and $p_C$ denote the proportion of "True" votes in sets B and C, respectively, and let $n_B = NK$ and $n_C = NK$ represent the total number of evaluations in each set. The pooled proportion is defined as

$$\hat{p} = \frac{p_B n_B + p_C n_C}{n_B + n_C}.$$

The test statistic is:

$$z = \frac{p_B - p_C}{\sqrt{\hat{p}(1 - \hat{p})\left(\frac{1}{n_B} + \frac{1}{n_C}\right)}},$$

and the corresponding two-sided $p$-value is:

$$p = 2(1 - \Phi(|z|)),$$

where $\Phi(\cdot)$ denotes the cumulative distribution function of the standard normal distribution.

## I.5 RESULTS AND ANALYSIS

### I.5.1 OVERALL PLAUSIBILITY COMPARISON

As summarized in Table 8, experts clearly distinguished among the literature-supported relations (set A), the CausalAffect-discovered relations (set B), and the random negative controls (set C). Literature-supported relations received uniformly high plausibility scores, whereas randomly generated relations were overwhelmingly rejected. Crucially, the relations discovered by *CausalAffect* achieved an **Expert True Rate of 0.75**, which is **more than three times higher** than the random baseline (set C: 0.23), and moves substantially toward the plausibility level of literature-supported edges (set A: 0.88). This large separation indicates that the causal dependencies inferred by our model are **not random statistical artifacts**, but instead correspond to psychologically meaningful patterns.

Moreover, the low variance observed in set A (0.21) and set C (0.20) demonstrates that expert judgments were stable and internally consistent when evaluating clearly valid or clearly invalid relations. The higher variance in set B reflects the inherent challenge of evaluating model-proposed causal hypotheses, especially those involving subtle excitatory and inhibitory effects not widely documented in prior literature.

Table 8: Expert validation statistics across the three evaluation sets.

| Relation Set | #Pairs | Expert True Rate | Maj-True Rate | High-Conf True | Interpretation |
|---|---|---|---|---|---|
| **A: Literature-supported** | 39 | $0.88 \pm 0.21$ | 0.92 | 0.90 | Experts strongly endorse well-established psychological relations. |
| **B: CausalAffect** | 100 | $0.75 \pm 0.24$ | 0.84 | 0.70 | Majority of model-discovered relations judged plausible; performance far above random controls. |
| **C: Random negatives** | 96 | $0.23 \pm 0.20$ | 0.09 | 0.19 | Experts reliably reject implausible, randomly generated relations. |

## I.6 RELIABILITY OF EXPERT JUDGMENTS

To assess the reliability of expert evaluations, we focus on the two relation sets with **known correctness**: the literature-supported relations (set A), which are psychologically valid, and the randomly generated relations (set C), which are psychologically invalid. As shown in Table 9, experts demonstrate a strong ability to discriminate between these two reference sets, achieving an **accuracy of 0.87**,

**precision of 0.89**, **recall of 0.88**, and an overall **F1 score of 0.88**. These results indicate that experts rarely misclassify invalid relations as plausible and consistently recognize established psychological dependencies.

Furthermore, the **Fleiss' $\kappa$ of 0.61** reflects **moderate-to-substantial agreement** across the five experts—a level of consensus considered strong for psychological and facial-behavior annotation tasks, where subjective interpretation is unavoidable. Together, these reliability measures confirm that expert assessments are stable, consistent, and suitable as an external validation signal for evaluating the psychological plausibility of the CausalAffect-discovered causal relations.

Table 9: Reliability on ground-truth reference sets (A = true, C = false).

| Metric | Value | Interpretation |
|---|---|---|
| Accuracy | 0.87 | Experts reliably separate valid from invalid relations. |
| Precision | 0.89 | Very low false-positive tendency. |
| Recall | 0.88 | True relations are consistently recognized. |
| F1 Score | 0.88 | Strong overall correctness of expert decisions. |
| Fleiss' $\kappa$ | 0.61 | Moderate-to-substantial agreement among five experts. |

### I.7 SUBTLE INHIBITORY RELATIONS (REVIEWER'S KEY CONCERN)

A central concern raised by the reviewer is whether the **inhibitory causal relations** discovered by CausalAffect may simply reflect statistical artifacts rather than psychologically meaningful mechanisms. To directly address this, we further stratified the CausalAffect-discovered relation set (B) by both **relation type** (AU→AU vs. AU→Expression) and **causal polarity** (excitatory vs. inhibitory), and evaluated expert judgments separately for each subset.

As shown in Table 10, all four subsets receive plausibility scores substantially above the random-negative baseline. In particular, the inhibitory subsets achieve **True Rates of 0.67 (AU→AU)** and **0.77 (AU→Expression)**, together with high majority endorsement ($\geq 0.83$). These values are markedly higher than the random control True Rate (0.23), demonstrating that experts interpret the inhibitory edges as **psychologically plausible**, not as random or dataset-driven correlations. This provides direct evidence that the inhibitory mechanisms captured by CausalAffect correspond to meaningful **regulatory patterns** in facial musculature and emotional expression.

Table 10: Breakdown for CausalAffect-discovered relations (Set B).

| Subset | #Pairs | True Rate | Majority-True Rate | Interpretation |
|---|---|---|---|---|
| AU→AU, excitatory | 32 | 0.72 | 0.78 | Positive AU interactions broadly accepted by experts. |
| AU→AU, inhibitory | 23 | 0.67 | 0.83 | Inhibitory AU–AU relations judged plausible and non-random. |
| AU→Expression, excitatory | 18 | 0.85 | 0.90 | Strong alignment with known affective expression dynamics. |
| AU→Expression, inhibitory | 17 | 0.77 | 0.87 | Inhibitory cues viewed as meaningful psychological structure. |

Taken together, these results show that the causal relations learned by CausalAffect—including the **novel and subtle inhibitory edges** highlighted by the reviewer—are consistently validated by external experts as **psychologically plausible** and are far more meaningful than random controls. Combined with the strong reliability on gold-standard sets (A vs. C), this demonstrates that the inhibitory relations discovered by the model are unlikely to be dataset-specific artifacts and instead constitute valid and interpretable causal hypotheses about AU and expression structure.

## J HYPERPARAMETER DESIGN AND SENSITIVITY ANALYSIS

Although CausalAffect contains multiple components, an important property of our framework is that these modules are **weakly coupled**. Each $\lambda$ hyperparameter regulates the gradient flow **within its own module**, rather than jointly pushing on a single shared representation as in traditional multi-loss architectures. As a result, different $\lambda$ values do not interfere with each other during optimization. Changing one $\lambda$ only affects the **convergence speed** of its corresponding module, without destabilizing the overall training. This modularity also ensures reproducibility: every $\lambda$

is tied to a specific, fully isolated functional component, and all values are kept fixed across all datasets and experiments. We will release all training code, pretrained models, and configuration files upon acceptance. Below, we provide the rationale for each hyperparameter choice together with comprehensive sensitivity analyses.

## J.1 AU BOTTLENECK HYPERPARAMETERS

The AU bottleneck promotes a disentangled and compact AU representation by balancing **information compression**, **semantic alignment**, and **inter-AU decorrelation**. The three hyperparameters operate as follows:

- $\lambda_{\text{ib}}$ regulates HSIC-based information compression.
- $\lambda_{\text{align}}$ aligns AU features with one-hot AU labels.
- $\lambda_{\text{decorr}}$ penalizes covariance to reduce redundancy.

We choose $\lambda_{\text{ib}} = 1$ and $\lambda_{\text{align}} = 1$ so that compression and alignment exert balanced supervision. Smaller values produce under-regularized and noisy AU representations, whereas larger values cause excessive compression or overfitting. The decorrelation term is quadratic in covariance and therefore naturally stronger; it must operate as a lightweight regularizer rather than a dominant learning signal. For this reason, we set $\lambda_{\text{decorr}} = 0.2$, which effectively reduces redundancy while preserving natural physiological AU co-activation patterns.

Table 11: Sensitivity for AU bottleneck hyperparameters.

| $\lambda$ | Value | RAF-DB Acc. | BP4D AU F1 |
|---|---|---|---|
| $\lambda_{\text{ib}}$ | 0.5 | 84.4 | 66.3 |
| $\lambda_{\text{ib}}$ | **1.0 (Best)** | **85.0** | **67.1** |
| $\lambda_{\text{ib}}$ | 2.0 | 84.6 | 66.5 |
| $\lambda_{\text{align}}$ | 0.5 | 84.2 | 66.2 |
| $\lambda_{\text{align}}$ | **1.0 (Best)** | **85.0** | **67.1** |
| $\lambda_{\text{align}}$ | 2.0 | 84.7 | 66.8 |
| $\lambda_{\text{decorr}}$ | 0.1 | 84.7 | 66.6 |
| $\lambda_{\text{decorr}}$ | **0.2 (Best)** | **85.0** | **67.1** |
| $\lambda_{\text{decorr}}$ | 0.4 | 84.2 | 66.4 |

The sensitivity study above highlights three consistent trends. First, both HSIC compression ($\lambda_{\text{ib}}$) and semantic alignment ($\lambda_{\text{align}}$) exhibit a clear *U-shaped* pattern: weak regularization leads to noisy, entangled AU features, whereas overly strong penalties remove essential intra-AU variation. The best results emerge when the two forces are balanced at $\lambda = 1.0$, producing the cleanest and most discriminative AU representations. Second, the decorrelation term behaves differently because of its quadratic nature. A small weight ($\lambda_{\text{decorr}} = 0.2$) effectively suppresses redundancy and encourages disentanglement, but higher values overly suppress natural AU co-activation patterns—especially those grounded in physiological muscle synergies—resulting in measurable performance drops.

Overall, these results confirm that the AU bottleneck benefits from a **balanced tri-regularization** strategy: moderate compression, moderate alignment, and light decorrelation. This combination yields the most faithful and semantically grounded AU latent space, which subsequently enables reliable AU–AU and AU→Expression causal modeling.

## J.2 DAG REGULARIZATION HYPERPARAMETERS

Both the global and sample-adaptive AU→AU graphs employ the same soft acyclicity constraint. When $\lambda_{\text{DAG}}$ is too small, the constraint becomes ineffective and yields dense, noisy, and potentially cyclic structures. When it is too large, the constraint becomes over-restrictive and suppresses meaningful AU dependencies.

A moderate value of $\lambda_{\text{DAG}}^g = \lambda_{\text{DAG}}^s = 0.5$ achieves the best trade-off, preventing cycles while retaining sufficient expressiveness in both global and sample-specific causal graphs.

Table 12: Sensitivity for $\lambda_{\text{DAG}}^g$ and $\lambda_{\text{DAG}}^s$.

| Value | RAF-DB Acc. | BP4D AU F1 | Interpretation |
|---|---|---|---|
| 0.1 | 84.8 | 66.7 | Acyclicity too weak, cyclic edges emerge. |
| **0.5 (Best)** | **85.0** | **67.1** | Best structural fidelity. |
| 1.0 | 84.7 | 66.4 | Over-regularized, graph becomes under-expressive. |

The sensitivity experiment shows that the soft acyclicity regularizer exhibits a clear and stable trend. When the penalty weight is too small ($\lambda_{\text{DAG}} = 0.1$), the acyclicity constraint is insufficient to suppress cyclic or noisy edges, causing the AU→AU graph to become dense and unstable. Such overly flexible structures dilute genuine directional dependencies and reduce downstream performance. Increasing the penalty to a moderate value ($\lambda_{\text{DAG}} = 0.5$) yields the best results by enforcing acyclicity strongly enough to remove spurious cycles while still allowing the graph to encode physiologically meaningful AU interactions. When the penalty becomes too large ($\lambda_{\text{DAG}} = 1.0$), the graph becomes overly sparse and under-expressive: important AU relations are pruned away, limiting the capacity of both the global and sample-adaptive graph to model valid causal pathways. These results confirm that a **moderate** acyclicity weight achieves the optimal balance between structural faithfulness and expressive flexibility.

### J.3 COUNTERFACTUAL SUPERVISION HYPERPARAMETERS

Counterfactual supervision enforces two complementary principles:

- **Consistency**: factual and counterfactual predictions remain similar when the intervened AU is not a causal parent.
- **Discrepancy**: predictions diverge when the intervened AU *is* a causal parent.

These forces must be balanced. Too small a weight weakens counterfactual reasoning; too large a weight leads to trivial solutions. We tie both hyperparameters and set $\lambda_{\text{consist}} = \lambda_{\text{discrep}} = 0.5$, achieving the desired equilibrium.

Table 13: Sensitivity for counterfactual supervision.

| $\lambda_{\text{consist}} = \lambda_{\text{discrep}}$ | RAF-DB Acc. | BP4D AU F1 | Interpretation |
|---|---|---|---|
| 0.25 | 84.0 | 66.2 | Penalty too weak, interventions ineffective. |
| **0.5 (Best)** | **85.0** | **67.1** | Balanced consistency–discrepancy. |
| 1.0 | 84.4 | 66.4 | Penalty too strong, suppresses causal diversity. |

The sensitivity results highlight the importance of balancing the two counterfactual forces: consistency and discrepancy. When the shared penalty weight is too small ($\lambda_{\text{consist}} = \lambda_{\text{discrep}} = 0.25$), the model does not sufficiently penalize mismatches between factual and counterfactual predictions, causing the intervention mechanism to become ineffective. In this regime, the model tends to ignore counterfactual signals and relies solely on factual supervision, which weakens its ability to distinguish causal from non-causal dependencies.

At the opposite extreme, setting the penalty to a large value ($\lambda = 1.0$) overemphasizes counterfactual constraints and encourages trivial behaviors: either enforcing all predictions to remain invariant (over-consistency) or forcing unnecessary divergence (over-discrepancy). Such collapse reduces causal diversity and slightly destabilizes factual predictions.

The intermediate value $\lambda = 0.5$ achieves the optimal balance. It enforces sufficient consistency to suppress non-causal edges while maintaining enough discrepancy to highlight true causal parents. This balanced setting yields the strongest overall performance and produces the most faithful counterfactual behavior in both AU detection and expression recognition tasks.

# K  QUALITATIVE ANALYSIS FROM A VIDEO AND TEMPORAL PERSPECTIVE

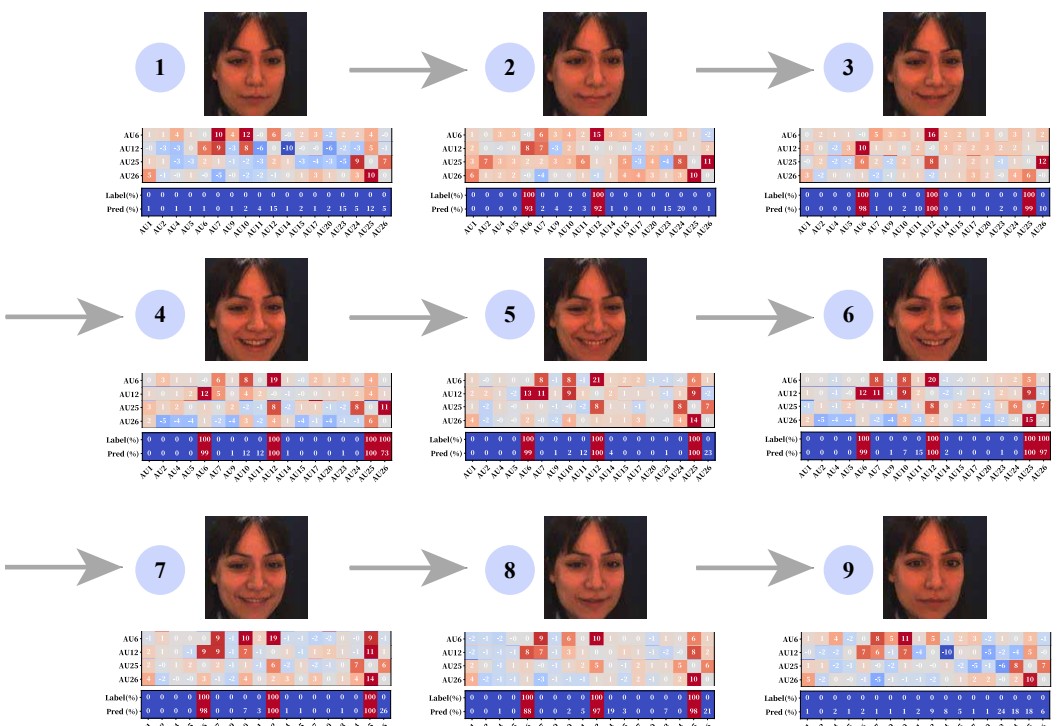

Figure 8: Visualization of the sample-adaptive AU-to-AU causal graphs extracted from a nine-frame sequence, sampled at every 5 frames. Each row represents the outgoing causal influences of a source AU toward all target AUs. Warmer colors indicate excitatory causal effects, while cooler colors reflect inhibitory influences. The causal structures exhibit clear temporal dynamics, with weak and diffuse relations during the onset phase, a tightly coupled AU6–AU12–AU25–AU26 subgraph at the smile apex, and a progressive attenuation of causal connectivity during the offset phase.

To understand how facial Action Units (AUs) interact throughout the evolution of an expression, we examine the AU→AU causal graphs extracted directly from a continuous video sequence. As facial expressions unfold over time rather than in isolated static frames, analyzing the causal structure from a temporal perspective is essential for capturing the dynamics of muscle coordination. From the full video, we sample one frame every 5 frames to obtain nine representative snapshots that jointly cover the onset, apex, and offset phases of a smile. For each selected frame, the model infers a sample-adaptive AU→AU causal graph, where each row reflects the causal influence of a source AU on all other target AUs.

From a video-centric viewpoint, the causal patterns exhibit a temporal trajectory that closely parallels the physical progression of the expression. In the early onset frames, the causal heatmaps remain **weak** and **spatially diffuse**, consistent with the gradual and subtle activation of facial muscles as the smile begins to form. The video at this stage shows only mild motion around the eyes and mouth, and correspondingly, the inferred causal interactions are sparse and low in magnitude.

As the expression intensifies **toward the apex**, the video shows increasingly coordinated facial movements: eye corner tightening, lip corner elevation, and progressive mouth opening. From the perspective of positive reciprocal causation, this stage is characterized by mutually reinforcing causal interactions among AU6, AU12, AU25, and AU26. Instead of a unidirectional driving pattern, these AUs exert bidirectional excitatory influences on one another that amplifies the overall expressiveness of the smile. This reciprocal structure is clearly reflected in the causal graphs: each of the four AUs not only contributes to the activation of the others but is simultaneously strengthened by them, producing a dense and self-supporting causal subgraph. Such positive feedback within the AU system gives rise to the compact temporal signature observed at the smile's peak, demonstrating that the

model captures how facial components co-evolve through dynamically reinforced causal links rather than isolated muscle activations. The strong consistency between the video's motion synchrony and the inferred reciprocal causal patterns confirms that the learned structure is deeply aligned with the underlying biomechanical coordination of facial expressions.

During the offset phase, the facial movements in the video gradually relax, and the expression fades back toward neutrality. The inferred causal graphs reflect this temporal decay: the previously strong excitatory relations weaken, high-intensity regions in the heatmaps dissipate, and the graph returns to a more diffuse and low-activation configuration. The temporal alignment between decreasing motion energy in the video and weakened causal coupling further validates the model's responsiveness to the evolving visual dynamics.

Viewed from a video and sequence modeling perspective, this analysis demonstrates that the causal graphs evolve in synchrony with the temporal flow of the expression. The sample-adaptive causal structures not only capture the instantaneous AU dependencies in each frame but also track how these dependencies grow, peak, and dissipate in accordance with the natural motion pattern of the underlying video. This highlights the model's ability to perform fine-grained temporal causal reasoning, making it well-suited for understanding dynamic facial behavior beyond static image analysis.

