# Affective Causal Graph Learner for Human-Aligned Facial Behavior Understanding

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

| Total Training Time | $\sim$5.8h | $\sim$1.29h | $\sim$0.57h | $\sim$0.46h |
| Peak GPU Memory Usage | 28–30 GB | 18–20 GB | 18–19 GB | 10–12 GB |
| Inference Speed | $\sim$200 FPS | $\sim$250 FPS | $\sim$250 FPS | $\sim$270 FPS |

Table 4: Training and inference efficiency of CausalAffect across different experimental configurations.

**Large-Scale Configuration:** The largest-scale training setting corresponds to **CausalAffect (+All)**, which involves approximately 800K images from six datasets. A batch size of 360 (60 images per dataset) was used. Training for 80 epochs required around 260 seconds per epoch, resulting in a total training time of $\sim$5.8 hours. The peak GPU memory usage was between 28–30 GB, while the inference speed reached $\sim$200 FPS. This demonstrates that CausalAffect remains computationally tractable even at large scale, making it feasible for deployment on high-performance GPUs.

**Medium- and Small-Scale Configurations:** For smaller-scale experiments, training time and memory requirements were substantially reduced: **AffectNet+BP4D** ($\sim$257K images): batch size 120, training $\sim$1.29 hours, memory usage 18–20 GB, inference speed $\sim$250 FPS. **DISFA+RAF-DB** ($\sim$100K images): batch size 120, training $\sim$0.57 hours, memory usage 18–19 GB, inference speed $\sim$250 FPS. **EmotioNet** ($\sim$24K images): batch size 60, training $\sim$0.46 hours, memory usage 10–12 GB, inference speed $\sim$270