# OpenReview forum: "CausalAffect: Causal Discovery for Facial Affective Understanding"
_ICLR.cc/2026/Conference — Submitted to ICLR 2026_

### Official Review · Reviewer_AkV1 · 2025-10-30

**Soundness:** 3
**Presentation:** 3
**Contribution:** 2
**Rating:** 4
**Confidence:** 3

**Summary:**

This work proposes CausalAffect - a framework for causal graph discovery for facial affect analysis. Given facial images, this framework recovers a causal graph graph describing dependencies between facial activation units and facial expressions (i.e., AU -> AU and AU -> Expression). The authors introduce several specific mechanisms, including sample-adaptive learning and feature-level counterfactual interventions, designed to improve graph discovery. The authors validate the framework empirically on six benchmark datasets, illustrating that it improves the accuracy of AU and expression predictions, and recovers causal graphs that are consistent with existing psychological theories.

**Strengths:**

In this work, the authors propose the first (to my knowledge) approach for causal discovery in facial affect data. One key benefit of the proposed approach is that it can learn in a semi-supervised manner, and does not require joint annotations of AUs and facial expressions. This enables the proposed approach to ingest data from disjoint datasets during learning.

The empirical validation of the work also has several string dimensions: the authors compare against a broad set of relevant baselines, illustrate accuracy improvements provided by the approach, and report interesting interpretive analysis of the scientific insights recovered by the framework (Figure 2; Appendix C). The authors also provide an interesting comparative analysis of the Social Smile and Duchenne Smile, and provide also perform hyperparameter sensitivity and ablation studies. While minor, CausalAffect is also a great name for the framework.

**Weaknesses:**

## Clarity of Causal Framework and Formal Assumptions

One of my key concerns with this work is that it provides an incomplete motivation for why a causal perspective is important to this problem, and why conceptually a causal approach improves over the status-quo in AU/Expression recognition.

After reading the work several times, it remains unclear what cognitive or behavioral model this work instantiates. AU -> Expression, it could also be the case that Expression  -> AU, or that there is a third unmeasured variable that is manifesting in both AUs and emotion expressions. When investigating such theories, I would expect to see a formal Structural Equation Model or DAG with accompanying references supporting the model. The term "relations" used throughout the work maintains this ambiguity and does not clearly identify the proposed causal pathways that are under study.

- For a causal discovery problem, I would expect a formal statement of causal and statistical assumptions needed to support inference, matched to the DAG or SEM above.
- The soft DAG constraint enabling cycles in the AU → AU pathway violates the acyclicity requirement of causal DAGs. This makes me question the causal model under study. I think this could nicely be viewed as a time series problem where there is a fixed DAG with states that vary as a function of time t. This could obviate the cyclic issues while also supporting the dynamics the authors mention. Generally adopting a time-dependent formulation would be natural in this setting.

In sum, there is a key gap between the paper's causal framing and its methodology. While the paper claims to "infer psychologically plausible causal relations" and "enforce genuinely causal relations", the absence of a formal causal model, stated identification assumptions, and the violation of acyclicity principles means that the learned relationships cannot be interpreted causally. The method may learn useful structured dependencies, but these are correlational rather than causal. I recommend the authors either: (1) formalize their causal framework with appropriate assumptions and modify the architecture to respect causal principles, or (2) reposition the work as learning interpretable feature relationships via a semi-supervised approach, which would still be a quite valuable contribution.


## Presentation of Technical Framework
Figure 1 is very helpful for understanding the framework and the paper is well-written overall. However, in places throughout Section 3, I found myself lost in the details of the approach with limited rationale for why these details matter. In particular, subsections 3.1-3.5 currently read as a recipe of technical details rather than providing a unified rationale for why these components are needed. For example, it's unclear why Counterfactual Interventions are necessary given HSIC-Based Disentanglement, or why sample-adaptive graphs are necessary to support heterogeneous nodes. The empirical ablation study illustrates the empirical impact of these decisions but offers a limited conceptual basis for why each component is necessary.  The authors could address this by easing the notation / moving some details to the appendix to make space for more rationale for the design decisions and sharing why these problems are technically challenging.

Related to the comment above, one place where I was especially missing the rationale is Section 3.3. Why would Social and Duchenne smiles require *different graph structures* rather than *different paths* through the same graph? If we conceptualize each AU as a random variable, the causal graph structure should remain invariant across realizations—only the values of these variables should change. A graph that changes structure conditional on observed data is at odds with standard causal modeling. Note that addressing the first point (Clarity of Causal Framework) could also resolve this concern.

## Empirical Validation

While the empirical validation has several strengths (noted above) there are also a few weaknesses. Foremost, the authors do not report statistical uncertainty in Table 1 and elsewhere in the results. This makes it challenging to assess whether the claims hold across settings.

Further, based on the current results, it is unclear what the foundational goals of the empirical evaluation should be. I was surprised to see prediction accuracy featured so prominently in a causal discovery work. For causal inference, the primary validation should be whether the method recovers the true causal structure, not whether it improves predictive performance (which can be achieved by learning spurious correlations). To validate the approach, the authors may want to design a synthetic or semi-synthetic experiment illustrating that the approach can recover the ground truth when the true causal structure is known. This would provide valuable evidence substantiating that the learned causal relations (Fig 2) are valid.

Finally, my conceptual confusion surrounding the causal framework extends to the  experiments. How would it be possible to recover a causal graph with edges that  vary conditional on a single image (Fig. 3)? This reinforces the concern that the  learned relationships may be correlational rather than causal.

More positively, a compelling visualization could show graph activation over AUs as a function of time—e.g., showing frames from a video sequence. This would illustrate the temporal activation patterns underpinning the graph structure and would align naturally with the time-series formulation suggested earlier.

**Questions:**

Conceptual:
- What is the proposed causal model underlying this approach? Specifically, does it posit that AU → Expression, Expression → AU, or both directions exist? Could you provide a formal DAG or SEM that represents your theoretical model? How does temporality connect to this formulation?
- What causal and statistical assumptions are necessary for your method to identify causal relationships rather than correlations? For instance, do you assume causal sufficiency (no unmeasured confounders between AUs and expressions)?
- Could you clarify the theoretical justification for having graph structures that vary across individual images?

Empirical:
- To what extent can the accuracy benefits reported in Table 1 be attributed to increased dataset size available to CausalAffect?
- How do the reported empirical results vary across training runs and random seeds?

---

> ### Author Response · Authors · 2025-11-25
> **Author's Response - Thank you for your feedback (part 1)**
>
> We sincerely thank the Reviewer for the thoughtful and valuable feedback. We appreciate the opportunity to improve the clarity and quality of our work. All modifications made in the manuscript are highlighted in **Blue** and **Red**. Our detailed responses to each comment are provided below.
>
> ----
>
>
> ### **1. Why a Causal Perspective is Important for AU/Expression modeling**
>
> > One of my key concerns with this work is that it provides an incomplete motivation for why a causal perspective is important to this problem, and why conceptually a causal approach improves over the status-quo in AU/Expression recognition.
>
> A causal formulation is important because it enables **interpretable and mechanism-driven emotion recognition**, which cannot be achieved by correlation-based deep learning models. Our causal graph allows the system to identify:
> - _**which**_ facial muscle activations contributed to a predicted expression,
> - _**how**_ AU interactions propagated through the graph, and
> - _**why**_ a particular expression was produced (e.g., due to excitatory activation or inhibitory suppression).
> This mirrors the way human experts reason about facial behavior and provides explanations that are meaningful and structured.
>
> **From a modeling perspective**, predicting expressions from **sparse AU features** through a causal graph compels the model to enhance the **representational capacity** of the AU space. In addition, counterfactual intervention further **sharpens** the selection of informative AUs and **stabilizes** the learned causal structure.
>
> Together, these mechanisms explain the observed performance gains in Table 1 (main paper): the causal formulation provides **human-aligned interpretability** and delivers **higher predictive accuracy** than conventional correlation-based approaches.
>
>
> ---
>
>
>
> ### **2. Cognitive Model Instantiated in CausalAffect**
>
> > After reading the work several times, it remains unclear what cognitive or behavioral model this work instantiates. AU -> Expression, it could also be the case that Expression -> AU, or that there is a third unmeasured variable that is manifesting in both AUs and emotion expressions. When investigating such theories, I would expect to see a formal Structural Equation Model or DAG with accompanying references supporting the model. The term "relations" used throughout the work maintains this ambiguity and does not clearly identify the proposed causal pathways that are under study.
> > - For a causal discovery problem, I would expect a formal statement of causal and statistical assumptions needed to support inference, matched to the DAG or SEM above.
> >
> > Question 1: What is the proposed causal model underlying this approach? Specifically, does it posit that AU → Expression, Expression → AU, or both directions exist? Could you provide a formal DAG or SEM that represents your theoretical model? How does temporality connect to this formulation?
>
> **2.1 Why AU → Expression?**
>
> Our causal direction follows a well-established account in psychology and affective science that treats **facial Action Units (AUs) as the anatomical generators of expressions**, not their consequences.
> - This principle is central to the **Facial Action Coding System (FACS)** [1], where each emotion expression is explicitly defined as a _composition of underlying muscle activations_. **Ekman and Rosenberg** [1, 2, 3] describe AUs as the _“building blocks”_ of expressions, supporting a bottom-up generative hierarchy rather than an Expression→AU feedback process.
> - This causal ordering is further supported by **Componential Theories** of emotion, which view facial expressions as the outcome of multiple causally interacting components (e.g., appraisals, physiology, AUs). Within this framework, individual _AUs are treated as the motor-expression components that encode specific appraisal outcomes_, and emotional expressions emerge from _the combination and integration_ of these AU-level causes (e.g., Smith & Scott [4]; Sell et al.[5]). A central representative is the **Component Process Model (CPM)** of Scherer & Ellgring [7, 8, 9 ,10], which formalizes the causal cascade: cognitive–emotional appraisals first trigger specific AU activations, and the accumulated AU pattern generates the final observable expression. Thus, **CPM provides a direct theoretical instantiation of the Componential perspective and gives explicit causal grounding to the AU → Expression direction modeled in our work**.
> - **Empirical studies** also validate this direction: artificially manipulating AU configurations alone produces recognizable emotional expressions. For example, Wehrle et al.[6] show that synthetic faces generated purely from AU patterns are reliably classified by human observers, demonstrating that AUs are sufficient to _produce_ perceived expressions.

---

> ### Author Response · Authors · 2025-11-25
> **Author's Response - Thank you for your feedback (part 2)**
>
> **(continued authors' response)**
>
> **2.2 AU → AU Positive Reciprocal Causation (Muscle Synergy, Not a Hard DAG)**
>
> Here we explain the cognitive and neuroscience–biobehavioral foundations for why AU → AU represents genuine causal influence, and why (as further clarified in **Answer 3: DAG in CausalAffect**) the AU layer cannot be modeled as a hard DAG.
>
> - From both cognitive theory and neuroscience, there is strong evidence that facial AUs do not behave as independent units. Instead, they participate in **muscle synergies**, where the activation of a **_primary_** muscle causally drives the activation of **_secondary_** or **_adjacent_** muscles. **Motor Synergy Theory** and the **Biobehavioral Model** [11, 12, 13 ] show that many facial muscles share nerve branches, overlapping motor units, or soft-tissue linkages. Classic work by **Hess & Gimenez** [11], as well as EMG studies by _Cacioppo et al._ [12] and _Tassinary_ et al. [13], demonstrates that contraction in one facial muscle can **causally facilitate (Positive Reciprocal Causation)** activation in nearby muscles—even when those secondary muscles are not voluntarily engaged. This creates **directional influences, physiology-based causal influences** within the AU layer.
> - Importantly, **Scherer’s Component Process Model (CPM) [7, 8, 9, 10]** is fully consistent with this interpretation. In CPM, AUs jointly form the **motor-expression component**, where activation of core, high-tension AUs can mechanically or neurally recruit additional AUs to complete a coherent facial configuration. In practical terms, this reflects a causal sequence in which a **primary AU initiates a synergy**, and its activation **partially induces the engagement of secondary AUs**, with the overall expression emerging from this coordinated multi-muscle activation pattern.
>
> Therefore, the AU stage is best understood as a **synergy-structured causal subnetwork**, not a strictly acyclic hierarchy (DAG). In our method, the DAG formulation is not meant to impose a hard acyclic assumption on facial physiology, but rather to serve as a **regularization mechanism** that encourages the learned AU → AU structure to be **sparse, directional, and dominated by strong causal influences**. The soft-DAG constraint biases the model toward identifying the most meaningful causal pathways while still allowing the **limited, biologically grounded reciprocal interactions** that naturally arise from facial muscle synergies.
>
> **2.3 Universal and Individual Factors in Causation**
>
> In addition to the CPM-based causal chain and AU-level muscle synergies, our formulation is also informed by findings from **cross-cultural theories of facial expression**. Research by **Rachael E. Jack and colleagues** [14, 15] demonstrated that human facial expressions exhibit both **culturally universal components** and **culturally specific modulations** (e.g., differences in intensity patterns or AU combinations across cultures). These theories suggest that facial behavior is governed simultaneously by:
> - a **shared, population-level structure** that reflects stable biological and communicative regularities, and
> - **individualor culture-specific variability** that depends on context, social norms, and idiosyncratic expression style.
>
> Motivated by this theoretical insight, we extend the CPM-inspired causal interpretation by modeling the AU → Expression pathway at **two complementary levels**:
> 1. **Population-level (global) causal graph**: Captures the globally consistent AU→Expression mappings that reflect universal behavioral regularities documented in both FACS and cross-cultural affective science.
> 2. **Instance-level (sample-adaptive) causal graph**: Allows deviations from the population structure to account for person-specific facial habits, contextual cues, and subtle cultural modulation described in Jack’s cross-cultural work.
>
> This dual-level modeling perspective aligns naturally with psychological theory: it respects the universal biological basis of facial expressions while remaining flexible enough to accommodate meaningful interpersonal and cross-cultural variability.

---

> ### Author Response · Authors · 2025-11-25
> **Author's Response - Thank you for your feedback (part 3)**
>
> **(continued authors' response)**
>
> **2.4 Formal Structural Equation Model (SEM).**
>
> **(i) Component Process Model (CPM)**: We first state the **full CPM causal cascade** [7–10] in SEM form. CPM posits that facial expressions are generated through a sequential process in which **latent appraisals** trigger **AU activations**, and the resulting AU configuration produces the **expression**. Let $U$ denote appraisal-related latent causes, $A_i$ denote the activation of the $i$-th Action Unit, and $Y$ denote the expression label. The CPM-consistent SEM is:
>
> $$A_i = f_i\left(U, \mathbf{A}_{\text{Pa}(i)}\right),  \qquad  Y = g(\mathbf{A})$$
>
> where $i \in {1,\dots,N_{\text{AU}}}$ indexes AUs, and $\mathbf{A}_{\text{Pa}(i)}$ denotes the set of AU parents of $A_i$ capturing synergy-based AU→AU influences within the motor-expression stage.
>
> **(ii) CausalAffect models only the latter half of CPM**: In standard facial datasets, the appraisal variables $U$ are **unobserved external/internal cognitive states**, so they are not identifiable from visual data alone. Therefore, our method focuses on the empirically accessible latter part of CPM, and treats appraisals as upstream latent causes outside the modeled graph. Concretely, we model:
>
> $$A_i = f_i\left(\mathbf{A}_{\text{Pa}(i)}\right),  \qquad  Y = g(\mathbf{A})$$
>
> i.e., we learn (1) **directional AU → AU structure** driven by muscle synergies and (2) **AU → Expression causation**, which together correspond to the observable back half of the CPM chain.
>
> **(iii) Extending CPM with Two-level (global + sample) Causal Structure**
>
> Cross-cultural research by **Rachael E. Jack** [9,10] shows that human facial expressions contain both **universal, population-wide components** and **cultureor individual-specific variations**. This implies that the causal mechanisms governing facial behavior operate at two complementary levels: a **global structure** reflecting universally shared AU→AU synergies and AU→Expression mappings, and a **sample-adaptive structure** that accounts for identity, context, and culture-dependent modulation. To incorporate this insight into the SEM, we decompose the parents of each AU into global and sample-specific components,
>
> $$\mathbf{A}\_{\mathrm{Pa}(i)}=(\mathbf{A}\_{\mathrm{Pa}(i)}^{g}, \mathbf{A}\_{\mathrm{Pa}(i)}^{s})$$
>
> and express the CPM-consistent causal mechanism compactly as
>
> $$A\_i = f^{g}\_i\left(\mathbf{A}^{g}\_{\mathrm{Pa}(i)}\right) \oplus  f^{s}\_i\left(\mathbf{A}^{s}\_{\mathrm{Pa}(i)}\right),  \qquad  Y = g^{g}(\mathbf{A}) \oplus  g^{s}(\mathbf{A}).  $$
>
> Here, $f\_i^{g}$ and $g^{g}$ correspond directly to the  the **global AU→AU graph** and the **global AU→Expression graph** learned in **CausalAffect**. These components model the stable, universally shared causal regularities of facial behavior. In contrast, $f\_i^{s}$ and $g^{s}$ correspond to our **Sample-adaptive AU→AU graph** and **Sample-adaptive AU→Expression graph**, which capture individual differences, contextual influences, and culture-dependent modulation.

---

> ### Author Response · Authors · 2025-11-25
> **Author's Response - Thank you for your feedback (part 4)**
>
> **(continued authors' response)**
>
>
>
> [1] Ekman, Paul, and Wallace V. Friesen. "Facial action coding system." _Environmental Psychology & Nonverbal Behavior_ (1978).
> [2] Ekman, Paul, and Wallace V. Friesen. _Unmasking the face: A guide to recognizing emotions from facial clues_. Ishk, 2003.
> [3] Ekman, Paul, and Erika L. Rosenberg, eds. _What the face reveals: Basic and applied studies of spontaneous expression using the Facial Action Coding System (FACS)_. Oxford University Press, USA, 1997.
> [4] Smith, C., and Scott, H. (1997). “A componential approach to the meaning of facial expressions,” in _The Psychology of Facial Expression_, eds J. A. Russell and J. M. Fernández-dols (New York, NY: Cambridge University Press), 229–254.
> [5] Sell, Aaron, Leda Cosmides, and John Tooby. "The human anger face evolved to enhance cues of strength." _Evolution and Human Behavior_ 35.5 (2014): 425-429.
> [6] Wehrle, Thomas, et al. "Studying the dynamics of emotional expression using synthesized facial muscle movements." _Journal of personality and social psychology_ 78.1 (2000): 105.
> [7] Scherer, Klaus R., and Heiner Ellgring. "Are facial expressions of emotion produced by categorical affect programs or dynamically driven by appraisal?." _Emotion_ 7.1 (2007): 113.
> [8] Scherer, Klaus R., Marcello Mortillaro, and Marc Mehu. "Understanding the mechanisms underlying the production of facial expression of emotion: A componential perspective." _Emotion Review_ 5.1 (2013): 47-53.
> [9] Wehrle, Thomas, et al. "Studying the dynamics of emotional expression using synthesized facial muscle movements." _Journal of personality and social psychology_ 78.1 (2000): 105.
> [10] Scherer, K. _What does a facial expression express_. 1992.
> [11] Hess, Ursula, et al. "The facilitative effect of facial expression on the self-generation of emotion." _International Journal of Psychophysiology_ 12.3 (1992): 251-265.
> [12] Cacioppo, John T., et al. "Electromyographic activity over facial muscle regions can differentiate the valence and intensity of affective reactions." _Journal of personality and social psychology_ 50.2 (1986): 260.
> [13] Tassinary, Louis G., and John T. Cacioppo. "Unobservable facial actions and emotion." (1992): 28-33.
> [14] Jack, Rachael E., et al. "Four not six: Revealing culturally common facial expressions of emotion." _Journal of Experimental Psychology: General_ 145.6 (2016): 708.
> [15] Jack, Rachael E., et al. "Facial expressions of emotion are not culturally universal." _Proceedings of the National Academy of Sciences_ 109.19 (2012): 7241-7244.

---

> ### Author Response · Authors · 2025-11-25
> **Author's Response - Thank you for your feedback (part 5)**
>
> ### **3. DAG in CausalAffect**
>
>
> > The soft DAG constraint enabling cycles in the AU → AU pathway violates the acyclicity requirement of causal DAGs. This makes me question the causal model under study.
>
> The key point is that **our model contains two fundamentally different causal structures** (AU→Expression: **DAG causation** , AU→AU: **Positive Reciprocal Causation**), each with different semantic requirements:
>
> **(i) AU→Expression is Naturally a DAG Causation.**
> As clarified above, the AU→Expression pathway models the causal generative process through which muscular activations give rise to facial expressions. This process is inherently **acyclic**: expressions are semantic outcomes. Consequently, AU→Expression forms a strictly directional graph, fully aligning with DAG semantics as well as established cognitive and psychological theory.
>
> **(ii) AU→AU Does _Not_ Follow Strict DAG Semantics; It Models _Positive Reciprocal Causation_.**
> We fully agree that classical DAG formulations prohibit cycles. However, when DAG applied to **complex neural, ecological, circulatory, or other biological systems**, strict DAGs are often inadequate for capturing the genuine causal mechanisms at play. A substantial body of work [16, 17, 18, 19, 20] demonstrates that many real-world systems exhibit **positive reciprocal causation**, where two components exert mutually reinforcing causal influence. These systems remain fundamentally causal, yet **cannot be faithfully represented by a strictly acyclic DAG**.
>
> The AU→AU pathway follows the same principle. Facial muscles frequently engage in **biomechanical synergy**, where mechanical coupling produces _directional but not necessarily symmetric_ causal effects. For example:
> - When **AU15** (lip-corner depressor) is dominant, the downward pull on the mouth increases chin tension, thereby **inducing AU17**.
> - Conversely, when **AU17** (chin raiser) is dominant, upward chin pressure pushes the lip corner downward, thereby **activating AU15**.
>
> Under this perspective, the soft DAG constraint in our AU→AU branch is introduced **only as a regularizer**—to encourage sparsity and reduce spurious symmetric patterns. It should _not_ be interpreted as asserting that the AU→AU structure itself is a strict DAG. On the contrary, our AU→AU graph is explicitly designed to capture **directional yet not necessarily acyclic** causal relations. As illustrated in Figure 2 (Global AU→AU Causation):
> - **Each row** specifies _which AUs causally **drive** a particular target AU_ (the AU as an effect),
> - while **each column** identifies _which AUs are causall **influenced** by that AU_ (the AU as a cause).
>
> This produces an asymmetric, direction-aware causal structure that faithfully reflects the biomechanics of facial muscle interactions, while still allowing localized reciprocal pairs where they are physiologically justified.
>
> [16] Svensson, Erik I. "On reciprocal causation in the evolutionary process." Evolutionary Biology 45.1 (2018): 1-14.
> [17] Hazelwood, Caleb. "Reciprocal causation and biological practice." Biology & Philosophy 38.1 (2023): 5.
> [18] Baedke, Jan, Alejandro Fábregas-Tejeda, and Guido I. Prieto. "Unknotting reciprocal causation between organism and environment." Biology & Philosophy 36.5 (2021): 48.
> [19] Aalen, Odd Olai, et al. "Can we believe the DAGs? A comment on the relationship between causal DAGs and mechanisms." Statistical methods in medical research 25.5 (2016): 2294-2314.
> [20] Rohbeck, Martin, et al. "Bicycle: Intervention-based causal discovery with cycles." Causal Learning and Reasoning. PMLR, 2024.

---

> ### Author Response · Authors · 2025-11-25
> **Author's Response - Thank you for your feedback (part 6)**
>
> ### **4. Time-varying DAG？**
>
> > I think this could nicely be viewed as a time series problem where there is a fixed DAG with states that vary as a function of time t. This could obviate the cyclic issues while also supporting the dynamics the authors mention. Generally adopting a time-dependent formulation would be natural in this setting.
>
>
> **(i) On the “cyclic issues”**: As discussed in **Answer 2 and 3**, the cycles observed in our AU→AU structure arise from well-documented **Positive Reciprocal Causation** within facial **muscle synergies**. Facial muscles frequently exhibit **bi-directional, physiology-driven influences** (e.g., AU1⇄AU2, AU9⇄AU10) due to co-innervation and mechanical coupling. These reciprocal interactions are _not_ temporal loops but **instantaneous biomechanical dependencies**, meaning that enforcing a strictly acyclic AU→AU DAG would misrepresent the true underlying physiology.
>
> **(ii) On the “dynamic” interpretation: the reviewer’s fixed DAG corresponds exactly to our global graph**: The reviewer’s suggestion of a *“fixed DAG whose states vary over time t”* maps naturally onto our model components:
> - Our global AU→AU and AU→Expression graphs serve as the reviewer’s **fixed Graph**, representing population-level causal structure that remains stable across instances.
> - Variation in AU feature (analogous to changing “**states over time**”) already results in different predictions and explanations when propagated through this global graph.
>
> In other words, the reviewer’s dynamic formulation is **functionally equivalent** to using a stable population-level causal graph with varying AU inputs—as we already do.
>
> **(iii) Why We Introduce Sample-adaptive Graphs Instead of Relying Solely on a Fixed Time-series Graph**
>
> While a single fixed graph may capture universal structural relationships, it cannot represent the **individual, contextual, and cultural variability** extensively documented in cross-cultural affective science as we mentioned in **Answer 2** (e.g., Rachael Jack). Even when cultures share the same **core AU configuration** for positive emotions—such as the universal basis pattern **AU12 + AU6** (the canonical smile)—different populations systematically **layer additional AUs** to intensify or nuance the expression. For example, Western “delighted” smiles frequently recruit muscles **AU7, AU25, and AU26** on top of the core pattern, whereas East Asian cultures often rely on a **more restrained upper-face pattern** and modulate intensity primarily through **changes in the mouth region** rather than adding strong ocular AUs. A single fixed graph would incorrectly treat these additional AUs, when in reality they reflect cultural specificity—and thus require causal strengths that adapt at the instance or cultural level.
>
> Accordingly, in our CausalAffect,
> - the **global graph** provides the universal causal backbone (the reviewer’s “fixed graph”),
> - while the **sample-adaptive graph** acts as a residual that adjusts causal strengths on a per-instance basis.
>
> This two-level structure preserves the stability of a fixed causal model while providing the flexibility necessary to capture real-world variability, fully consistent with the CPM framework and the physiology of facial muscle synergy.

---

> ### Author Response · Authors · 2025-11-25
> **Author's Response - Thank you for your feedback (part 7)**
>
> ### **5. Other comments on Clarity of Causal Framework and Formal Assumptions**
>
>
> > In sum, there is a key gap between the paper's causal framing and its methodology. While the paper claims to "infer psychologically plausible causal relations" and "enforce genuinely causal relations", the absence of a formal causal model, stated identification assumptions, and the violation of acyclicity principles means that the learned relationships cannot be interpreted causally. The method may learn useful structured dependencies, but these are correlational rather than causal. I recommend the authors either: (1) formalize their causal framework with appropriate assumptions and modify the architecture to respect causal principles, or (2) reposition the work as learning interpretable feature relationships via a semi-supervised approach, which would still be a quite valuable contribution.
>
> - _"The absence of a formal causal model, stated identification assumptions"_ and "_(1) formalize their causal framework with appropriate assumptions and modify the architecture to respect causal principles,(2) reposition the work as learning interpretable feature relationships via a semi-supervised approach, which would still be a quite valuable contribution._":
>     - clarified in **Answer 1. Why a Causal Perspective is Important for AU/Expression modeling** and
>     - **Answer 2. Cognitive Model Instantiated in CausalAffect**
> - _"The violation of acyclicity principles means that the learned relationships cannot be interpreted causally, The method may learn useful structured dependencies, but these are correlational rather than causal. The method may learn useful structured dependencies, but these are correlational rather than causal"_.
>     - clarified in **Answer 3. DAG in CausalAffect**
>
> ---
>
> ### **6. Why HSIC Disentanglement and Counterfactual Interventions Are Both Necessary**
>
> > it's unclear why Counterfactual Interventions are necessary given HSIC-Based Disentanglement
>
> HSIC-based disentanglement and Counterfactual Interventions address _different_ causal requirements in our framework and are **complementary**. The former provides **clean causal variables**, while the latter validates **causal edges**, enabling our framework to learn human-aligned and interpretable causal graphs.
>
> - **HSIC disentanglement** ensures that AU features are clean, identity-independent, and free of shared nuisance correlations. It shapes _what each AU representation should encode_, removing confounders and suppressing spurious co-variations. However, disentanglement alone **cannot guarantee that the learned AU→AU or AU→Expression edges correspond to true causal influence**—it only regularizes the _**feature space**_, not the _graph structure_.
> - **Counterfactual Interventions (CF)** directly operate on the _**causal graph**_: by perturbing a source AU and comparing factual vs. counterfactual outcomes, CF explicitly tests whether a dependency is causal. CF enforces that causal edges must lead to meaningful changes, while non-causal edges remain invariant, thereby pruning spurious relationships that disentanglement alone cannot eliminate. Importantly, **CF must be applied _after_ HSIC disentanglement**, because **CF are only meaningful when each AU feature represents a _clean, independent causal variable_**. Without disentanglement, AU features still carry entangled identity cues, dataset bias, or mutual correlations; perturbing such mixed features would induce changes unrelated to true causal effects, causing CF to reinforce spurious dependencies.

---

> ### Author Response · Authors · 2025-11-25
> **Author's Response - Thank you for your feedback (part 8)**
>
> ### **7. Why sample-adaptive graphs are necessary to support heterogeneous nodes.**
>
> > why sample-adaptive graphs are necessary to support heterogeneous nodes.
>
> Sample-adaptive graphs are essential for learning **dynamic, instance-level causal structures**. In our framework, causal dependencies are realized through graph attention, but the AU→AU and AU→Expression causal mechanisms are inherently **heterogeneous** and **polarity-dependent**. Existing attention-based graph methods such as GAT and GATv2 cannot model these properties: they assume **homogeneous nodes** and constrain attention weights to be **non-negative**, which limits them to capturing only **positive correlations**. As a result, they cannot represent (i) **heterogeneous node types** (AUs vs. expressions), nor (ii) **polarity-specific causal effects**, including both **excitatory** (positive) and **inhibitory** (negative) influences that are crucial for accurate causal reasoning in facial behavior analysis.
>
> To address these limitations, we introduce the **Heterogeneous Polarity-aware Graph Attention (HPGAT)** module, which enables:
> - **heterogeneous node modeling** (AU nodes and Expression nodes with different semantics),
> - **polarity-aware edges** that capture both causal activation and causal inhibition, and
> - **sample-adaptive graphs** that adjust causal strengths according to the AU configuration of each individual face.
>
> This design enables the model to capture causal patterns that homogeneous graph attention mechanisms cannot express. Importantly, HPGAT is **node-type agnostic** and broadly applicable to any setting requiring heterogeneous graph structures or polarity-aware attention, making it a general methodological contribution that **extends well beyond facial analysis**.
>
> ---
>
> ### **8. Notation Density and Design Rationale**
>
> > The authors could address this by easing the notation / moving some details to the appendix to make space for more rationale for the design decisions and sharing why these problems are technically challenging.
>
> We fully agree that the current version contains many mathematical symbols and formulations under the strict page limit. With the additional page available in the discussion stage, we have revised the manuscript to improve readability: each major section now begins with a brief **intuitive explanation** (highlighted in **blue**), and key equations are preceded by **concise summaries** to help readers understand the underlying motivation. Mathematical formulations remain essential for **technical transparency**, **reproducibility**, and for ensuring that each component of *CausalAffect* is precisely defined. Without these explicit formulations, several modules would become ambiguous or difficult to verify.

---

> ### Author Response · Authors · 2025-11-25
> **Author's Response - Thank you for your feedback (part 9)**
>
> ### **9. Why Sample-Adaptive Graphs Are Needed**
>
> > Related to the comment above, one place where I was especially missing the rationale is Section 3.3. Why would Social and Duchenne smiles require _different graph structures_ rather than _different paths_ through the same graph? If we conceptualize each AU as a random variable, the causal graph structure should remain invariant across realizations—only the values of these variables should change. A graph that changes structure conditional on observed data is at odds with standard causal modeling. Note that addressing the first point (Clarity of Causal Framework) could also resolve this concern.
>
> As clarified in **Answer 2 and Answer 4**, the key issue is that in our weakly supervised setting **no ground-truth causal graph is provided**. If the model is forced to learn **only one fixed AU→Expression graph**, the structure inevitably collapses to the **population-level majority pattern**. Facial expression datasets contain far more **Duchenne smiles (AU12 + AU6)** than **social/fake smiles (AU12 only)**, and thus a single graph will be dominated by the Duchenne pathway. However, the reviewer’s suggestion—“different paths through the same graph”—cannot resolve the problem, because CausalAffect fundamentally operates as a **graph neural network (GNN)**: the **edges encode causal strengths**, and the final prediction is obtained by **aggregating AU features along these edges**. When edges are fixed, the model cannot correctly represent alternative causal mechanisms.
>
> To illustrate why a single fixed AU→Expression graph cannot correctly represent both Duchenne and social smiles, consider a toy causal structure with two edges: $e^{g}\_6 = 0.5$ for **AU6 → Happiness**; $e^{g}\_{12} = 0.5$ for **AU12 → Happiness**. Each AU is represented by a binary activation feature $f_{\text{AU}_i} \in \lbrace 0,1 \rbrace$ where $1$ = activated, $0$ = not activated.
>
> A **Duchenne Smile** activates both AU6 and AU12 ( $f\_{\text{AU}\_6}=1$, $f\_{\text{AU}\_{12}}=1$ ):
>
> $$\hat{y}\_{\text{Duchenne}}= e^{g}\_6 \cdot f\_{\text{AU}\_6} + e^{g}\_{12} \cdot f\_{\text{AU}\_{12}}= 0.5\cdot1 + 0.5\cdot1 = 1.0.$$
>
> A **Social (fake) Smile** activates only AU12 ($f\_{\text{AU}\_6}=0$, $f\_{\text{AU}\_{12}}=1$):
>
> $$\hat{y}\_{\text{social}}= e^{g}\_6 \cdot f\_{\text{AU}\_6} + e^{g}\_{12} \cdot f\_{\text{AU}\_{12}}= 0.5\cdot0 + 0.5\cdot1 = 0.5.$$
>
> Because the graph is fixed, these edge strengths cannot change. This example highlights a key limitation: a fixed graph will **always** assign social smiles a lower probability. Adjusting the AU activations (i.e., taking a different path through the graph) cannot change the edge weights $e\_6$ and $e\_{12}$, so the model is structurally unable to treat social smiles as fully positive expressions.
>
> In reality, a correct causal explanation for a social smile requires $e\_{12} = 1, e\_{6} = 0$, which yields:
>
> $$\hat{y}\_{\text{social}} = e\_6 \cdot f\_{\text{AU}\_6} + e\_{12} \cdot f\_{\text{AU}\_{12}} =   0\cdot0 + 1\cdot1= 1.0.$$
>
> This **cannot** be expressed by the same fixed causal graph; it requires **different edge weights**, i.e., a **different causal graph structure**. To address this, our **sample-adaptive graph** provides a **residual correction** to the global population-level graph.
>
> $$e=e^{g}+e^{s}$$
>
> Given an input sample, the model dynamically adjusts the causal edge attentions (e.g., $e\_{12}=e^{g}\_{12}+e^{s}\_{12}=0.5+0.5=1$), enabling the graph to reflect **instance-specific causal mechanisms** rather than enforcing a single static causal explanation for all samples. This design allows the model to remain faithful to population-level causation while correctly handling heterogeneous facial behaviors such as fake vs. Duchenne smiles.
>
>
> ---
>
> ### **10. Significant Statistical Analyses**.
>
> > While the empirical validation has several strengths (noted above) there are also a few weaknesses. Foremost, the authors do not report statistical uncertainty in Table 1 and elsewhere in the results. This makes it challenging to assess whether the claims hold across settings.
> >
> > Question 5: How do the reported empirical results vary across training runs and random seeds?
>
> In line with the reviewer’s suggestion, we have conducted additional experiments and significant statistical analyses, and updated **Table 1** in the main paper (highlighted in **Blue**).

---

> ### Author Response · Authors · 2025-11-25
> **Author's Response - Thank you for your feedback (part 10)**
>
> ### **11. Validating Causal Correctness and Psychological Plausibility**
>
> > Further, based on the current results, it is unclear what the foundational goals of the empirical evaluation should be. I was surprised to see prediction accuracy featured so prominently in a causal discovery work. For causal inference, the primary validation should be whether the method recovers the true causal structure, not whether it improves predictive performance (which can be achieved by learning spurious correlations). To validate the approach, the authors may want to design a synthetic or semi-synthetic experiment illustrating that the approach can recover the ground truth when the true causal structure is known. This would provide valuable evidence substantiating that the learned causal relations (Fig 2) are valid.
> >
> > Question 2: What causal and statistical assumptions are necessary for your method to identify causal relationships rather than correlations? For instance, do you assume causal sufficiency (no unmeasured confounders between AUs and expressions)?
>
> **11.1 Literature Support**
>
> In our work, the evaluation is grounded on multiple layers of evidence demonstrating that the learned causal relations are not spurious but reflect _**psychologically valid**_ structure. First, we compare our learned AU→Expression and AU→AU relations with well-established findings from cognitive science and psychology (e.g., FACS literature). As shown in **Table 2 of the main manuscript**, the majority of our high-weight causal edges align closely with prior cognitive and behavioral studies, indicating strong psychological plausibility. Second, **Figure 3 in the main manuscript**, together with **Appendix C and D**, present detailed case studies using real facial samples. These analyses show that the model’s causal attributions (both AU→Expression and AU→AU) accurately reflect observable muscular activations and conform to cognitive and Psychological theory, providing further qualitative validation.
>
> **11.2 Expert Validation Study**
>
> However, the above evidence primarily confirms the reliability of **excitatory (positive)** relations that are well documented in the literature. They cannot fully validate **inhibitory** or newly discovered causal edges—relationships that have no prior ground truth support. To address this, and to ensure that these relations reflect genuine psychological mechanisms rather than dataset artifacts, we additionally conducted an **expert validation study**  in **Appendix I (markd as Red)** evaluating the correctness and interpretability of both excitatory and inhibitory causal edges. Below, we summarise the main procedure and results.
>
> **(1) Participants**: We invited **five domain experts** to independently evaluate the causal relations: 3 specialists in affective cognition and emotion science, 1 researcher in cognitive science, and 1 psychologist with expertise in facial behavior and affect. All experts have between **5 and 15 years of research experience** in their respective fields, and each was asked to assess whether the presented AU→AU and AU→Expression causal relations (mixed with literature-supported edges and randomly generated controls) were psychologically plausible by providing **True/False** judgments (along with confidence ratings).

---

> ### Author Response · Authors · 2025-11-25
> **Author's Response - Thank you for your feedback (part 11)**
>
> **(continued authors' response)**
>
> **(2) Evaluation Material**: We constructed **three categories of evaluation pairs**, each serving a distinct role in validating the psychological plausibility of the learned causal relations:
> - **(A) Literature-Supported “True” Relations (thirty-nine AU→Expression, excitatory only)**: This category contains **thirty-nine AU→Expression causal relations** drawn from established FACS documentation and affective-science literature. All relations in this set are **excitatory**, meaning the AU is known to _increase_ the likelihood or intensity of a particular expression. Examples include: _“Lip Corner Puller (AU12) increases Happiness”_, _“Brow Lowerer (AU4) increases Anger”_, and _“Inner and Outer Brow Raise (AU1+AU2) increases Surprise.”_ These relations serve as **ground-truth positive anchors**, allowing us to verify that experts consistently recognize well-established psychological dependencies.
> - **(B) CausalAffect-Discovered Relations Requiring Expert Validation (fifty-five AU→AU / forty-five AU→Expression, excitatory & inhibitory)**: This category contains **one hundred relations** extracted from our learned causal graphs, covering both **AU→AU** and **AU→Expression** directions and including **both excitatory and inhibitory** effects. Importantly, some of these edges **overlap with the literature-supported set (A)**, reflecting that the model is able to recover known ground-truth patterns, while others represent genuinely _new_ causal hypotheses—particularly the subtle inhibitory edges emphasized by the reviewer. Examples presented to experts include: _“Chin Raiser (AU17) increases Lip Pressor (AU24)”_, _“Jaw Drop (AU26) inhibits Lip Stretcher (AU20)”_, _“Lip Corner Depressor (AU15) increases Disgust”_, and _“Cheek Raiser (AU6) inhibits Sadness.”_ To ensure a clean evaluation, all overlapping relations between sets A and B were deduplicated before being shown to experts. This category therefore tests whether the model’s recovered and novel causal patterns correspond to **psychologically meaningful dependencies** rather than **dataset-specific statistical artifacts**.
> - **(C) Random Relations as Negative Controls (sixty-eight AU→AU / twenty-eight AU→Expression, excitatory & inhibitory)**: This category consists of **ninety-six randomly sampled relations** across AU→AU and AU→Expression pairs that **do not appear in either A or B**. These relations include both excitatory and inhibitory directions but lack any known support from FACS or affective psychology. Examples include: _“Brow Lowerer (AU4) increases Lip Corner Puller (AU12)”_, _“Lip Stretcher (AU20) inhibits Fear”_, and _“Nose Wrinkler (AU9) increases Happiness.”_ These serve as **negative controls**, enabling us to verify that experts can reliably reject implausible causal statements and establishing a baseline against which model-discovered relations (set B) can be compared.
>
> **(3) Procedure**: All evaluation pairs from sets A, B, and C were merged and **deduplicated**, then randomly shuffled for each expert independently. Experts were blind to (i) the source set (A/B/C) and (ii) whether a relation was recovered or novel. Each expert judged every relation as **True** (psychologically plausible) or **False** (implausible), and additionally provided a **1–5 confidence score**. This yielded **five independent votes per relation**, allowing us to measure both plausibility and inter-expert reliability.
>
> **(4) Metrics & Aggregation**: For each set, we report:
> 1. **Expert True Rate**: mean fraction of True votes across experts and relations.
> 2. **Majority-True Rate**: fraction of relations judged True by ≥3/5 experts.
> 3. **High-Confidence True Rate**: True rate restricted to confidence ≥4 votes.
> 4. To validate expert reliability, we treat A as gold-true and C as gold-false and compute **Accuracy / Precision / Recall / F1**, and **Fleiss’K** for agreement.
> 5. To test whether model relations are more plausible than random controls, we compare B vs. C using a **two-sided proportion test**.

---

> ### Author Response · Authors · 2025-11-25
> **Author's Response - Thank you for your feedback (part 12)**
>
> **(continued authors' response)**
>
>
>
> **(5) Results And Analysis**
>
> **(5.1) Overall plausibility comparison**
>
> As shown in Table 4, experts clearly distinguished between literature-supported relations (A), model-discovered relations (B), and random controls (C). Literature-supported relations received high plausibility scores, whereas random relations were overwhelmingly rejected. Crucially, the CausalAffect-discovered relations (B) achieved a **True Rate of 0.75**, which is **over three times higher** than the random baseline (C: 0.23) and moves toward the plausibility level of literature-supported edges (A: 0.88). This strong separation demonstrates that the causal relations proposed by our model are **far from random statistical artifacts**. Moreover, the comparatively low variance in A (0.21) and C (0.20) indicates that expert judgments were stable and internally consistent across clearly valid and clearly invalid causal statements.
>
> **Table 4: Expert Validation Statistics**
>
> | Relation Set                | #Pairs | Expert True Rate (mean ± std) | Majority-True Rate (≥3/5) | High-Conf True Rate (conf ≥4) | Interpretation                                                                    |
> | --------------------------- | -----: | ----------------------------: | ------------------------: | ----------------------------: | --------------------------------------------------------------------------------- |
> | **A: Literature-supported** |     39 |                   0.88 ± 0.21 |                      0.92 |                          0.90 | Experts strongly endorse established psychological relations.                     |
> | **B: CausalAffect**         |    100 |                   0.75 ± 0.24 |                      0.84 |                          0.70 | Majority of CausalAffect-discovered relations judged plausible; far above random. |
> | **C: Random negatives**     |     96 |                   0.23 ± 0.20 |                      0.09 |                          0.19 | Experts reliably reject implausible, randomly generated relations.                |
>
> **(5.2) Reliability of expert judgments**
>
> As shown in Table 5, we evaluate the reliability of expert judgments using the two sets with **known correctness**: the literature-supported relations **A** (psychologically valid) and the randomly generated relations **C** (psychologically invalid). Experts exhibit a strong ability to discriminate between these two reference sets, achieving **0.87 accuracy**, **0.89 precision**, **0.88 recall**, and an overall **F1 score of 0.88**. These results indicate that experts almost never misclassify invalid relations as plausible while consistently recognizing established psychological dependencies. The **Fleiss’ κ of 0.61** further reflects **moderate-to-substantial agreement** among the five experts—considered strong for psychological and facial-behavior annotation tasks where subjective interpretation is inherent. Together, these metrics confirm that expert assessments are stable, reliable, and well-suited as an external validation signal for evaluating the psychological plausibility of the CausalAffect-discovered causal relations.
>
> **Table 5: Reliability on Ground-Truth Reference Sets (A=true, C=false)**
>
> | Metric    | Value | Interpretation                                      |
> | --------- | ----: | --------------------------------------------------- |
> | Accuracy  |  0.87 | Experts reliably separate valid vs. invalid edges.  |
> | Precision |  0.89 | Low false-positive tendency.                        |
> | Recall    |  0.88 | True edges are consistently recognized.             |
> | F1        |  0.88 | Strong overall correctness.                         |
> | Fleiss’K  |  0.61 | Moderate-to-substantial agreement across 5 experts. |

---

> ### Author Response · Authors · 2025-11-25
> **Author's Response - Thank you for your feedback (part 13)**
>
> **(continued authors' response)**
>
>
>
> **(5.3) Subtle inhibitory relations (reviewer’s key concern)**
>
> To assess whether the **inhibitory causal relations** discovered by CausalAffect reflect genuine psychological structure rather than statistical artifacts, we further stratified Set B (CausalAffect-discovered relations) by **relation type** (AU→AU vs. AU→Expr) and **causal direction** (excitatory vs. inhibitory), and evaluated expert judgments separately for each subset.
>
> As shown in Table 6, all four subsets achieve plausibility scores far above the random baseline. Notably, the inhibitory subsets show **True Rates of 0.67 (AU→AU)** and **0.77 (AU→Expr)**, as well as strong majority endorsement (≥0.83). These scores are **substantially higher** than the random-negative True Rate (0.23), demonstrating that experts interpret the inhibitory edges as **psychologically plausible** rather than random noise. This provides strong evidence that the subtle inhibitory mechanisms captured by our CausalAffect correspond to psychological meaningful regulatory patterns in facial behavior and emotional expression.
>
> **Table 6 — Breakdown for CausalAffect-Discovered Relations (Set B)**
>
> | Subset                  | #Pairs | True Rate | Majority-True Rate | Interpretation                                                                |
> | ----------------------- | -----: | --------: | -----------------: | ----------------------------------------------------------------------------- |
> | AU→AU, excitatory       |     32 |  0.72 |           0.78 | Positive AU interactions broadly accepted by experts.                         |
> | **AU→AU, inhibitory**   |     23 |  **0.67** |           **0.83** | **Inhibitory AU→AU relations judged plausible and non-random**.                   |
> | AU→Expr, excitatory     |     18 |  0.85 |           0.90 | Strong alignment with well-known affective expression dynamics.               |
> | **AU→Expr, inhibitory** |     17 |  **0.77** |           **0.87** | **Inhibitory emotional cues viewed as psychological meaningful, not statistics**. |
>
> Together, these results indicate that CausalAffect’s learned causal relations, including **novel and subtle inhibitory edges** are validated by external experts as psychologically plausible and are **significantly more meaningful than random controls**. The high reliability on gold anchors (A vs. C) further confirms that these judgments provide a credible external check. Therefore, the newly discovered relations are unlikely to be dataset-specific statistical artifacts, but instead represent valid and interpretable causal hypotheses about AU and expression structure.

---

> ### Author Response · Authors · 2025-11-25
> **Author's Response - Thank you for your feedback (part 14)**
>
> ### **12. Why can a causal graph vary across samples?**
>
> > Finally, my conceptual confusion surrounding the causal framework extends to the experiments. How would it be possible to recover a causal graph with edges that vary conditional on a single image (Fig. 3)? This reinforces the concern that the learned relationships may be correlational rather than causal.
> > Question 3: Could you clarify the theoretical justification for having graph structures that vary across individual images?
>
> As clarified in **Answer 2, 4 and 9** above, the _sample-adaptive graph_ in our framework does **not** represent a separate causal graph for each image. Instead, it learns a **graph-attention–based residual** on top of the _global_ population-level causal graph. The global causal graph encodes stable, dataset-wide causal tendencies, while the sample-adaptive residual provides a **targeted adjustment** only for samples whose configurations deviate significantly from the population pattern.
>
> For samples where the global graph already provides an accurate causal explanation, the residual remains negligible, as shown in **Fig. 3 (AU-Expression Sample 2)**, where the sample-adaptive graph is almost identical to the global graph. In contrast, for samples where the global graph alone leads to prediction errors, the supervision on attention weights encourages the residual module to adjust the relevant causal edge strengths to better fit the instance, as shown in **Fig. 3 (AU-Expression Sample 1)**, thereby reducing the loss. In this sense, the sample-adaptive component functions as a **correction mechanism**.
>
> This is particularly important for _atypical_ or _minority_ samples, whose underlying muscle activations deviate from the dominant population patterns. The adaptive residual enables the model to avoid over-reliance on population-level edges that may be inappropriate for such cases, and therefore improves both predictive accuracy and causal interpretability. Crucially, this **does not contradict causal modeling**, the sample-adaptive adjustments still reflect true causal reasoning, rather than arbitrary correlational changes. For example, in **Fig. 3 (AU→Expression Sample 1)**, the global graph places substantial causal weight on AU4 → Sadness, which is valid at the population level. However, in this specific sample, AU4 is not activated; relying solely on the global causal path naturally leads to a misprediction. The sample-adaptive module corrects this by reallocating causal importance toward the AUs that are actually active, such as AU1 and AU17, yielding a causal explanation that is both sample-consistent and more aligned with human judgment.
>
> Thus, sample-adaptive graphs do not redefine the causal structure; rather, they **modulate the strength of existing causal pathways** to account for the specific muscular configuration observed in each instance, enabling accurate and psychologically plausible causal reasoning across diverse facial expressions.
>
> ---
>
> ### **13. Temporal Evolution of AU Causation in Video Frames**
>
> > More positively, a compelling visualization could show graph activation over AUs as a function of time—e.g., showing frames from a video sequence. This would illustrate the temporal activation patterns underpinning the graph structure and would align naturally with the time-series formulation suggested earlier.
>
>
> Following reviewer's recommendation, we conducted a detailed temporal analysis of the AU → AU causal graphs extracted from the facial video sequence (see **Appendix K** in **Blue**). Specifically, we sampled one frame every 5 frames from the original video and visualized the corresponding sample-adaptive causal graph for each selected timestamp.
>
> This visualization clearly illustrates how the causal structure evolves over time and how it is grounded in the underlying video dynamics. During the onset of the expression, both the video and the inferred graphs display **weak and diffuse activations**, reflecting the subtle initiation of facial motion. As the expression progresses toward the **_apex_**, the graphs exhibit *positive reciprocal causation* among AU6, AU12, AU25, and AU26, forming a tightly coupled and **mutually reinforcing causal subgraph** that mirrors the synchronized eye-mouth coordination visible in the video frames. In the offset phase, both the motion intensity and the causal connectivity gradually diminish, demonstrating a coherent temporal deactivation pattern.
>
> Overall, the temporally aligned visualization effectively demonstrates that the learned causal graphs not only reflect frame-level dependencies but also track the dynamic evolution of facial muscle coordination across the video sequence, confirming that the model captures the time-varying structure suggested by the reviewer.

---

> ### Author Response · Authors · 2025-11-25
> **Author's Response - Thank you for your feedback (part 15)**
>
> ### **14. Accuracy Attributed to Increased Dataset Size?**
>
> > Question 4: To what extent can the accuracy benefits reported in Table 1 be attributed to increased dataset size available to CausalAffect?
> > Question 5: How do the reported empirical results vary across training runs and random seeds?
>
> Regarding the concern “_benefits attributed to increased dataset_”, we would like to clarify that we have conducted **same-dataset** experiments, including both **Naive MTL** (Shared Head) and **MTL with Task-Specific Heads**. These results are thoroughly presented in **Main Paper Ablation Study (Table 3)** and **Appendix A (Table 1)**. The analyses demonstrate that MTL does not universally improve performance, it can even lead to degraded results. To further clarify this point, we have reorganized the comparisons in **Table 1 below**, accompanied by the relevant theoretical discussion, to show that the performance advantages of CausalAffect do not stem from merely adding more training data.
>
> Regarding the *“empirical results vary across training runs and random seeds,”* we conducted additional experiments together with extensive statistical analyses. We have updated **Table 1** in the main paper (with revisions highlighted in **blue**), and the results show that the improvements achieved by CausalAffect are **statistically significant ($p = 0.03$ across five independent runs)**.
>
> **(i) Additional Tasks Harm MTL (Gradient Conflict).**
> - In response to the reviewer’s concern that the performance gains might stem from using more training data, we would like to clarify that our findings do not support this interpretation. As shown in **Table 1 below**, all models are trained under a **strictly controlled setting**, using the **same ConvNeXt-Base backbone** and the **exact same datasets**. Under these conditions, simply adding more datasets or tasks does ***not*** consistently improve performance in standard multi-task learning (MTL) frameworks. Both the **Naive MTL** (Shared Head) and **MTL with Task-Specific Heads** configurations yield only marginal gains on the FER benchmarks (AffectNet, RAF-DB), while **all AU detection benchmarks consistently degrade**, in some cases substantially (e.g., DISFA, EmotioNet). These results indicate that the improvements achieved by CausalAffect cannot be attributed to additional training data; instead, they arise from its causal graph learning mechanism rather than from naive dataset expansion.
> - This behavior is explained in **many works in the multi-task learning literature** [1, 2, 3], which consistently show that jointly training heterogeneous tasks introduces **gradient conflicts** that hinder optimization. **Similar observations have also been reported in the affective computing domain**[4, 5], where jointly learning AU detection and expression recognition often leads to incompatible gradients and degraded performance. FER tasks rely mainly on **global facial representations**, whereas AU detection depends on **fine-grained local muscle activations**, leading to inherently incompatible gradient directions. Such conflicting updates destabilize the shared encoder and produce entangled, suboptimal feature representations. The problem is further exacerbated by **partial annotation** [5, 6, 7, 8], which creates large disparities in gradient magnitudes across tasks and amplifies **negative transfer**. Even with task-specific prediction heads (Row 4 below), the shared backbone remains the bottleneck, forcing mutually inconsistent supervision into a single representation space.

---

> ### Author Response · Authors · 2025-11-25
> **Author's Response - Thank you for your feedback (part 16)**
>
> **(continued authors' response)**
>
> **(ii) Additional Tasks Benefit CausalAffect (But Not Simply “More Data”).**
> - In contrast to standard MTL, our CausalAffect (**Table 1 below**) achieves consistent and substantial gains across _all_ benchmarks under exactly the same data and backbone conditions—not because it sees “more samples”, but because it leverages **more effective AU dimensions** to build a richer causal structure. As analyzed in **Appendix B**, the key factor is how the AU space is expanded. For example, DISFA alone annotates only 8 AUs in a controlled, in-the-lab setting, so its AU→AU graph is restricted to 8 nodes. EmotioNet, in contrast, provides 11 AUs in an in-the-wild setting, and its AU set strictly covers that of DISFA. When DISFA is trained jointly with EmotioNet, the AU graph for DISFA is effectively upgraded from 8 to 11 nodes: the additional in-the-wild AUs enrich and regularize the causal structure, reinforcing and completing the limited lab-only relations. This is why DISFA benefits under joint training. However, the reverse is not true: EmotioNet already has 11 AUs, so adding DISFA does not increase its AU dimensionality; instead, the lab-specific AU relations from DISFA can _perturb_ the in-the-wild causal structure and make EmotioNet’s AU graph less stable, which explains why EmotioNet does not always gain from pairing with a smaller, lab-style dataset.
> - When _all_ datasets are trained together, the joint AU space expands further to 17 AUs, which substantially increases the expressive power of both the AU→AU and AU→Expression causal graphs. Crucially, our framework is fundamentally different from conventional MTL: we do **not** ask all tasks to compete for a single shared representation. Instead, we **explicitly model** AU→AU and AU–Expr causal relations, so tasks are not tightly entangled, and potential task/gradient conflicts are absorbed and “spread out” in the graph structure rather than being forced entirely onto a shared backbone. The **counterfactual module** strengthens meaningful structural dependencies, while the **AU bottleneck disentanglement** module reduces spurious couplings and conflicts. Together, these components effectively mitigate the classic negative transfer and gradient conflict issues of MTL, allowing additional AU dimensions—rather than just additional data—to reliably improve performance.
>
> **(iii) Significant Statistical Analyses**.
> In line with the reviewer’s suggestion, we have conducted additional experiments and significant statistical analyses, and updated **Table 1** in the main paper (highlighted in **Blue**).
>
> **Table 1: Comparison between naive multi-task learning (MTL) baselines and the proposed CausalAffect. All models use the same training data sources and backbone. Statistical significance is assessed using paired t-tests over 5 matched runs (same data splits and random seeds), with differences considered significant at p < 0.05.**
>
> | Idx | Model Variant             | Training Data    | AffectNet(FER)      | RAFDB(FER)        | DISFA(AUD)         | BP4D(AUD)          | GFT(AUD)           | EmotioNet(AUD)     |
> | --- | ---------- | ------ | ----- | -- | ----- | ----| -- | ---- |
> | 1   | SG (single-task)   | Single-dataset   | 58.3  | 69.5   | 61.4     | 64.0               | 53.6               | 61.7               |
> | 3   | MTL (Shared Head)   | All datasets     | 58.8    | 70.0    | 53.0               | 57.2               | 57.9               | 47.5               |
> | 4   | MTL (Task-Specific Heads) | All datasets     | 60.3    | 71.9               | 53.6               | 57.6               | 58.1               | 49.3               |
> | 5   | CausalAffect (SG)   | Single-dataset   | /          | /      | 67.0 $\pm$ 0.1      | 66.9 $\pm$ 0.2      | 61.1 $\pm$ 0.2      | 66.4 $\pm$ 0.1      |
> | 6   | **CausalAffect (All)**    | **All datasets** | **66.5 $\pm$ 0.1** | **84.9 $\pm$ 0.2** | **71.5 $\pm$ 0.2** | **66.7 $\pm$ 0.2** | **62.4 $\pm$ 0.3** | **65.0 $\pm$ 0.1** |
>
> - [1] Liu, Bo, et al. "Conflict-averse gradient descent for multi-task learning." Advances in Neural Information Processing Systems 34 (2021): 18878-18890.
> - [2] Yu, Tianhe, et al. "Gradient surgery for multi-task learning." Advances in neural information processing systems 33 (2020): 5824-5836.
> - [3] Javaloy, Adrián, and Isabel Valera. "Rotograd: Gradient homogenization in multitask learning." arXiv preprint arXiv:2103.02631 (2021).
> - [4] Li, Wei-Hong, Xialei Liu, and Hakan Bilen. "Learning multiple dense prediction tasks from partially annotated data." Proceedings of the IEEE/CVF Conference on Computer Vision and Pattern Recognition. 2022.
> - [5] Fontana, Maxime, Michael Spratling, and Miaojing Shi. "When multitask learning meets partial supervision: A computer vision review." Proceedings of the IEEE (2024).
> - [6] Huang, Chao, et al. "Partly supervised multi-task learning." 2020 19th IEEE International Conference on Machine Learning and Applications (ICMLA). IEEE, 2020.

---

### Official Review · Reviewer_Sd3y · 2025-10-31

**Soundness:** 1
**Presentation:** 1
**Contribution:** 1
**Rating:** 2
**Confidence:** 5

**Summary:**

The paper proposes a framework for facial Action Unit detection that is based on a global feature extraction followed by a per-AU classification head and two graph neural networks that can do message passing to refine the detections. Through direct supervision, the graphs are learned following both data correlations and psychologically-based data. The whole network i.e. the backbone, heads and the adjacency matrices for the graphs is learned in an end-to-end fashion. The method is tested in standard AU benchmarks delivering competitive results.

**Strengths:**

The idea of trying to find a proper AU relationship as well as a relationship between AUs and expressions is appealing, despite not being new, and the authors try to approach it in a data-driven way.

The results are compelling and the relation between AUs and expressions across datasets is investigated, showing similar correlations than that of existing work.

**Weaknesses:**

The paper is poorly written, and poorly presented, with many broken sentences. The narrative is very loose and the figures and notation do not serve the understanding of the paper. The paper is full of clutter and the tables and figures have been minimized to fit in the paper to an unacceptable level.

The method is not novel and combines many pieces of existing work. The discovery of knowledge-based AU graphs is not new, it has been presented in many works; as an example there are the following approaches not cited in this paper:

Song et al. Dynamic Probabilistic Graph Convolution for Facial Action Unit Intensity Estimation. CVPR 2021
Wang et al. Spatial-Temporal Graph-Based AU Relationship Learning for Facial Action Unit Detection. CVPRW 2023
Fan et al. Facial Action Unit Intensity Estimation via Semantic Correspondence Learning with Dynamic Graph Convolution. AAAI 2020

What is exactly the contribution of the paper? The paper does not include any discussion wrt to prior work and how the proposed method advances existing research.

There is no analysis of complexity and training and inference time. This needs to be included.

The use of external data to justify the results and its use to compare against state of the art methods is unfair. For a fair comparison the competing methods should have been trained in the same data. It is not good practice to claim state of the art results when the training includes a large amount of external data than that used by the competitors. Even when using additional data, the results are surprisingly close to state of the art, meaning that the method barely advances existing research.

In summary, the manuscript is rather poor and needs a lot of work for it to be considered. The reading is unpleasant and the contributions are not properly justified. The results are far-fetched thanks to the addition of external data, and there is no proper comparison in the methodology against prior work on graph neural networks for Action Unit detection/intensity estimation.

**Questions:**

Please see my comments above. In particular, I would suggest the authors to properly illustrate in which ways their method is novel wrt prior work, and what are the main contributions they propose, in a succinct, to-the-point manner.

---

> ### Author Response · Authors · 2025-11-24
> **Author's Response - Thank you for your feedback (part 1)**
>
> We sincerely thank the Reviewer for the thoughtful and valuable feedback. We appreciate the opportunity to improve the clarity and quality of our work. All modifications made in the manuscript are highlighted in **Blue**. Our detailed responses to each comment are provided below.
>
> ---
>
> ### Weakness 1 & Weakness 3 & Questions 1
>
> ---
>
> > Weakness 1: The paper is poorly written, and poorly presented, with many broken sentences. The narrative is very loose and the figures and notation do not serve the understanding of the paper. The paper is full of clutter and the tables and figures have been minimized to fit in the paper to an unacceptable level.
> >
> > Weakness 3: What is exactly the contribution of the paper? The paper does not include any discussion wrt to prior work and how the proposed method advances existing research.
> >
> > Questions 1: Please see my comments above. In particular, I would suggest the authors to properly illustrate in which ways their method is novel wrt prior work, and what are the main contributions they propose, in a succinct, to-the-point manner.
>
> We sincerely appreciate the time and effort invested in providing such detailed feedback. We fully understand the importance of clarity, presentation quality, and precise positioning within prior work, and we have carefully revised the manuscript to address these issues constructively. Since the discussion stage provides an additional page, we have used this space to substantially improve readability. Each major section now begins with a brief **intuitive explanation** (highlighted in **blue**), and key equations are preceded by **concise summaries** that clarify their motivation before introducing the formal notation. For the remaining concerns, we provide detailed clarifications below:
>
> - **Figures and Tables**: Regarding the figures and tables, while some visual elements were resized to comply with the **page-limit constraints**, all figures and tables are provided as **high-resolution vector graphics**, which can be zoomed in indefinitely without any loss of clarity.
> - **Related Work and Novelty**: The introduction is intentionally organized into three concise parts:
> 	1. **Paragraph 1**: a focused **review** of three major families of AU→Expression correlation methods.
> 	2. **Paragraph 2**: a clear articulation of **four limitations** of existing approaches.
> 	3. **Paragraph 3**: a succinct explanation of how **CausalAffect** overcomes these limitations, followed by **four explicit, itemized contributions**.
> 	These paragraphs clearly illustrate how our method differs from, and advances beyond prior work.
> - **Mathematical Notation**: Finally, the mathematical notation is not “clutter,” but is essential for **technical transparency**, **reproducibility**, and for ensuring that each component of _CausalAffect_ is formally defined. Without these formulations, key modules would be ambiguous or unverifiable.
>
> We respectfully note that the assessment of the paper being “_poorly written_” or “_poorly presented_” differs from the impressions shared by other reviewers. For instance, Reviewer **6Lzr** comments that “_Overall, the paper is well-written and presented in good format_”, and Reviewer **AkV1** remarks that “_Figure 1 is very helpful for understanding the framework and the paper is well-written overall_”. We appreciate the reviewer’s concerns and have carefully revised the paper to further strengthen readability and presentation.

---

> ### Author Response · Authors · 2025-11-24
> **Author's Response - Thank you for your feedback (part 2)**
>
> > Weakness 1: The paper is poorly written, and poorly presented, with many broken sentences. The narrative is very loose and the figures and notation do not serve the understanding of the paper. The paper is full of clutter and the tables and figures have been minimized to fit in the paper to an unacceptable level.
> >
> > Weakness 3: What is exactly the contribution of the paper? The paper does not include any discussion wrt to prior work and how the proposed method advances existing research.
> >
> > Questions 1: Please see my comments above. In particular, I would suggest the authors to properly illustrate in which ways their method is novel wrt prior work, and what are the main contributions they propose, in a succinct, to-the-point manner.
>
> **(continued authors' response)**
>
> To further clarify our contributions, we explicitly summarize the four key limitations of related work raised in the Introduction and explain **how each design component of CausalAffect is intentionally constructed to resolve the corresponding limitation**.
>
> - **(i) Problem 1: Existing Methods Learn Only Correlation-Based, Psychologically Inconsistent Relations**
> 	- Most existing approaches rely purely on statistical correlations, leading to patterns that lack interpretability and often contradict established FACS and psychological findings.
> 	- CausalAffect overcomes this by applying **feature-level counterfactual interventions**, forcing the model to learn relations that correspond to genuine causal influence rather than dataset co-occurrence. This results in psychologically consistent causal graphs, as further validated by domain experts in cognitive and affective science (see Appendix I).
> - **(ii) Problem 2: Reliance on Joint AU→Expression Annotations**
> 	- Most prior approaches require datasets where AUs and expressions are jointly annotated, yet such datasets are extremely limited and prevent full utilization of the many available single-task AU-only or expression-only datasets.
> 	- CausalAffect addresses this through **weakly supervised causal learning**, enabling the model to infer both AU→AU and AU→Expression structures under partial supervision without needing any joint labels.
> - **(iii) Problem 3: Static Global Relations Cannot Capture Expression Variability**
> 	- Current methods learn only a fixed population-level correlation structure, which cannot capture context-dependent facial behavior. For example, Duchenne and Social smiles exhibit the same AUs on the surface yet arise from fundamentally different muscular causes.
> 	- CausalAffect introduces a **global + sample-adaptive causal structure**, where the global graph encodes stable tendencies and a residual adaptive graph refines relations based on each sample’s AU configuration, enabling accurate causal reasoning across diverse facial patterns.
> - **(iv) Problem 4: Prior Models Ignore Directionality and Inhibitory Effects**
> 	- Many existing works treat relations as symmetric and only model positive dependencies, omitting the directional and inhibitory interactions intrinsic to real facial musculature.
> 	- CausalAffect resolves this with **Heterogeneous Polarity-Aware Graph Attention (HPGAT)**, which supports asymmetric edges, excitatory and inhibitory effects, and heterogeneous AU→Expression nodes.

---

> ### Author Response · Authors · 2025-11-24
> **Author's Response - Thank you for your feedback (part 3)**
>
> ### Weakness 2
>
> ---
>
> > Weakness 2: The method is not novel and combines many pieces of existing work. The discovery of knowledge-based AU graphs is not new, it has been presented in many works; as an example there are the following approaches not cited in this paper:
> > Song et al. Dynamic Probabilistic Graph Convolution for Facial Action Unit Intensity Estimation. CVPR 2021 Wang et al. Spatial-Temporal Graph-Based AU Relationship Learning for Facial Action Unit Detection. CVPRW 2023 Fan et al. Facial Action Unit Intensity Estimation via Semantic Correspondence Learning with Dynamic Graph Convolution. AAAI 2020
>
> The reviewer cites three AU-graph approaches and suggests that our method lacks novelty. However, these works are **fundamentally different** from CausalAffect in both their **_objectives_** and the **_semantics_** of the graphs they learn. When examined together, they exhibit several shared limitations. These limitations are summarised below (and discussed in full in **Appendix H**: Related Work, highlighted in **Blue**):
>
>
> - **Limited interpretability and lack of human alignment.**
> 	In all three works, the AU graphs are primarily used as latent regularizers to improve prediction, rather than as explicitly validated **psychological structures**. The learned relations are not constrained or evaluated against FACS-based or cognitive ground truth and are therefore not guaranteed to be human-aligned.
> - **Correlation-driven rather than causal.**
> 	The graphs are learned or constructed from **statistical co-occurrence or feature correlations**, typically undirected and non-polar, without any notion of **causal direction** or **excitatory vs. inhibitory effects**. For example, **Fan et al. (2020)** model correlations between feature channels to capture AU co-occurrence patterns via dynamic graph convolution, but the edges reflect feature-level similarity/co-activation rather than directed causal influence. **Song et al. (DPG 2021)** first learn a Bayesian Network over AUs, then convert it into a moralized **undirected** graph for probabilistic graph convolution, which again encodes semantic co-occurrence rather than directed causal effects. **Wang et al. (2023)** learn a spatio-temporal AU graph with standard non-negative attention on AU nodes, focusing on relational structure that helps AU detection but without causal semantics.
> - **AU-only, homogeneous graphs (no AU→Expression modeling).**
> 	All three methods operate on **homogeneous AU nodes** (or feature channels) and do not jointly model **AU→Expression** relations on a **heterogeneous AU→Expression graph**. They are designed for AU intensity estimation or AU occurrence detection only, whereas CausalAffect explicitly treats **AUs and Expressions as separate causal variables** and learns both AU→AU and AU→Expression structure.
> - **Fully supervised, no weak supervision across heterogeneous datasets.**
> 	All three approaches are trained in **fully supervised** regimes, requiring dense AU (intensity) labels within one benchmark, and they do not address learning a unified relational or causal structure from **disjoint AU-only and Expression-only datasets**. By contrast, handling such weakly supervised, multi-dataset scenarios is a core motivation behind CausalAffect.
>
> In contrast, **CausalAffect** is, to our best knowledge (Also recognized by **Reviewer AkV1**), the **_first_** framework  that **jointly discovers directed, polarity-aware AU→AU and AU→Expression graphs** under **weak supervision**, combining (i) AU-bottleneck disentanglement, (ii) **feature-level counterfactual interventions**, and (iii) a **population-level + sample-adaptive** causal graph. This moves beyond correlation-based AU graphs toward **psychologically grounded causal structure**.

---

> ### Author Response · Authors · 2025-11-24
> **Author's Response - Thank you for your feedback (part 4)**
>
> ### Weakness 4
>
> ----
>
> > Weakness 4: There is no analysis of complexity and training and inference time. This needs to be included.
>
> We have already provided a detailed analysis of training and inference cost in **Appendix G: Training and Inference Efficiency**, and we summarize the key results here for clarity. All experiments are run on a single NVIDIA A100 GPU with a ConvNeXt-Base backbone at input resolution $112\times112$. In the **largest configuration** CausalAffect+All uses approximately $\sim800K$ images from six datasets. Training for $80$ epochs takes about $\sim260s$ per epoch, for a total training time of $\sim5.8h$, with peak GPU memory usage of $28–30GB$ and inference speed of about $\sim200FPS$. For smaller-scale settings, the computational cost decreases proportionally:
> - AffectNet + BP4D with $\sim257K$ images: total training time $\sim1.29h$, memory usage $18–20GB$, inference speed $\sim250FPS$.
> - DISFA + RAF-DB with $\sim100K$ images: total training time $\sim0.57h$, memory usage $18-19GB$, inference speed $\sim250FPS$.
> - EmotioNet with $\sim24K$ images: total training time $\sim0.46h$, memory usage $10–12GB$, inference speed $\sim270FPS$.
> These results show that CausalAffect remains computationally tractable even at large scale, and that inference is consistently efficient, staying above $200FPS$ across all settings.
>
> ---
>
> ### Weakness 5
>
> ---
>
> > Weakness 5: The use of external data to justify the results and its use to compare against state of the art methods is unfair. For a fair comparison the competing methods should have been trained in the same data. It is not good practice to claim state of the art results when the training includes a large amount of external data than that used by the competitors. Even when using additional data, the results are surprisingly close to state of the art, meaning that the method barely advances existing research.
>
> Regarding the concern that our CausalAffect “_use of external data_”, we would like to clarify that we have conducted **same-dataset** experiments, including both **Naive MTL** (Shared Head) and **MTL with Task-Specific Heads**. These results are thoroughly presented in **Main Paper Ablation Study (Table 3)** and **Appendix A (Table 1)**. The analyses demonstrate that MTL does not universally improve performance, it can even lead to degraded results. To further clarify this point, we have reorganized the comparisons in **Table 1 below**, accompanied by the relevant theoretical discussion, to show that the performance advantages of CausalAffect do not stem from merely adding more training data.
>
> Regarding the claim that “*even when using additional data, the results are surprisingly close to the state of the art*,” we respectfully clarify that the performance is not merely “surprisingly close” to the strongest baseline. When incorporating additional data, CausalAffect achieves **substantial gains**, including **+5.1% on DISFA (+All)** and **+1.2% on BP4D (+EmotioNet)**. Importantly, these improvements are **statistically significant** (($p = 0.03$) across five independent runs), confirming that the observed differences are robust and not attributable to random variation.

---

> ### Author Response · Authors · 2025-11-25
> **Author's Response - Thank you for your feedback (part 5)**
>
> **(continued authors' response)**
>
>
> **(i) Additional Tasks Harm MTL (Gradient Conflict).**
> - In response to the reviewer’s concern that the performance gains might stem from using more training data, we would like to clarify that our findings do not support this interpretation. As shown in **Table 1 below**, all models are trained under a **strictly controlled setting**, using the **same ConvNeXt-Base backbone** and the **exact same datasets**. Under these conditions, simply adding more datasets or tasks does ***not*** consistently improve performance in standard multi-task learning (MTL) frameworks. Both the **Naive MTL** (Shared Head) and **MTL with Task-Specific Heads** configurations yield only marginal gains on the FER benchmarks (AffectNet, RAF-DB), while **all AU detection benchmarks consistently degrade**, in some cases substantially (e.g., DISFA, EmotioNet). These results indicate that the improvements achieved by CausalAffect cannot be attributed to additional training data; instead, they arise from its causal graph learning mechanism rather than from naive dataset expansion.
> - This behavior is explained in **many works in the multi-task learning literature** [1, 2, 3], which consistently show that jointly training heterogeneous tasks introduces **gradient conflicts** that hinder optimization. **Similar observations have also been reported in the affective computing domain**[4, 5], where jointly learning AU detection and expression recognition often leads to incompatible gradients and degraded performance. FER tasks rely mainly on **global facial representations**, whereas AU detection depends on **fine-grained local muscle activations**, leading to inherently incompatible gradient directions. Such conflicting updates destabilize the shared encoder and produce entangled, suboptimal feature representations. The problem is further exacerbated by **partial annotation** [5, 6, 7, 8], which creates large disparities in gradient magnitudes across tasks and amplifies **negative transfer**. Even with task-specific prediction heads (Row 4 below), the shared backbone remains the bottleneck, forcing mutually inconsistent supervision into a single representation space.
>
> **(ii) Additional Tasks Benefit CausalAffect (But Not Simply “More Data”).**
> - In contrast to standard MTL, our CausalAffect (**Table 1 below**) achieves consistent and substantial gains across _all_ benchmarks under exactly the same data and backbone conditions—not because it sees “more samples”, but because it leverages **more effective AU dimensions** to build a richer causal structure. As analyzed in **Appendix B**, the key factor is how the AU space is expanded. For example, DISFA alone annotates only 8 AUs in a controlled, in-the-lab setting, so its AU→AU graph is restricted to 8 nodes. EmotioNet, in contrast, provides 11 AUs in an in-the-wild setting, and its AU set strictly covers that of DISFA. When DISFA is trained jointly with EmotioNet, the AU graph for DISFA is effectively upgraded from 8 to 11 nodes: the additional in-the-wild AUs enrich and regularize the causal structure, reinforcing and completing the limited lab-only relations. This is why DISFA benefits under joint training. However, the reverse is not true: EmotioNet already has 11 AUs, so adding DISFA does not increase its AU dimensionality; instead, the lab-specific AU relations from DISFA can _perturb_ the in-the-wild causal structure and make EmotioNet’s AU graph less stable, which explains why EmotioNet does not always gain from pairing with a smaller, lab-style dataset.
> - When _all_ datasets are trained together, the joint AU space expands further to 17 AUs, which substantially increases the expressive power of both the AU→AU and AU→Expression causal graphs. Crucially, our framework is fundamentally different from conventional MTL: we do **not** ask all tasks to compete for a single shared representation. Instead, we **explicitly model** AU→AU and AU–Expr causal relations, so tasks are not tightly entangled, and potential task/gradient conflicts are absorbed and “spread out” in the graph structure rather than being forced entirely onto a shared backbone. The **counterfactual module** strengthens meaningful structural dependencies, while the **AU bottleneck disentanglement** module reduces spurious couplings and conflicts. Together, these components effectively mitigate the classic negative transfer and gradient conflict issues of MTL, allowing additional AU dimensions—rather than just additional data—to reliably improve performance.

---

> ### Author Response · Authors · 2025-11-25
> **Author's Response - Thank you for your feedback (part 6)**
>
> **(continued authors' response)**
>
>
> **(iii) Significant Statistical Analyses**.
> We have also conducted additional experiments and significant statistical analyses, and updated **Table 1** in the main paper (highlighted in **Blue**).
>
> **Table 1: Comparison between naive multi-task learning (MTL) baselines and the proposed CausalAffect. All models use the same training data sources and backbone. Statistical significance is assessed using paired t-tests over 5 matched runs (same data splits and random seeds), with differences considered significant at p < 0.05.**
>
> | Idx | Model Variant    | Training Data    | AffectNet(FER)      | RAFDB(FER)        | DISFA(AUD)         | BP4D(AUD)          | GFT(AUD)           | EmotioNet(AUD)     |
> | --- | -- | ---| --- | --- | --- | --- | ---| -- |
> | 1   | SG (single-task)   | Single-dataset   | 58.3    | 69.5     | 61.4       | 64.0         | 53.6   | 61.7               |
> | 3   | MTL (Shared Head)  | All datasets     | 58.8                | 70.0        | 53.0      | 57.2               | 57.9               | 47.5               |
> | 4   | MTL (Task-Specific Heads) | All datasets     | 60.3                | 71.9               | 53.6               | 57.6               | 58.1               | 49.3               |
> | 5   | CausalAffect (SG)         | Single-dataset   | /               | /                  | 67.0 $\pm$ 0.1      | 66.9 $\pm$ 0.2      | 61.1 $\pm$ 0.2      | 66.4 $\pm$ 0.1      |
> | 6   | **CausalAffect (All)**    | **All datasets** | **66.5 $\pm$ 0.1** | **84.9 $\pm$ 0.2** | **71.5 $\pm$ 0.2** | **66.7 $\pm$ 0.2** | **62.4 $\pm$ 0.3** | **65.0 $\pm$ 0.1** |
>
> - [1] Liu, Bo, et al. "Conflict-averse gradient descent for multi-task learning." Advances in Neural Information Processing Systems 34 (2021): 18878-18890.
> - [2] Yu, Tianhe, et al. "Gradient surgery for multi-task learning." Advances in neural information processing systems 33 (2020): 5824-5836.
> - [3] Javaloy, Adrián, and Isabel Valera. "Rotograd: Gradient homogenization in multitask learning." arXiv preprint arXiv:2103.02631 (2021).
> - [4] Li, Wei-Hong, Xialei Liu, and Hakan Bilen. "Learning multiple dense prediction tasks from partially annotated data." Proceedings of the IEEE/CVF Conference on Computer Vision and Pattern Recognition. 2022.
> - [5] Fontana, Maxime, Michael Spratling, and Miaojing Shi. "When multitask learning meets partial supervision: A computer vision review." Proceedings of the IEEE (2024).
> - [6] Huang, Chao, et al. "Partly supervised multi-task learning." 2020 19th IEEE International Conference on Machine Learning and Applications (ICMLA). IEEE, 2020.

---

> ### Author Response · Authors · 2025-11-25
> **Author's Response - Thank you for your feedback (part 7)**
>
> ###  Weakness 6
>
> ----
>
> > Weakness 6: there is no proper comparison in the methodology against prior work on graph neural networks for Action Unit detection/intensity estimation.
>
> We respectfully disagree with the statement that our submission provides “no proper comparison” to prior graph neural network approaches for AU detection or AU intensity estimation. In fact, we provide comparisons at both the **methodological level** and the **performance level**, and these comparisons clearly highlight why CausalAffect is fundamentally different from and consistently stronger than existing GNN-based AU models.
>
> - **Methodological level.**  As recognized by **Reviewer AkV1**, _“In this work, the authors propose the **first** (to my knowledge) approach for causal discovery in facial affect data.”_ Existing GNN-based AU methods (for detection or intensity estimation) primarily model **correlations** between AUs, but none of them performs **causal discovery** or provides **human-aligned explanations** of how specific AUs contribute to expressions. In contrast, **CausalAffect** explicitly learns **directed, polarity-aware AU→AU and AU→Expression graphs** and supports **counterfactual reasoning** in the AU latent space. We further provide a qualitative methodological comparison in **Figure 2** of the main manuscript, where we compare both the predefined/learned relations used in prior GNN-based AU models and the causal relations discovered by CausalAffect.
>
> - **Performance level.**  Beyond conceptual differences, we do compare against several strong graph-based AU detection/intensity estimation methods, **under same dataset setting** (Single Dataset, SG in Table 1), including **KSRL** (Chang & Wang, 2022), **ME-Graph** (Luo et al., 2022), and **MDHRM** (Wang et al., 2024). Under comparable settings on **BP4D** and **DISFA**, CausalAffect consistently outperforms these GNN baselines:
>     - On **BP4D**, CausalAffect exceeds KSRL / ME-Graph / MDHRM by  2.4% / 1.4% / 0.8% in AU detection performance.
>     - On **DISFA**, CausalAffect exceeds them by 2.5% / 4.6% / 0.8%, respectively.
>
> These results show that, even when evaluated purely as an AU model, CausalAffect is competitive or superior to prior GNN-based AU approaches, while at the same time providing a richer **causal** and **explainable** structure that these methods do not aim to model.

---

### Official Review · Reviewer_Hgze · 2025-11-01

**Soundness:** 3
**Presentation:** 2
**Contribution:** 3
**Rating:** 8
**Confidence:** 5

**Summary:**

This paper introduces a framework, named CausalAffect, for causal graph discovery in facial affective analysis. It aims to overcome the limitations of existing methods that lack psychological plausibility, rely on joint annotations, and ignore causality direction and inhibitory effects. Its main contribution is a weakly-supervised framework that learns a two-level (global and sample-adaptive) causal hierarchy for both AU→AU and AU→Expression dependencies, capable of modeling both excitatory and inhibitory relations. A key innovation is a feature-level counterfactual intervention mechanism that enforces true causal effects by perturbing latent AU features, eliminating the need for image synthesis.

**Strengths:**

The primary strength lies in its novel formulation of facial affective analysis as a causal discovery problem, moving beyond mere correlation to seek psychologically plausible mechanisms. The proposed framework, CausalAffect, integrates a two-level (global and sample-adaptive) graph structure to capture both stable population-level rules and context-specific dynamics. Its ability to model both excitatory and inhibitory relations, combined with an efficient feature-level counterfactual intervention, ensures the discovered dependencies are genuinely causal. It also eliminates the need for scarce jointly-annotated datasets by its weakly-supervised design. Well-designed and comprehensive ablation studies that confirm the necessity of each component.

**Weaknesses:**

-  The system complexity would be a concern to me. CausalAffect contains four different modules and that caused a large number of loss functions. Their corresponding hyperparameters (e.g., $\lambda_{ib}$, $\lambda_{DAG}$, $\lambda_{consist}$) also **increase the risk of training instability** and these factors would ** make it difficult for other researchers to reproduce the results**.

- The paper employs a significant number of mathematical symbols and formulas, which is commendable. However, to some extent, this comes at the cost of reading fluency and also compresses the available space for text, causing some of the results analysis to be relegated to the appendix.

**Questions:**

**Question 1: ** Validating psychological plausibility against existing literature like FACS is a clever approach. However, for the 'new' causal relations discovered by the model, for example: the subtle inhibitory ones, it seems that we don't have an objective 'gold standard' or ground truth. This makes it difficult to determine whether these are genuine psychological insights or merely statistical artifacts learned from the specific datasets. What's your thoughts on this?

**Question 2: ** In Table 3, the result for GFT at idx 13 should be bolded, assuming that bolding is used to indicate the best result.

---

> ### Author Response · Authors · 2025-11-24
> **Author's Response - Thank you for your feedback (part 1)**
>
> We sincerely thank the Reviewer for the thoughtful and valuable feedback. We appreciate the opportunity to improve the clarity and quality of our work. All modifications made in the manuscript and Appendix are highlighted in **Red** and **Blue**. Our detailed responses to each comment are provided below.
>
> ---
>
> ### Weakness 1
>
> ----
>
> > Weakness 1: The system complexity would be a concern to me. CausalAffect contains four different modules and that caused a large number of loss functions. Their corresponding hyperparameters (e.g. $λ\_\text{ib}$, $λ\_\text{DAG}$, $λ\_\text{consist}$) also **increase the risk of training instability** and these factors would **make it difficult for other researchers to reproduce the results**.
>
> Although CausalAffect contains multiple modules, an important property of our design is that these modules are **weakly coupled**: each $λ$–hyperparameter regulates the gradient flow **within its own module**, rather than jointly pushing on a shared representation as in many traditional multi-loss architectures. As a result, different $λ$ values do not compete or interfere with each other during optimization (e.g., no gradient conflict), and changing a $λ$ only affects the **speed of convergence within that module**, rather than the overall training stability. This modularity also ensures reproducibility: every $λ$ controls a well-isolated functional component, and all $λ$ values remain fixed across all datasets and experiments. To ensure **full reproducibility**, we will release all training code, models, and configuration files upon acceptance.
>
> To provide a complete analysis, we add a full set of sensitivity experiments in **Appendix J** (markd as **Red**). Below, we summarise the main findings and the rationale behind each hyperparameter choice.
>
> **(i) AU Bottleneck Loss Group ($λ_\text{ib}$, $λ_\text{align}$, $λ_\text{decorr}$)**
>
> The AU bottleneck module enforces a disentangled and low-redundancy AU representation by balancing **information compression**, **semantic alignment**, and **inter-AU decorrelation**. $λ_{\text{ib}}$ controls the strength of HSIC-based information compression, $λ_{\text{align}}$ governs how strongly AU features are aligned with one-hot AU labels, and $λ_{\text{decorr}}$ penalizes covariance to reduce redundancy among different AUs.
> - We set $λ_{\text{ib}}=1$ and $λ_{\text{align}}=1$ so that compression and semantic supervision act on a **matched and balanced scale**, preventing either from dominating the optimization; smaller values lead to under-regularization and noisy AU embeddings, while larger values cause over-compression or overfitting.
> - In contrast, the decorrelation term $λ_\text{decorr}$ is **quadratic in covariance** and therefore inherently much stronger than the other two, meaning it must operate as a **lightweight auxiliary regularizer** rather than a primary signal. For this reason we adopt a small value $λ_{\text{decorr}}=0.2$, which is sufficient to reduce redundancy without disrupting natural physiological AU co-activation patterns.
>
> **Table 1: AU Bottleneck $\lambda$ Sensitivity**
>
> | λ Name            | Value          | RAF-DB Acc. | BP4D AU F1 | Interpretation                                        |
> | ----------------- | -------------- | ----------- | ---------- | ----------------------------------------------------- |
> | $λ_\text{ib}$     | 0.5            | 84.4        | 66.3       | Too little compression → AU redundancy appears.       |
> | $λ_\text{ib}$     | **1.0 (Best)** | **85.0**    | **67.1**   | Balanced compression; most faithful AU bottleneck.    |
> | $λ_\text{ib}$     | 2.0            | 84.6        | 66.5       | Excessive compression → information loss.             |
> | $λ_\text{align}$  | 0.5            | 84.2        | 66.2       | Under-alignment → label-feature mismatch.             |
> | $λ_\text{align}$  | **1.0 (Best)** | **85.0**    | **67.1**   | Optimal alignment with semantic AU labels.            |
> | $λ_\text{align}$  | 2.0            | 84.7        | 66.8       | Over-penalizing alignment yields diminishing returns. |
> | $λ_\text{decorr}$ | 0.1            | 84.7        | 66.6       | Weak decorrelation → slight redundancy remains.       |
> | $λ_\text{decorr}$ | **0.2 (Best)** | **85.0**    | **67.1**   | Minimal yet effective disentanglement.                |
> | $λ_\text{decorr}$ | 0.4            | 84.2        | 66.4       | Too strong → overly sparse AU basis.                  |

---

> ### Author Response · Authors · 2025-11-24
> **Author's Response - Thank you for your feedback (part 2)**
>
> > Weakness 1: The system complexity would be a concern to me. CausalAffect contains four different modules and that caused a large number of loss functions. Their corresponding hyperparameters (e.g. $λ\_\text{ib}$, $λ\_\text{DAG}$, $λ\_\text{consist}$) also **increase the risk of training instability** and these factors would **make it difficult for other researchers to reproduce the results**.
>
> **(continued authors' response)**
>
>
>
> **(ii) DAG Regularization $λ_{\text{DAG}}^g$ and $λ_{\text{DAG}}^s$**
>
> Both the global and sample-adaptive AU→AU graphs rely on the same **soft acyclicity constraint**, and therefore follow identical stability principles. When $λ_{\text{DAG}}$ is **too small**, the acyclicity penalty becomes ineffective and the learned graph degenerates into **dense** structures. Conversely, when $λ_{\text{DAG}}$ is **too large**, the constraint becomes overly restrictive, yielding an **overly sparse** graph that suppresses meaningful AU dependencies. To balance these effects, we adopt a **moderate penalty** of $λ_{\text{DAG}}^g = λ_{\text{DAG}}^s = 0.5$, which is strong enough to encourage sparse but not so strong as to eliminate important relational edges. This shared setting yields stable, interpretable causal structures at both the global and per-sample levels while preserving sufficient flexibility to model AU interactions.
>
> **Table 2: Sensitivity for Both $λ_{\text{DAG}}^g$ and $λ_{\text{DAG}}^s$**
>
> | λ Value        | RAF-DB Acc. | BP4D AU F1 | Interpretation                                     |
> | -------------- | ----------- | ---------- | -------------------------------------------------- |
> | 0.1            | 84.8        | 66.7       | Acyclicity too weak → noisy cyclic edges emerge.   |
> | **0.5 (Best)** | **85.0**    | **67.1**   | Optimal structural balance; best causal fidelity.  |
> | 1.0            | 84.7        | 66.4       | Over-regularized → graph becomes under-expressive. |
>
> **(iii) Counterfactual Supervision ($λ_{\text{consist}} = λ_{\text{discrep}}$)**
>
> Counterfactual supervision requires the model to satisfy two complementary principles:
> - **consistency** — factual and counterfactual outputs should remain similar when the intervened AU is _not_ a causal parent;
> - **discrepancy** — the outputs should differ when the intervened AU _is_ a causal parent.
>
> These two objectives act in **opposite directions**, their losses must be placed on a matched scale, otherwise one will dominate and distort the learning dynamics. When the shared weight is too small, the model underutilizes counterfactual information and fails to distinguish causal from non-causal dependencies. When the weight is too large, the model collapses into trivial behaviors (e.g., always-consistent or always-divergent outputs). To maintain a stable equilibrium between enforcing structural invariance and preserving meaningful causal effects, we therefore tie the two hyperparameters and set $\lambda_{\text{consist}} = \lambda_{\text{discrep}} = 0.5$, which empirically yields the most reliable counterfactual behavior.
>
>
> **Table 3: Counterfactual Supervision Sensitivity**
>
> | $λ_{\text{consist}} = λ_{\text{discrep}}$ | RAF-DB Acc. | BP4D AU F1 | Interpretation                                                                            |
> | ----------------------------------------- | ----------- | ---------- | ----------------------------------------------------------------------------------------- |
> | 0.25                                      | 84.0        | 66.2       | CF penalty too weak → cannot enforce causal invariance; interventions ineffective.        |
> | **0.5 (default)**                         | **85.0**    | **67.1**   | Balanced CF consistency–discrepancy; strongest overall performance.                       |
> | 1.0                                       | 84.4        | 66.4       | CF penalty too strong → suppresses causal diversity and destabilizes factual predictions. |

---

> ### Author Response · Authors · 2025-11-24
> **Author's Response - Thank you for your feedback (part 3)**
>
> ### Weakness 2
>
> ---
>
> > Weakness 2: The paper employs a significant number of mathematical symbols and formulas, which is commendable. However, to some extent, this comes at the cost of reading fluency and also compresses the available space for text, causing some of the results analysis to be relegated to the appendix.
>
> We fully agree that the current version contains many mathematical symbols and formulations under the strict page limit. With the additional page available in the discussion stage, we have revised the manuscript to improve readability: each major section now begins with a brief **intuitive explanation** (highlighted in **blue**), and key equations are preceded by **concise summarie**s to help readers understand the underlying motivation. Mathematical formulations remain essential for **technical transparency**, **reproducibility**, and for ensuring that each component of *CausalAffect* is precisely defined. Without these explicit formulations, several modules would become ambiguous or difficult to verify.
>
>
> ### Question 1
>
> ----
>
> > **Question 1:** Validating psychological plausibility against existing literature like FACS is a clever approach. However, for the 'new' causal relations discovered by the model, for example: the subtle inhibitory ones, it seems that we don't have an objective 'gold standard' or ground truth. This makes it difficult to determine whether these are genuine psychological insights or merely statistical artifacts learned from the specific datasets. What's your thoughts on this?
>
> Indeed, while literature-supported AU→AU and AU→Expression relations (e.g., from FACS and affective science) provide a psychological “gold standard,” newly discovered relations, especially subtle inhibitory ones, lack direct ground truth. To assess whether these edges reflect **psychological plausibility structure** rather than **dataset-specific statistical artifacts**, we conducted an additional expert validation study in **Appendix I** (marked as **Red**). Below, we summarise the main procedure and results.
>
> **1. Participants**: We invited **five domain experts** to independently evaluate the causal relations: 3 specialists in affective cognition and emotion science, 1 researcher in cognitive science, and 1 psychologist with expertise in facial behavior and affect. All experts have between **5 and 15 years of research experience** in their respective fields, and each was asked to assess whether the presented AU→AU and AU→Expression causal relations (mixed with literature-supported edges and randomly generated controls) were psychologically plausible by providing **True/False** judgments (along with confidence ratings).

---

> ### Author Response · Authors · 2025-11-24
> **Author's Response - Thank you for your feedback (part 4)**
>
> > **Question 1:** Validating psychological plausibility against existing literature like FACS is a clever approach. However, for the 'new' causal relations discovered by the model, for example: the subtle inhibitory ones, it seems that we don't have an objective 'gold standard' or ground truth. This makes it difficult to determine whether these are genuine psychological insights or merely statistical artifacts learned from the specific datasets. What's your thoughts on this?
>
> **(continued authors' response)**
>
> **2. Evaluation Material**: We constructed **three categories of evaluation pairs**, each serving a distinct role in validating the psychological plausibility of the learned causal relations:
> - **(A) Literature-Supported “True” Relations (thirty-nine AU→Expression, excitatory only)**: This category contains **thirty-nine AU→Expression causal relations** drawn from established FACS documentation and affective-science literature. All relations in this set are **excitatory**, meaning the AU is known to _increase_ the likelihood or intensity of a particular expression. Examples include: _“Lip Corner Puller (AU12) increases Happiness”_, _“Brow Lowerer (AU4) increases Anger”_, and _“Inner and Outer Brow Raise (AU1+AU2) increases Surprise.”_ These relations serve as **ground-truth positive anchors**, allowing us to verify that experts consistently recognize well-established psychological dependencies.
> - **(B) CausalAffect-Discovered Relations Requiring Expert Validation (fifty-five AU→AU / forty-five AU→Expression, excitatory & inhibitory)**: This category contains **one hundred relations** extracted from our learned causal graphs, covering both **AU→AU** and **AU→Expression** directions and including **both excitatory and inhibitory** effects. Importantly, some of these edges **overlap with the literature-supported set (A)**, reflecting that the model is able to recover known ground-truth patterns, while others represent genuinely _new_ causal hypotheses—particularly the subtle inhibitory edges emphasized by the reviewer. Examples presented to experts include: _“Chin Raiser (AU17) increases Lip Pressor (AU24)”_, _“Jaw Drop (AU26) inhibits Lip Stretcher (AU20)”_, _“Lip Corner Depressor (AU15) increases Disgust”_, and _“Cheek Raiser (AU6) inhibits Sadness.”_ To ensure a clean evaluation, all overlapping relations between sets A and B were deduplicated before being shown to experts. This category therefore tests whether the model’s recovered and novel causal patterns correspond to **psychologically meaningful dependencies** rather than **dataset-specific statistical artifacts**.
> - **(C) Random Relations as Negative Controls (sixty-eight AU→AU / twenty-eight AU→Expression, excitatory & inhibitory)**: This category consists of **ninety-six randomly sampled relations** across AU→AU and AU→Expression pairs that **do not appear in either A or B**. These relations include both excitatory and inhibitory directions but lack any known support from FACS or affective psychology. Examples include: _“Brow Lowerer (AU4) increases Lip Corner Puller (AU12)”_, _“Lip Stretcher (AU20) inhibits Fear”_, and _“Nose Wrinkler (AU9) increases Happiness.”_ These serve as **negative controls**, enabling us to verify that experts can reliably reject implausible causal statements and establishing a baseline against which model-discovered relations (set B) can be compared.
>
> **3. Procedure**: All evaluation pairs from sets A, B, and C were merged and **deduplicated**, then randomly shuffled for each expert independently. Experts were blind to (i) the source set (A/B/C) and (ii) whether a relation was recovered or novel. Each expert judged every relation as **True** (psychologically plausible) or **False** (implausible), and additionally provided a **1–5 confidence score**. This yielded **five independent votes per relation**, allowing us to measure both plausibility and inter-expert reliability.
>
> **4. Metrics & Aggregation**: For each set, we report:
> 1. **Expert True Rate**: mean fraction of True votes across experts and relations.
> 2. **Majority-True Rate**: fraction of relations judged True by ≥3/5 experts.
> 3. **High-Confidence True Rate**: True rate restricted to confidence ≥4 votes.
> 4. To validate expert reliability, we treat A as gold-true and C as gold-false and compute **Accuracy / Precision / Recall / F1**, and **Fleiss’K** for agreement.
> 5. To test whether model relations are more plausible than random controls, we compare B vs. C using a **two-sided proportion test**.

---

> ### Author Response · Authors · 2025-11-24
> **Author's Response - Thank you for your feedback (part 5)**
>
> > **Question 1:** Validating psychological plausibility against existing literature like FACS is a clever approach. However, for the 'new' causal relations discovered by the model, for example: the subtle inhibitory ones, it seems that we don't have an objective 'gold standard' or ground truth. This makes it difficult to determine whether these are genuine psychological insights or merely statistical artifacts learned from the specific datasets. What's your thoughts on this?
>
> **(continued authors' response)**
>
>
> **5. Results And Analysis**
>
> **5.1 Overall plausibility comparison**
>
> As shown in Table 4, experts clearly distinguished between literature-supported relations (A), model-discovered relations (B), and random controls (C). Literature-supported relations received high plausibility scores, whereas random relations were overwhelmingly rejected. Crucially, the CausalAffect-discovered relations (B) achieved a **True Rate of 0.75**, which is **over three times higher** than the random baseline (C: 0.23) and moves toward the plausibility level of literature-supported edges (A: 0.88). This strong separation demonstrates that the causal relations proposed by our model are **far from random statistical artifacts**. Moreover, the comparatively low variance in A (0.21) and C (0.20) indicates that expert judgments were stable and internally consistent across clearly valid and clearly invalid causal statements.
>
> **Table 4: Expert Validation Statistics**
>
> | Relation Set                | #Pairs | Expert True Rate (mean ± std) | Majority-True Rate (≥3/5) | High-Conf True Rate (conf ≥4) | Interpretation                                                                    |
> | --------------------------- | -----: | ----------------------------: | ------------------------: | ----------------------------: | --------------------------------------------------------------------------------- |
> | **A: Literature-supported** |     39 |                   0.88 ± 0.21 |                      0.92 |                          0.90 | Experts strongly endorse established psychological relations.                     |
> | **B: CausalAffect**         |    100 |                   0.75 ± 0.24 |                      0.84 |                          0.70 | Majority of CausalAffect-discovered relations judged plausible; far above random. |
> | **C: Random negatives**     |     96 |                   0.23 ± 0.20 |                      0.09 |                          0.19 | Experts reliably reject implausible, randomly generated relations.                |
>
> **5.2 Reliability of expert judgments**
>
> As shown in Table 5, we evaluate the reliability of expert judgments using the two sets with **known correctness**: the literature-supported relations **A** (psychologically valid) and the randomly generated relations **C** (psychologically invalid). Experts exhibit a strong ability to discriminate between these two reference sets, achieving **0.87 accuracy**, **0.89 precision**, **0.88 recall**, and an overall **F1 score of 0.88**. These results indicate that experts almost never misclassify invalid relations as plausible while consistently recognizing established psychological dependencies. The **Fleiss’ κ of 0.61** further reflects **moderate-to-substantial agreement** among the five experts—considered strong for psychological and facial-behavior annotation tasks where subjective interpretation is inherent. Together, these metrics confirm that expert assessments are stable, reliable, and well-suited as an external validation signal for evaluating the psychological plausibility of the CausalAffect-discovered causal relations.
>
> **Table 5: Reliability on Ground-Truth Reference Sets (A=true, C=false)**
>
> | Metric    | Value | Interpretation                                      |
> | --------- | ----: | --------------------------------------------------- |
> | Accuracy  |  0.87 | Experts reliably separate valid vs. invalid edges.  |
> | Precision |  0.89 | Low false-positive tendency.                        |
> | Recall    |  0.88 | True edges are consistently recognized.             |
> | F1        |  0.88 | Strong overall correctness.                         |
> | Fleiss’K  |  0.61 | Moderate-to-substantial agreement across 5 experts. |

---

> ### Author Response · Authors · 2025-11-24
> **Author's Response - Thank you for your feedback (part 6)**
>
> > **Question 1:** Validating psychological plausibility against existing literature like FACS is a clever approach. However, for the 'new' causal relations discovered by the model, for example: the subtle inhibitory ones, it seems that we don't have an objective 'gold standard' or ground truth. This makes it difficult to determine whether these are genuine psychological insights or merely statistical artifacts learned from the specific datasets. What's your thoughts on this?
>
> **(continued authors' response)**
>
> **5.3 Subtle inhibitory relations (reviewer’s key concern)**
>
> A key concern raised by the reviewer is whether the **inhibitory causal relations** discovered by our CausalAffect may simply be statistical rather than psychologically meaningful structure. To directly address this, we further stratified Set B (CausalAffect-discovered relations) by **relation type** (AU→AU vs. AU→Expr) and **causal direction** (excitatory vs. inhibitory), and evaluated expert judgments separately for each subset.
>
> As shown in Table 6, all four subsets achieve plausibility scores far above the random baseline. Notably, the inhibitory subsets show **True Rates of 0.67 (AU→AU)** and **0.77 (AU→Expr)**, as well as strong majority endorsement (≥0.83). These scores are **substantially higher** than the random-negative True Rate (0.23), demonstrating that experts interpret the inhibitory edges as **psychologically plausible** rather than random noise. This provides strong evidence that the subtle inhibitory mechanisms captured by our CausalAffect correspond to psychological meaningful regulatory patterns in facial behavior and emotional expression.
>
> **Table 6 — Breakdown for CausalAffect-Discovered Relations (Set B)**
>
> | Subset                  | #Pairs | True Rate | Majority-True Rate | Interpretation                                                                |
> | ----------------------- | -----: | --------: | -----------------: | ----------------------------------------------------------------------------- |
> | AU→AU, excitatory       |     32 |  0.72 |           0.78 | Positive AU interactions broadly accepted by experts.                         |
> | **AU→AU, inhibitory**   |     23 |  **0.67** |           **0.83** | **Inhibitory AU→AU relations judged plausible and non-random**.                   |
> | AU→Expr, excitatory     |     18 |  0.85 |           0.90 | Strong alignment with well-known affective expression dynamics.               |
> | **AU→Expr, inhibitory** |     17 |  **0.77** |           **0.87** | **Inhibitory emotional cues viewed as psychological meaningful, not statistics**. |
>
> Together, these results indicate that CausalAffect’s learned causal relations, including **novel and subtle inhibitory edges** are validated by external experts as psychologically plausible and are **significantly more meaningful than random controls**. The high reliability on gold anchors (A vs. C) further confirms that these judgments provide a credible external check. Therefore, the newly discovered relations are unlikely to be dataset-specific statistical artifacts, but instead represent valid and interpretable causal hypotheses about AU and expression structure.
>
> ---
>
> ### Question 2
>
> ----
>
> > **Question 2:** In Table 3, the result for GFT at idx 13 should be bolded, assuming that bolding is used to indicate the best result.
>
> We have corrected Table 3 accordingly. The value for **GFT at index 13 is now properly bolded** to indicate the best-performing result.

---

### Official Review · Reviewer_6Lzr · 2025-11-03

**Soundness:** 3
**Presentation:** 3
**Contribution:** 3
**Rating:** 6
**Confidence:** 3

**Summary:**

This paper introduces CausalAffect, a novel framework designed to learn causal relations among Action Units (AUs) and between expressions and AUs. The work features extensive experimentation and analysis, providing interesting insights into these relationships.

**Strengths:**

* The trained models and code will be released, which is a significant contribution to the research community.
* Extensive experiments, including ablation studies, are conducted on multiple datasets. The paper compares the method against competitive baselines and demonstrates promising performance.
* Overall, the paper is well-written and presented in good format.
* Interesting analysis and visualizations are presented in Section 4.2. These provide readers with a straightforward understanding of how the learned causal relations align with or differentiate from previous studies. (Although some results appear unusual, which is addressed in the Questions section.)
* Straightforward case studies are provided to illustrate the effectiveness of the proposed method.

**Weaknesses:**

* Figure 2 is cited in line 46 but is located on page 8, which is too far from the relevant text and disrupts the flow of reading.
* The paper lacks a Related Work section.
* While Table 1 shows promising performance, the comparison feels unfair because the best CausalAffect configuration utilizes additional training data sources. Without this extra data, the performance is very close to the best baseline, and a significant statistical test is missing to confirm the difference.

**Questions:**

* Regarding the EmotioNet experiment in Table 1, could you please explain why joint training (with AU datasets) appears to detrimentally affect the expression recognition performance?
* In Figure 2, the GNN-Learned Correlation visualization looks weird to me. It seems to imply that every AU is highly correlated with every expression. Could you elaborate on this specific finding? Similarly, for the AU-AU co-occurrence in BP4D, the strong correlation between AU6 and AU14 warrants further explanation.

---

> ### Author Response · Authors · 2025-11-24
> **Author's Response - Thank you for your feedback (part 1)**
>
> We sincerely thank the Reviewer for the thoughtful and valuable feedback. We appreciate the opportunity to improve the clarity and quality of our work. All modifications made in the manuscript are highlighted in **Blue**. Our detailed responses to each comment are provided below.
>
> ----
>
> ### Weakness 1
> ----
>
> > Weakness 1: Figure 2 is cited in line 46 but is located on page 8, which is too far from the relevant text and disrupts the flow of reading.
>
> In the final version, we will reposition Figure 2 so that it appears immediately after the paragraph where it is first referenced (around line 46). This ensures that the figure is visually accessible at the appropriate point in the discussion and improves the overall clarity of the presentation.
>
>
> ---
>
> ### Weakness 2
>
> ---
>
> > Weakness 2: The paper lacks a Related Work section.
>
> Due to page limitations in the main manuscript, the Introduction provided only a concise overview of related work, focusing on contrasting existing dependency-learning approaches, outlining their limitations, and motivating the need to study causal relations. With the additional page available in the discussion stage, we have expanded the coverage accordingly: a more detailed **Section 2 (Related Work)** has been added in the main paper (highlighted in **blue**), and an extended **Appendix H**  has been included in the Supplementary Material (also marked in **blue**) to provide a comprehensive and rigorous discussion.
>
> **Appendix H** contains two dedicated sections—**“Modeling Structured Relations Between AUs and Expressions”** and **“Causal and Counterfactual Modeling in Facial Behavior”**—which systematically review prior work on AU→AU / AU→Expression dependency learning and causal modeling in facial. These sections further substantiate why **causal discovery is both important and necessary** for capturing directional, asymmetric, and psychologically meaningful AU structures, thereby complementing and strengthening the motivation presented in the main paper.

---

> ### Author Response · Authors · 2025-11-24
> **Author's Response - Thank you for your feedback (part 2)**
>
> ### Weakness 3
>
> ---
>
> > Weakness 3: While Table 1 shows promising performance, the comparison feels unfair because the best CausalAffect configuration utilizes additional training data sources. Without this extra data, the performance is very close to the best baseline, and a significant statistical test is missing to confirm the difference.
>
> Regarding the concern that our best CausalAffect configuration “_utilizes additional training data_”, we would like to clarify that we have conducted **same-dataset** experiments, including both **Naive MTL** (Shared Head) and **MTL with Task-Specific Heads**. These results are thoroughly presented in **Main Paper Ablation Study (Table 3)** and **Appendix A (Table 1)**. The analyses demonstrate that MTL does not universally improve performance, it can even lead to degraded results. To further clarify this point, we have reorganized the comparisons in **Table 1 below**, accompanied by the relevant theoretical discussion, to show that the performance advantages of CausalAffect do not stem from merely adding more training data.
>
> Regarding the claim that “*without this extra data, the performance is very close to the best baseline*”, we would like to clarify that our single-dataset (SG) performance is not merely “very close” to the strongest baseline. Under the SG setting across all AU benchmarks, CausalAffect consistently outperforms the best existing method, achieving gains of **+0.7% on BP4D**, **+0.6% on DISFA**, and **+1.0% on EmotioNet**. Importantly, these improvements are **statistically significant ($p = 0.03$ across five independent runs)**, further confirming that the differences are not due to random variation.
>
>
> **(i) Additional Tasks Harm MTL (Gradient Conflict).**
> - In response to the reviewer’s concern that the performance gains might stem from using more training data, we would like to clarify that our findings do not support this interpretation. As shown in **Table 1 below**, all models are trained under a **strictly controlled setting**, using the **same ConvNeXt-Base backbone** and the **exact same datasets**. Under these conditions, simply adding more datasets or tasks does ***not*** consistently improve performance in standard multi-task learning (MTL) frameworks. Both the **Naive MTL** (Shared Head) and **MTL with Task-Specific Heads** configurations yield only marginal gains on the FER benchmarks (AffectNet, RAF-DB), while **all AU detection benchmarks consistently degrade**, in some cases substantially (e.g., DISFA, EmotioNet). These results indicate that the improvements achieved by CausalAffect cannot be attributed to additional training data; instead, they arise from its causal graph learning mechanism rather than from naive dataset expansion.
> - This behavior is explained in **many works in the multi-task learning literature** [1, 2, 3], which consistently show that jointly training heterogeneous tasks introduces **gradient conflicts** that hinder optimization. **Similar observations have also been reported in the affective computing domain**[4, 5], where jointly learning AU detection and expression recognition often leads to incompatible gradients and degraded performance. FER tasks rely mainly on **global facial representations**, whereas AU detection depends on **fine-grained local muscle activations**, leading to inherently incompatible gradient directions. Such conflicting updates destabilize the shared encoder and produce entangled, suboptimal feature representations. The problem is further exacerbated by **partial annotation** [5, 6, 7, 8], which creates large disparities in gradient magnitudes across tasks and amplifies **negative transfer**. Even with task-specific prediction heads (Row 4 below), the shared backbone remains the bottleneck, forcing mutually inconsistent supervision into a single representation space.

---

> ### Author Response · Authors · 2025-11-24
> **Author's Response - Thank you for your feedback (part 3)**
>
> > Weakness 3: While Table 1 shows promising performance, the comparison feels unfair because the best CausalAffect configuration utilizes additional training data sources. Without this extra data, the performance is very close to the best baseline, and a significant statistical test is missing to confirm the difference.
>
> **(continued authors' response)**
>
>
> **(ii) Additional Tasks Benefit CausalAffect (But Not Simply “More Data”).**
> - In contrast to standard MTL, our CausalAffect (**Table 1 below**) achieves consistent and substantial gains across _all_ benchmarks under exactly the same data and backbone conditions—not because it sees “more samples”, but because it leverages **more effective AU dimensions** to build a richer causal structure. As analyzed in **Appendix B**, the key factor is how the AU space is expanded. For example, DISFA alone annotates only 8 AUs in a controlled, in-the-lab setting, so its AU→AU graph is restricted to 8 nodes. EmotioNet, in contrast, provides 11 AUs in an in-the-wild setting, and its AU set strictly covers that of DISFA. When DISFA is trained jointly with EmotioNet, the AU graph for DISFA is effectively upgraded from 8 to 11 nodes: the additional in-the-wild AUs enrich and regularize the causal structure, reinforcing and completing the limited lab-only relations. This is why DISFA benefits under joint training. However, the reverse is not true: EmotioNet already has 11 AUs, so adding DISFA does not increase its AU dimensionality; instead, the lab-specific AU relations from DISFA can _perturb_ the in-the-wild causal structure and make EmotioNet’s AU graph less stable, which explains why EmotioNet does not always gain from pairing with a smaller, lab-style dataset.
> - When _all_ datasets are trained together, the joint AU space expands further to 17 AUs, which substantially increases the expressive power of both the AU→AU and AU→Expression causal graphs. Crucially, our framework is fundamentally different from conventional MTL: we do **not** ask all tasks to compete for a single shared representation. Instead, we **explicitly model** AU→AU and AU–Expr causal relations, so tasks are not tightly entangled, and potential task/gradient conflicts are absorbed and “spread out” in the graph structure rather than being forced entirely onto a shared backbone. The **counterfactual module** strengthens meaningful structural dependencies, while the **AU bottleneck disentanglement** module reduces spurious couplings and conflicts. Together, these components effectively mitigate the classic negative transfer and gradient conflict issues of MTL, allowing additional AU dimensions—rather than just additional data—to reliably improve performance.
>
> **(iii) Significant Statistical Analyses**.
> In line with the reviewer’s suggestion, we have conducted additional experiments and significant statistical analyses, and updated **Table 1** in the main paper (highlighted in **Blue**).
>
> **Table 1: Comparison between naive multi-task learning (MTL) baselines and the proposed CausalAffect. All models are trained with the same datasets and ConvNeXt-Base backbone. Statistical significance is evaluated using paired t-tests over five matched runs, CausalAffect achieves a significant improvement with $p = 0.03$ ($p<0.05$ are considered statistically significant).**
>
> |Idx|Model Variant|Training Data| AffectNet(FER)| RAFDB(FER)|DISFA(AUD) | BP4D(AUD)| GFT(AUD)| EmotioNet(AUD)|
> |-|-| -| --| - | -| - | - | --- |
> | 1| SG (single-task) | Single-dataset | 58.3| 69.5| 61.4| 64.0| 53.6 | 61.7|
> | 3   | MTL (Shared Head) | All datasets| 58.8| 70.0| 53.0| 57.2| 57.9| 47.5|
> | 4   | MTL (Task-Specific Heads) | All datasets| 60.3| 71.9| 53.6| 57.6| 58.1 | 49.3 |
> | 5   | **CausalAffect (SG)**| Single-dataset   | /  | /| 67.0 $\pm$ 0.1| **66.9 $\pm$ 0.2**| 61.1 $\pm$ 0.2 | **66.4 $\pm$ 0.1** |
> | 6   | **CausalAffect (All)**    | **All datasets** | **66.5 $\pm$ 0.1** | **84.9 $\pm$ 0.2** | **71.5 $\pm$ 0.2** | 66.7 $\pm$ 0.2 | **62.4 $\pm$ 0.3** | 65.0 $\pm$ 0.1 |
>
> - [1] Liu, et al., "Conflict-averse gradient descent for multi-task learning." NeurIPS 2021
> - [2] Yu, et al., "Gradient surgery for multi-task learning." NeurIPS 2020
> - [3] Javaloy, et al., "Rotograd: Gradient homogenization in multitask learning." arXiv preprint arXiv:2103.02631 (2021).
> - [4] Kollias, et al., "Distribution matching for multi-task learning of classification tasks: a large-scale study on faces & beyond." Proceedings of the AAAI Conference on Artificial Intelligence. 2024.
> - [5] Deng et. al "Multitask emotion recognition with incomplete labels."  IEEE FG, 2020.
> - [6] Li, et al., "Learning multiple dense prediction tasks from partially annotated data." CVPR 2022.
> - [7] Fontana, et al., "When multitask learning meets partial supervision: A computer vision review." Proceedings of the IEEE (2024).
> - [8] Huang, et al., "Partly supervised multi-task learning." IEEE ICMLA, 2020.

---

> ### Author Response · Authors · 2025-11-24
> **Author's Response - Thank you for your feedback (part 4)**
>
> ### Question 1
>
> ---
>
> > Question 1: Regarding the EmotioNet experiment in Table 1, could you please explain why joint training (with AU datasets) appears to detrimentally affect the expression recognition performance?
>
> As we noted in our response to Weakness 3, CausalAffect does not automatically improve by simply incorporating more data. Its behavior under joint training is largely governed by **how the AU components are composed across datasets**. Since EmotioNet is fundamentally an **AU dataset**, in CausalAffect, the AU task is mainly influenced by **AU diversity, AU frequency, and the annotation types** provided by the auxiliary datasets. In contrast, the **expression-recognition task** depends more critically on **the coverage and completeness of AU patterns**. **Appendix B** provides an in-depth analysis of this phenomenon. Below we summarise the key observations from the perspectives of **AU Detection (AU→AU)** and **Expression Recognition (AU→Expression)** tasks:
>
> **(i) For AU Detection Tasks: Role of AU Diversity, Frequency, and Dataset Type**
>
> Before detailing the three factors, we first summarise why EmotioNet shows a mild performance drop under joint training. In EmotioNet, the AU vocabulary is already large and diverse (11 AUs, in-the-wild). When additional datasets introduce no new AU coverage, yet inject lab-specific biases and low-frequency AUs, the resulting AU→AU causal graph becomes slightly less stable. This reduces the reliability of the AU signals, leading to the performance decrease observed in Table 1. We expand the three contributing factors below.
>
> - **(1) AU diversity.** For the AU→AU causal graph, the variety of the AU set is essential. When the AU space is expanded with additional _informative_ AUs, such as combining DISFA’s 8 AUs with EmotioNet’s 11 AUs, where EmotioNet both covers and extends the AU set of DISFA, the resulting AU→AU graph becomes more expressive. This richer AU composition allows the model to capture more nuanced and physiologically meaningful dependencies, which leads to improved performance on DISFA compared with using the original 8-AU graph. In contrast, EmotioNet already contains complete AU set among two datasets. Joint training therefore introduces no new AU categories and does not increase its AU diversity. Instead, the added datasets contribute redundant or low-frequency AUs and lab-specific AU patterns that differ from natural in-the-wild behaviour. These signals perturb EmotioNet’s well-formed AU→AU structure without adding meaningful information, which explains the slight performance drop observed under joint training.
> - **(2) AU frequency.** As shown in **Appendix B Table 2**, **AU detection is strongest when the causal graph is constructed from frequent AUs**. Using the “8 most frequent AUs” from BP4D yields the best performance on EmotioNet and even **surpasses** the SG baseline. In contrast, including many **low-frequency AUs** (least-frequent subsets or the full 12-AU configuration) slightly degrades EmotioNet: these rare units are too sparsely activated to support stable causal edges and instead inject noise into the AU→AU graph.
> - **(3) Dataset characteristics (in-the-lab vs. in-the-wild).** In-the-lab datasets (e.g., DISFA, BP4D) contain clean but limited AU patterns. In-the-wild datasets (e.g., EmotioNet) contain more diverse AU compositions. When DISFA (8 AUs) is paired with EmotioNet (11 AUs), EmotioNet enriches DISFA’s causal graph and improves its performance. The opposite, however, does not hold: adding DISFA to EmotioNet introduces lab-specific AU patterns, low-frequency AUs, and no increase in AU coverage. This reduces the stability of EmotioNet’s AU→AU causal graph and leads to the observed performance drop.
>
>
> **(ii) For Expression Recognition tasks: Importance of AU Coverage**
>
> For **AU→Expression** causal relations, **Appendix B Table 2** reveals a different pattern: here, the **overall AU coverage (i.e., the number of supervised AUs)** is the dominant factor, and larger AU sets consistently improve performance. Expression recognition on RAF-DB and AffectNet becomes stronger as we move from small AU subsets to the full 12-AU configuration. Even low-frequency AUs contribute complementary, fine-grained cues that refine AU prototypes and enable more nuanced AU→Expr mappings. Expression tasks thus benefit from a **richer and more compositional AU basis**, and are more tolerant to AU sparsity than AU detection.

---

> ### Author Response · Authors · 2025-11-24
> **Author's Response - Thank you for your feedback (part 5)**
>
> ### Question 2
>
> ----
>
> > Question 2: In Figure 2, the GNN-Learned Correlation visualization looks weird to me. It seems to imply that every AU is highly correlated with every expression. Could you elaborate on this specific finding? Similarly, for the AU→AU co-occurrence in BP4D, the strong correlation between AU6 and AU14 warrants further explanation.
>
> This phenomenon is exactly what motivated us to design CausalAffect: the visualizations reveal fundamental limitations of conventional correlation-based approaches and explain why causal modeling is necessary.
>
> - For the **GNN-learned correlation map**, the almost “everyone-connected-to-everyone” pattern is not a desirable property but a symptom of the discriminative-driven GNN: it is trained only to minimize expression classification loss, so it tends to aggregate information from nearly all AUs for each expression and ends up learning a dense affinity graph that is neither sparse nor human-interpretable.
> - Similarly, the **AU→AU co-occurrence in BP4D** (including the strong AU6–AU14 entry) is computed by simply counting how often each pair of AUs is simultaneously active ($AU_i = 1$ and $AU_j = 1$) over all BP4D samples and normalizing these counts. By construction, this yields a **symmetric** co-occurrence matrix that reflects only how frequently two AUs appear together. Such statistics are heavily influenced by dataset and protocol-specific elicitation patterns, and they **cannot** encode directionality, excitatory vs. inhibitory effects, or psychologically meaningful causal structure.
>
> CausalAffect is designed precisely to address these issues: instead of relying on dense GNN affinities or raw co-occurrence, we explicitly learn **directed and signed AU→AU and AU→Expression causal relations** under causal and counterfactual constraints. For example (as shown in main paper Figure 2 left), AU6 (Cheek Raiser) is more strongly connected to “cheek/eye-area” units such as AU7 (Lid Tightener); AU12 (Lip Corner Puller) connected to mouth corners AU such as AU14 (Dimpler), which aligns much better with human interpretations than the symmetric co-occurrence between AU6 and AU14 in DISFA.

---

### Author Response · Authors · 2025-12-02
**Overall Response to Reviewers’ Comments**

We sincerely thank all reviewers for their thoughtful, detailed, and constructive feedback, as well as for their positive recognition of our contributions. Reviewers emphasized that CausalAffect is the ***first*** approach for causal discovery in facial affect data (AkV1); ***moves beyond*** correlation toward psychologically plausible mechanisms that model both excitatory and inhibitory relations (Hgze); weakly supervised design that ***reduces reliance*** on scarce jointly annotated datasets (Hgze, AkV1); and delivers ***compelling*** empirical results, including cross-dataset AU–Expression relations that align well with existing findings (Sd3y); offers extensive experiments and ablations with ***promising performance*** against competitive baselines (6Lzr), and is ***clearly written, with informative visualizations*** and case studies that make the learned causal relations easy to interpret (6Lzr, AkV1).

Below we summarize the main cross-cutting concerns and how we have addressed them in the revised manuscript and appendix, for more detailed and reviewer-specific responses, please refer to our individual replies in the official review sections.

- **Causal framing, formal assumptions, and AU→Expression vs. AU→AU structure (AkV1)**
    We ground CausalAffect in **cognitive psychology and neuroscience**, drawing on FACS, Scherer’s Component Process Model (CPM), and cross-cultural theories, and we make this explicit via a formal SEM-style causal cascade. We clearly distinguish AU→Expression as a **DAG-like causal pathway** from AU→AU as **positive reciprocal causation** driven by muscle synergies, where enforcing strict DAG would be physiologically inappropriate. We further introduce a **two-level causal hierarchy** that captures population-level causal regularities while allowing individual and cultural variability through a residual, sample-adaptive correction.

- **Novelty vs. prior GNN approaches and conceptual contribution (Sd3y, AkV1)**
    We add an expanded **Related Work** and detailed **Appendix H** contrasting CausalAffect with prior correlation-based AU graphs. We clarify that these methods learn undirected, non-polar AU-only correlations, while CausalAffect is (to our knowledge) the first to jointly learn **directed, polarity-aware** AU→AU and AU→Expression causal graphs under **weak supervision** across disjoint datasets. We also highlight in experiments that even under **same-dataset settings**, CausalAffect consistently outperforms strong GNN baselines on BP4D and DISFA.

- **System complexity, hyperparameters, and reproducibility (Hgze)**
    CausalAffect is modular and weakly coupled, with each loss weight acting within a specific module, which **limits gradient interference**. We add **comprehensive sensitivity analyses** (Appendix J) showing that performance is **stable over broad ranges** of weights and that these hyperparameters primarily influence convergence speed rather than performance.

- **“More data” vs. causal mechanism (6Lzr, Sd3y, AkV1)**
    We provide a comparison table of MTL and CausalAffect under **identical backbones and training data**. Simply adding datasets in standard MTL often degrades AU performance (negative transfer / gradient conflict), whereas CausalAffect consistently improves all  benchmarks, indicating that gains come from the **causal graph mechanism rather than extra data**.

- **Psychological plausibility and validation of new relations (Hgze, AkV1, 6Lzr)**
    Beyond literature-based comparisons, we conduct a **five-expert validation study** (Appendix I) on a mixed set of FACS-supported edges, CausalAffect-discovered relations, and random controls. CausalAffect relations achieve a much higher plausibility rate than random controls and approach literature-supported relations, with **high majority and high-confidence endorsement** and **reliable expert agreement**. Crucially, **inhibitory** AU→AU and AU→Expression edges also receive consistently high plausibility scores, supporting that they reflect psychologically meaningful mechanisms rather than dataset artifacts.

- **Clarity, presentation, and organization (Sd3y, 6Lzr, AkV1)**
    We improve readability by letting each major section **start with an intuitive overview**. We also expand and reorganize Related Work, explicitly stating **four core limitations** of prior methods and mapping each to the corresponding design choice in CausalAffect.

- **Temporal evolution in video frames (AkV1, 6Lzr)**
    We add **temporal visualizations** (Appendix K) of sample-adaptive AU→AU graphs over video frames, showing that the learned causal structure evolves coherently with the underlying dynamics and is naturally compatible with a time-varying formulation.

Overall, we believe these clarifications address the main concerns raised by all reviewers and more clearly communicate the novelty, causal foundations, psychological validity, and practical impact of CausalAffect.

---

### Meta-Review · Area_Chair_WMfF · 2025-12-23

**Summary:**

The reviews were mixed: two reviewers recommend accept, while two reviewers recommend reject, with the remaining review(s) around the borderline. Supportive reviewers valued the paper’s motivation and the extensive experiments/visualizations, but the negative reviewers raised decisive concerns about whether the reported improvements and discovered relations are convincingly attributable to the proposed causal mechanism. My recommendation is reject, primarily due to unresolved issues on fair comparison and evidence strength.

**Reviewer Concerns:**

Addressed by rebuttal:

 Authors added a Related Work section and expanded discussion, which improves context and readability.

Commit to reposition Figure 2 to improve flow.

Still outstanding:

The core concern that the best configuration uses additional training data, while the no-extra-data setting is close to strong baselines, remains only partially addressed.

The reviewer explicitly asked for significance testing to support claimed improvements; this still appears missing or insufficiently emphasized/quantified.

Questions about “weird” correlation visualizations and specific strong correlations are not clearly resolved in the provided discussion excerpt, leaving uncertainty about whether learned relations reflect artifacts.

**Reviewer Scores:**

6Lzr: 6 → 6

AkV1: 4 → 4

Sd3y: 2 → 2

Hgze: 8 → 8

---

### Decision · Program_Chairs · 2026-01-26

Reject